# Independent and interacting value systems for reward and information in the human brain

Irene Cogliati Dezza[1,2,3]*, Axel Cleeremans[1], William H Alexander[3,4,5,6]

[1]Center for Research in Cognition & Neurosciences, ULB Neuroscience Institute, Université Libre de Bruxelles, Brussels, Belgium; [2]Department of Experimental Psychology, Faculty of Brain Sciences, & The Max Planck UCL Centre for Computational Psychiatry and Ageing Research, University College London, London, United Kingdom; [3]Department of Experimental Psychology, Ghent University, Ghent, Belgium; [4]Center for Complex Systems and Brain Sciences, Florida Atlantic University, Boca Raton, United States; [5]Department of Psychology, Florida Atlantic University, Boca Raton, United States; [6]Brain Institute, Florida Atlantic University, Boca Raton, United States

**Abstract** Theories of prefrontal cortex (PFC) as optimizing *reward* value have been widely deployed to explain its activity in a diverse range of contexts, with substantial empirical support in neuroeconomics and decision neuroscience. Similar neural circuits, however, have also been associated with information processing. By using computational modeling, model-based functional magnetic resonance imaging analysis, and a novel experimental paradigm, we aim at establishing whether a dedicated and independent value system for information exists in the human PFC. We identify two regions in the human PFC that independently encode reward and information. Our results provide empirical evidence for PFC as an optimizer of independent information and reward signals during decision-making under realistic scenarios, with potential implications for the interpretation of PFC activity in both healthy and clinical populations.

*For correspondence:
irene.cogliatidezza@gmail.com

Competing interest: The authors declare that no competing interests exist.

## Editor's evaluation

The paper proposes independent and dedicated reward value and information value systems that drive choice. The paper uses a combination of computational modeling and fMRI to provide evidence for these systems in the medial frontal cortex, respectively situated them in vmPFC and dACC.

## Introduction

A general organizational principle of *reward* value computation and comparison in prefrontal cortex (PFC) has accrued widespread empirical support in neuroeconomics and decision neuroscience (*Rangel et al., 2008*; *Doya, 2008*; *Montague et al., 2006*). Other perspectives, however, have suggested the existence of a second, independent value system for optimizing *information* within PFC (*Friston, 2010*; *FitzGerald et al., 2015*; *Friston, 2003*). Here, we aim at establishing whether a dedicated and independent value system for information actually exists in the human PFC. Characterizing independent value signals in the brain can offer insights into many disorders characterized by both reward and information-seeking abnormalities, such as schizophrenia (*Martinelli et al., 2018*), depression (*Hildebrand-Saints and Weary, 2016*), and addiction (*Dezza et al., 2021*).

Substantial empirical evidence in neuroeconomics and decision neuroscience (*Rangel et al., 2008*; *Doya, 2008*; *Montague et al., 2006*; *Bartra et al., 2013*; *Lopez-Persem et al., 2020*) suggests that PFC computes a cost-benefit analysis in order to optimize the net value of rewards (*Rangel et al., 2008*; *Doya, 2008*; *Montague et al., 2006*; *Rushworth et al., 2012*). PFC subregions, such as ventro-medial PFC (vmPFC) and dorsal anterior cingulate cortex (dACC), appear to encode reward signals across a wide range of value-based decision-making contexts, including foraging (*Kolling et al., 2012*; *Shenhav et al., 2016*), risk (*Kolling et al., 2014*), intertemporal (*Wittmann et al., 2016*; *Boorman et al., 2013*), and effort-based choice (*Arulpragasam et al., 2018*; *Skvortsova et al., 2014*). Inter-estingly, these regions are also activated when people seek information (*Charpentier and Cogliati Dezza, 2021*). VMPFC, for example, encodes the subjective value of information (*Kobayashi and Hsu, 2019*) as well as anticipatory information signals indicating that a reward will be received later on *Iigaya et al., 2020*. VMPFC also correlates with ongoing uncertainty during exploration tasks (*Trudel et al., 2021*) and with the upcoming delivery of information (*Charpentier et al., 2018*). DACC is acti-vated when people observe outcomes of options without actively engaging with them (*Blanchard and Gershman, 2018*). Its activity is also associated with perceptual uncertainty (*Jepma et al., 2012*) and it predicts future information sampling to guide actions (*Kaanders et al., 2020*).

This overlapping activity between reward and information suggests that these two adaptive signals are related. Indeed, information signals can be partly characterized by reward-related attributes such as valence and instrumentality (*Kobayashi et al., 2019*; *Sharot and Sunstein, 2020*), while reward signals also contain informative attributes (e.g., winning $50 on a lottery allows the recipient to gain the reward amount but also information about the lottery itself *Wilson et al., 2014*; *Smith et al., 2016*). Because of this 'shared variance' it may not be surprising that the neural substrates underlying information processing frequently overlap with those involved in optimizing reward.

This raises an interesting question as to whether information and reward are really two distinct signals. In other words, is information gain merely a kind of reward that is processed in the same fashion as more typical rewards, or is the calculation of information value at least partially independent

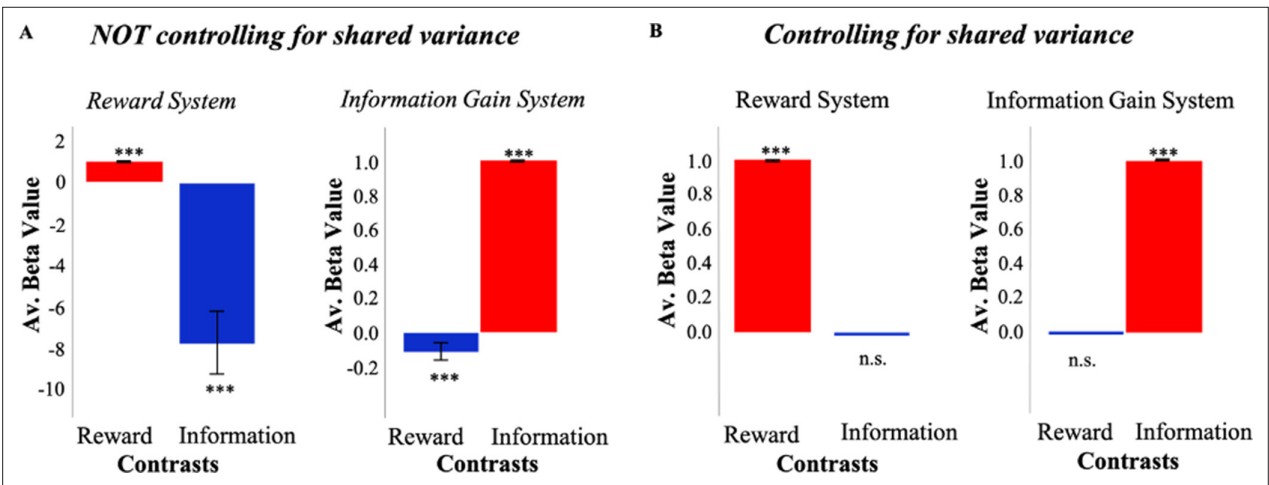

**Figure 1.** Simulations of a model with independent value systems. (**A**) When not controlling for shared variance between reward and information, an RL model which consists of independent reward (RelReward) and information value systems (Information Gain; see Materials and methods for more details) shows overlapping activity between reward and information signals. To simulate activity of the reward system, a linear regression predicting RelReward with RelReward as independent variable was adopted in the reward contrast; while a linear regression predicting RelReward with Information Gain was used in the information contrast. To simulate activity of the information system, a linear regression predicting Information Gain with RelReward as independent variable was adopted in the reward contrast; while a linear regression predicting Information Gain with Information Gain as independent variable was adopted in the information contrast. The model was simulated 63 times and model parameters were selected in the range of those estimated in our human sample. The figure shows averaged betas for these linear regressions. A one-sample t-test was conducted to test significance against zero. (**B**) When controlling for the shared variance, reward and information activities from the same RL model do not overlap anymore. To account for the shared variance, RelReward and Information Gain predictors were orthogonalized using serial orthogonalization. We simulated activity for both the reward system and information system in the same fashion as explained in (**A**). The analysis of those activities was however different. In the information contrast, we entered the orthogonalized (with respect to RelReward) Information Gain as an independent variable, while in the reward contrast, we entered the orthogonalized (with respect to Information Gain) RelReward. In all the panels, * is p<0.05, ** is p<0.01, *** is p<0.001. RL, reinforcement Learning.

of reward value computations? While it is possible that information gain may be valuable in the same way as reward, it may also be the case that the apparent overlap in brain regions underlying reward and information processing may be due to the shared variance between these two distinct signals.

In order to assess whether independent information and reward signals could produce overlapping activity, we developed a reinforcement learning (RL) model which consists of information and reward value systems (*Cogliati Dezza et al., 2017*) independently calculating information gain and reward (see Materials and methods). Simulations of this model suggest how functional magnetic resonance imaging (fMRI) analyses might identify overlapping activity between reward and information systems if the 'shared variance' between them is not taken into account (*Figure 1A*). Even though both signals are computed independently by distinct systems, a typical model-based analysis identifies information-related activity in the reward system, and reward-related activity in the information system.

The same simulations suggest how the spurious functional overlap between information and reward systems might be avoided (*Figure 1B*). Rather than regressing model activity against reward and information signals in isolation, orthogonalizing one regressor with respect to the other eliminates the apparent overlap in function between the systems. When information is orthogonalized with respect to reward, the reward system no longer exhibits information effects, and orthogonalizing reward with respect to information eliminates reward effects in the information system.

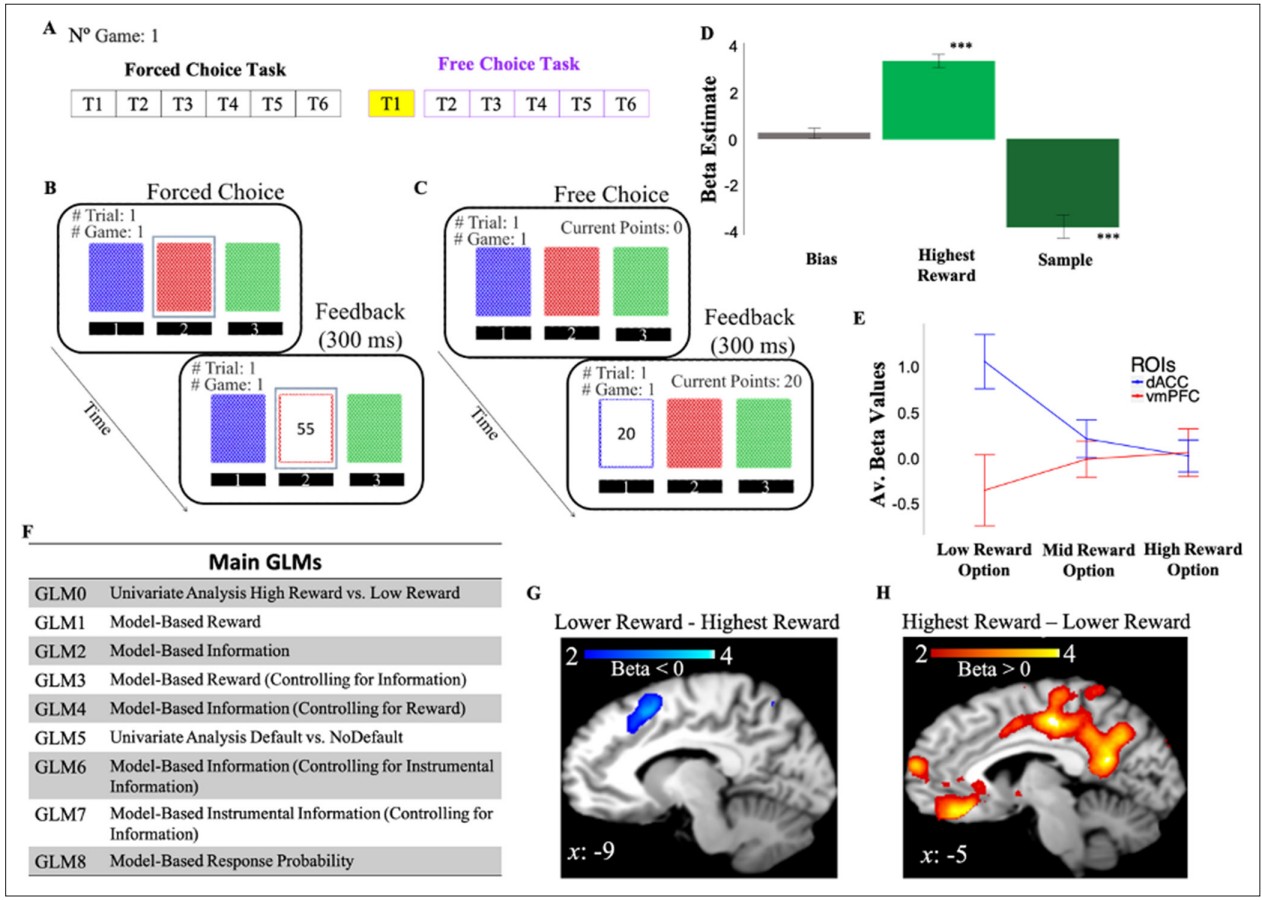

**Figure 2.** Behavioral task and behavior. (**A**) One game of the behavioral task consisted of six consecutive forced-choice trials and from 1 to 6 free-choice trials. fMRI analyses focused on the first free-choice trial (shown in yellow). (**B**) In the *forced-choice task,* participants chose a pre-selected deck of cards (outlined in blue). (**C**) In the *free-choice task,* they were instead free to choose a deck of cards in order to maximize the total number of points. (**D**) Participants' behavior was predicted by both experienced reward (Highest Reward) and the number of times the options were chosen in previous trials (Sample). The figure shows beta weights from a logistic regression with participants' exploitative choices as dependent variable and Highest Reward and Sample as independent variables. Exploitative choices were classified as those choices in which participants chose the option in the first free-choice trial associated with the highest average of points collected during the forced-choice task of the same game. (**E**) DACC and vmPFC activities follow a symmetrical opposite pattern. Activity is split as a function of reward levels (low, mid, and high). (**F**) Main GLMs adopted in the fMRI analyses. (**G**) DACC activity correlates with selecting the lower reward option. (**H**) VMPFC activity correlates with selecting the highest reward option. Activity scale represents z-score. dACC, dorsal anterior cingulate cortex; fMRI, functional magnetic resonance imaging; vmPFC, ventromedial prefrontal cortex.

These results suggest that by controlling for the shared variance between information and reward signals in this fashion, it is possible to establish whether a dedicated and independent value system for information actually exists in human PFC. To do so, we developed a novel behavioral task that allows us to jointly investigate both reward and information-seeking behaviors. Next, we adopted computational modeling techniques to dissociate the relative contribution of reward and information in driving choice behavior and model-based fMRI to localize their activity in the brain.

## Results

### Reward and information jointly influence choices

Human participants made sequential choices among three decks of cards over 128 games, receiving between 1 and 100 points after each choice (*Figure 2*; Materials and methods). The task consisted of two phases (*Figure 2A*): a learning phase (i.e., forced-choice task) in which participants were instructed which deck to select on each trial (*Figure 2B*), and a decision phase (i.e., free-choice task) in which participants made their own choices with the goal of maximizing the total number of points obtained at the end of the experiment (*Figure 2C*). Logistic regression of subjects' behavior on choices made on the first trial of the free-choice task shows that participants' choices were driven by both rewards (mean $\beta$=3.22, $t$(1,19)=12.4, p<10⁻⁹) and information that is the number of time the option was sampled in previous trials (mean $\beta$=−3.68, $t$(1,19)=−7.84, p<10⁻⁶) experienced during the learning phase (*Figure 2D*).

### The gambling task elicits activity in dACC and vmPFC

We first investigated whether our gambling task elicits dACC and vmPFC activity, both being regions involved in reward and information processing (*Charpentier and Cogliati Dezza, 2021*).

We conducted a one sample t-test on the beta weights estimated for *GLM0* which consists of two regressors, one modeling choice onset associated with selection of the highest rewarded options (Highest Reward), and another regressor modeling choice onset associated with lower rewarded options (Lower Reward). This and all subsequent fMRI analyses focus on the time window preceding the first free choices (Materials and methods). Results showed that vmPFC activity is positively correlated with the reward associated with the chosen option (*Highest reward – Lower Reward*; FWE *p*=0.009, uncorr *p*=0.000, voxel extent=203, peak voxel coordinates (–6, 30, –14), $t$(19)=5.48; *Figure 2H*), while dACC/preSMA activity was negatively correlated with the reward of the chosen option (*Lower Reward – Highest Reward*; FWE *p*=0.158, uncorr *p*=0.014, voxel extent=87, peak voxel coordinates (–2, 12, 58), $t$(19)=4.66; *Figure 2G*).

We note that the cluster of activity we identify as 'dACC' spans into supplementary motor areas. Many fMRI studies on value-based decision-making reporting similar activity patterns, however, commonly refer to activity around this area as dACC (*Shenhav et al., 2014*; *Vassena et al., 2020*). Additionally, in the Lower Reward – Highest Reward contrast activity did not survive correction for multiple comparisons. This might be due to individual differences in subjective reward value. We address this issue in the next section by adopting model-based approaches. We conducted, however, a small volume analysis using functionally defined regions taken from FIND lab (Stanford University; https://findlab.stanford.edu/functional_ROIs.html) corresponding to our prior hypotheses. Results show a significant cluster at voxel coordinates (–2, 12, 58) after correcting for multiple comparisons (FWE *p*=0.011).

Overall, these results indicate that our gambling task elicits activity in dACC and vmPFC, and this activity follows a symmetrically opposite pattern (*Figure 2E*).

### Apparent shared activity between reward and information

In the previous section, we showed that our gambling task elicits reward-related activity in both dACC and vmPFC. Here, we test whether this activity relates to both reward and information signals in both regions.

We fitted an RL model with information integration (*Cogliati Dezza et al., 2017*) to participants' behavior to obtain subjective evaluations of reward and information (Materials and methods; *Supplementary file 1*). Our model was better able to explain participants' behavior compared to an RL model without information integration (i.e., where only reward predictions influence choices;

fixed-effect: $BIC_{gkRL}$=391.2, $BIC_{standardRL}$=428.8, for individual BICs, see *Supplementary file 2*; random-effect: $xp_{gkRL}$=1, $xp_{standardRL}$=0) and to predict behavioral effects observed in our sample. For the latter, we simulated our model using the estimated free parameters and we performed a logistic regression for each simulation predicting the model's choices with reward and information (i.e., the number of times the option was sampled in previous trials) as fixed effects. As observed in our human sample, reward and information were significantly impacting model choices (both $p<10^{-5}$).

Moreover, the degree of accuracy of the fitting procedure was inspected by running a parameter recovery analysis. We simulated data from our model using the parameters obtained from the fitting procedure, and we fit the model to the simulated data to obtain the estimated parameters. We then ran a correlation for each pair of parameters (*Wilson and Collins, 2019*). This revealed high correlation coefficients for $\alpha$ ($r$=0.8, $p<10^{-3}$), $\beta$ ($r$=0.8, $p<10^{-3}$), $\omega$ ($r$=0.6, $p$=0.006), and $\gamma$ ($r$=0.6, $p$=0.002).

Next, we computed subjective evaluations of reward as relative reward values (*RelReward*; Materials and methods) and subjective evaluations of information as the prospective information gain for selecting a deck, derived from the behavioral fits of our RL model to participants' data (*Information Gain*; Materials and methods).

## Assessing alternative definitions of reward

Before assessing whether activity in vmPFC and dACC relates to both reward and information, we first investigated whether our chosen 'reward computation' (i.e., RelReward) was better able to explain activity in vmPFC than alternative reward computations. First, we compared activity in vmPFC between RelReward and expected reward values (ExpReward) of the chosen option. Results showed that RelReward correlated with vmPFC after controlling for (i.e., orthogonalizing with respect to) ExpReward (*GLM0exprel*: FWE $p<0.001$, voxel extent=1829, peak voxel coordinates (–6, 52, 14), $t(19)$=7.21), while ExpReward after controlling for RelReward did not reveal any activity (*GLM0relexp*). RelReward also best described vmPFC activity when alternative reward covariates such as the maximum value

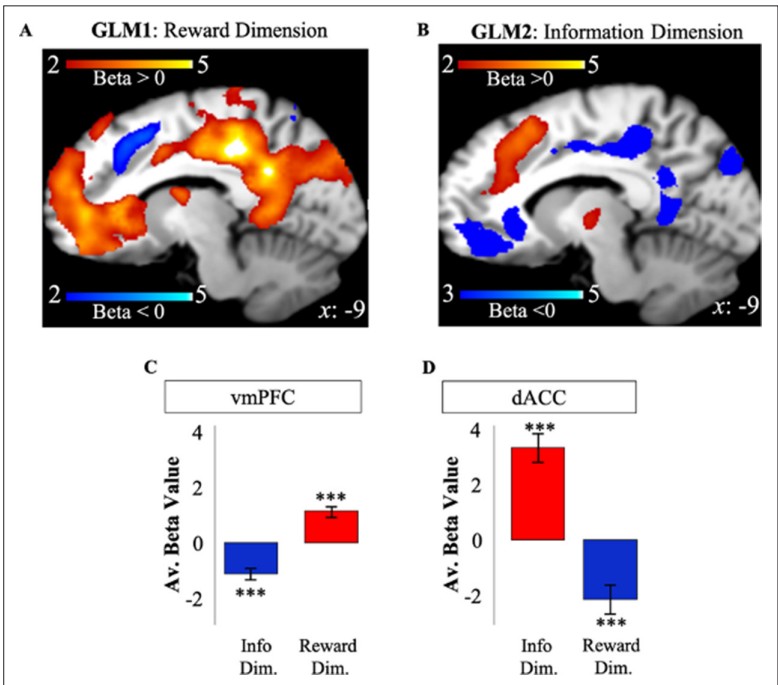

**Figure 3.** Apparent overlapping activity between reward and information. (**A**) VMPFC positively correlated with model-based relative reward value for the selected option (in red), while dACC negatively correlated with it (in blue). (**B**) DACC (in red) positively correlated with model-based information gain, while vmPFC negatively correlated with it (in blue). Activity scale represents z-score. (**C**) Averaged BOLD beta estimates for vmPFC in GLM1 (Reward Dim.=Reward Dimension) and GLM2 (Info Dim.=Information Dimension). (**D**) Averaged BOLD beta estimates for dACC in GLM1 (Reward Dim.=Reward Dimension) and GLM2 (Info Dim.=Information Dimension). In all the panels, * is $p<0.05$, ** is $p<0.01$, *** is $p<0.001$. dACC, dorsal anterior cingulate cortex; vmPFC, ventromedial prefrontal cortex.

of 3 decks (Max Value), the minimum value of the 3 decks (Min Value) and the averaged value of the 3 decks (Averaged Value) were entered in the same GLM allowing modulators to compete for variance (*GLM0rew*: betas from RelReward extracted from the vmPFC ROI identified in GLM2—to avoid biasing our analyses by choosing an ROI defined on RelReward—were significantly higher than the betas for Max Value, $p<0.05$; Min Value, $p<0.05$; Averaged Value, $p<10^{-3}$). Covariates such as the reward value variation for the chosen option (Standard Deviation) and the value of the chosen option minus the value of the best second option (Chosen-Second) were not included in the analysis due to high correlations with RelReward (*Supplementary file 3*). These analyses suggested that RelReward was a better choice, among those considered, to describe reward activity in vmPFC elicited by our task design. We can now address our main question on whether activity in vmPFC and dACC observed in our task relates to both reward and information signals in both regions.

## Overlapping reward and information activity

We tregressed the BOLD signal recorded on the first free-choice trial of each game on RelReward and Information Gain. RelReward and Information Gain were used as the only parametric modulators in two separate GLMs to identify BOLD activity related to reward (*GLM1*) and to information (*GLM2*), respectively, on the first free-choice trial (*Figure 2F*). Unless otherwise specified, all results for these and subsequent analyses are cluster-corrected with a voxel-wise threshold of 0.001.

Activity in vmPFC on the first free-choice trial correlated positively with RelReward (FWE $p<0.001$, voxel extent=1698, peak voxel coordinates (8, 28, −6), $t(19)$=6.62; *Figure 3A*) and negatively with Information Gain (FWE $p<0.001$, voxel extent=720, peak voxel coordinates (−10, 28, −2), $t(19)$=5.36; *Figure 3B*), while activity in dACC was negatively correlated with RelReward (FWE $p=0.001$, voxel extent=321, peak voxel coordinates (6, 24, 40), $t(19)$=4.59; *Figure 3A*) and positively with Information Gain (FWE $p<0.001$, voxel extent=1441, peak voxel coordinates (8, 30, 50), $t(19)$=7.13; *Figure 3B*).

Similar results were obtained when including ExpReward, instead of RelReward, as single parametric modulator (GLM1bis: ExpReward positively correlated with vmPFC – FWE $p<0.001$, voxel extent=530, peak voxel coordinates (−10, 34, 14), $t19)$=5.47 and negatively correlated with dACC – FWE $p=0.001$, voxel extent=205, peak voxel coordinates (8, 22, 40), $t(19$=5.47; *Supplementary file 6*).

## Independent value systems for information and reward after accounting for their shared variance

In the previous section, we showed that both dACC and vmPFC activity relate to both information and reward. However, GLM1 and GLM2 consider only the variance explained by reward and information, respectively. As explained above, simulations of our RL model which consists of independent value systems suggest that fMRI analyses might reveal overlapping activity if the shared variance between the two systems is not taken into account (*Figure 1A*).

To eliminate the shared variance between reward and information as a possible explanation for activity in dACC and vmPFC, we repeated our analyses while controlling for possible shared signals that may underlie our results for GLMs 1 and 2. To do so, we created two additional GLMs to investigate the effect of RelReward after controlling for Information Gain (*GLM3*), and the effect of Information Gain after controlling for RelReward (*GLM4*; Materials and methods). Using the coordinates in vmPFC and dACC observed in GLMs 1 and 2, we conducted an ROI analysis. To avoid double-dipping, we used dACC and vmPFC ROI coordinates from GLM2 (GLM1) to analyze data in GLM3 (GLM4).

Activity in vmPFC remained positively correlated with RelReward ($t(19)$=4.47, $p<0.001$, *Figure 4A*) after controlling for Information Gain in GLM3. In contrast, whereas RelReward was negatively correlated with dACC activity in GLM1 (*Figure 2*), no significant cluster was observed after the removing variance associated with Information Gain in GLM3 ($t(19)$=−1.14, $p=0.27$, *Figure 4A*) as predicted by our model simulations (*Figure 3B*). Similarly, after controlling for the effects of RelReward in GLM4, we observed significant activity in dACC positively correlated with Information Gain ($t(19)$=4.39, $p<0.001$), while we found no correlated activity in vmPFC ($t(19)$=−0.799, $p=0.434$, *Figure 4B*). These results were replicated after contrasting GLM3 and GLM4 using a paired t-test (GLM3>GLM4: vmPFC – FWE $p<0.001$, voxel extent=467, peak voxel coordinates (−4, 52, 16), $t(19)$=5.59; GLM4>GLM3: dACC – FWE $p<0.001$, voxel extent=833, peak voxel coordinates (10, 24, 46), $t(19)$=5.70). We also focus the

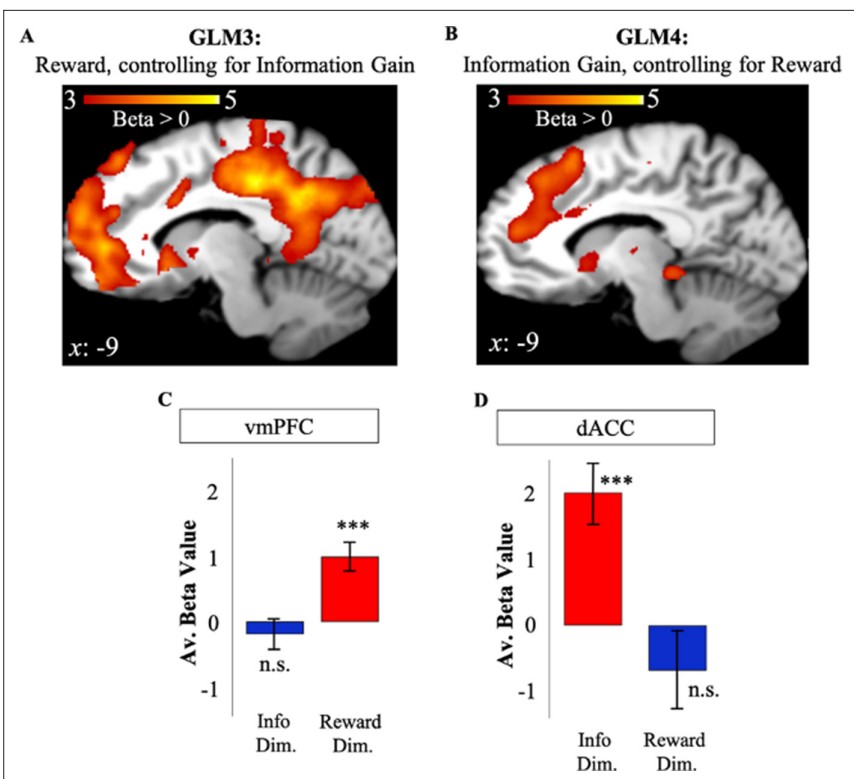

**Figure 4.** Independent value systems for reward and information in PFC. (**A**) After controlling for information (GLM3), vmPFC activity (in red) positively correlated with model-based relative reward value (RelReward), while no correlations were observed for dACC. (**B**) After controlling for reward (GLM4), dACC activity (in red) positively correlated with model-based information gain (Information Gain), while no correlation was observed for vmPFC. Averaged BOLD beta estimates for vmPFC in GLM1 (Reward Dim.=Reward Dimension) and GLM2 (Info Dim.=Information Dimension). (**D**) Averaged BOLD beta estimates for dACC in GLM1 (Reward Dim.=Reward Dimension) and GLM2 (Info Dim.=Information Dimension). In all the panels, * is $p<0.05$, ** is $p<0.01$, *** is $p<0.001$. dACC, dorsal anterior cingulate cortex; vmPFC, ventromedial prefrontal cortex.

analysis on trials in which subjects had equal information about the options (equal information conditions) and we observed no activity in dACC ROI associated to RelReward ($t(19)=-0.0297$, $p=0.9766$).

## Assessing alternative definitions of reward and information

In response to reviewers' suggestions, we repeated the above analysis using alternative definitions of reward and information.

For GLM4, similar results as those reported above were observed when ExpReward was entered in GLM4 instead of RelReward (GLM4bis: $t(19)=4.4$, $p<10^{-3}$; *Supplementary file 7*) and when accounting for covariates for reward that is Average Value, the value of the best second option-mean (value chosen, value third option) and Min Value (*Supplementary file 8*). The rationale behind choosing these reward covariates was as follows: we calculated Variance Inflation Factors (*Craney and Surles, 2002*) for the possible definitions of reward (suggested by reviewers) for each subject, and removed regressors with VIFs (averaged across subjects) above a threshold of 5 (interpreted as 80% of the variability in the regressor can be explained by the rest of the regressors). Due to high multicollinearity amongst the possible reward definitions, VIFs for each definition were indistinguishable (≈infinity). Rather than removing definitions at random, we introduced a small amount of normally-distributed random noise (mean=0, sigma=0.01) to the regressor values and conducted our analysis again. Our intent in introducing noise was to reduce correlations amongst regressors sufficiently to make ordinal judgments about their VIFs.

Using this approach, we removed variables with the highest VIFS iteratively. In order, these were: the value of the chosen reward minus the second highest reward (VIF≈$2×10^{5}$, 99.9995% of variability explained by other regressors), and the maximum reward (VIF≈320, 99.69% of variability explained

by other regressors, after removing Chosen-Second). After removing these two regressors, VIFs for the remaining regressors were all under 5 (average value VIF=3.52, value of the second-best option VIF=1.99, relative value VIF=2.21, and minimum value VIF=3.42). We note that the AverageReward regressor had the highest VIF initially (VIF≈3.6×10$^5$). However, we elected to retain this definition since behavior and brain activity have previously been linked to the overall level of reward across options (*Kolling et al., 2012*; *Cogliati Dezza et al., 2017*). Next, we ran an ROI analysis based on dACC and vmPFC coordinates observed in GLM1. Results showed significant activity in dACC which positively correlates with Information Gain ($t$(19)=2.73, $p$=0.013), while we found no correlated activity in vmPFC as was observed in GLM2 ($t$(19)=−1.5, $p$=0.1503).

For GLM3, similar results as those reported in the previous section were observed when accounting for covariates and an alternative definition of information. In particular, we entered the relative value of information (as it is possible that vmPFC computes the 'relativeness' of the chosen options rather than its reward value) and first choice reaction time as covariates (*GLM3bis*). An ROI analysis based on dACC and vmPFC coordinates observed in GLM2 showed that activity in vmPFC remained positively correlated with RelReward ($t$(19)=4.8, $p$<0.001) after controlling for Information Gain. In contrast, no significant cluster was observed after removing variance associated with Information Gain ($t$(19)=−0.68, $p$=0.505; *Supplementary file 9*).

## Interactions of observed activity with analysis type

To directly test our hypothesis that shared activity between reward and information in dACC and vmPFC is the product of confounded reward and information signals, we conducted a three-way ANOVA with ROI (dACC, vmpFC), Value Type (Information Gain, RelReward), Analysis type (confounded{GLM1&2}, non-confounded {GLM3&4}) and we tested the three-way interaction term. If independent reward and information value signals are encoded in the brain, the two-way interaction (ROI×Value Type) should be significantly modulated by the type of analysis adopted. Results showed a significant three-way interaction $F$(1,19)=37.77, $p$<0.001.

Finally, we check whether accounting for confounded reward and information signals had significant effects in both regions separately. To do so, we ran a two-way ANOVA with Value Type (Information Gain, RelReward), Analysis type (confounded{GLM1&2}, non-confounded {GLM3&4}) for the dACC ROI and a two-way ANOVA with Value Type (Information Gain, RelReward), Analysis type (confounded{GLM1&2}, non-confounded {GLM3&4}) for the vmPFC ROI. Results showed a significant two-way interaction for the dACC ROI ($F$(1,19)=25.7, $p$=0.0001) as well as for the vmPFC ROI ($F$(1,19)=13.7, $p$=0.0015).

Altogether these findings suggest a coexistence of two *independent* value systems for reward and information in the human PFC.

## dACC encodes information after accounting for choice difficulty or switching behavior

Activity in dACC has been often associated with task difficulty/conflict (*Shenhav et al., 2014*; *Botvinick et al., 2001*) or switching to alternative options (*Domenech et al., 2020*). To investigate whether this was the case in our task, we first correlated the standardized estimates of Information Gain with choice reaction times on the first free-choice trials. The correlation was run for each subject and correlation coefficients were tested against 0 using a Wilcoxon signed test. Overall, correlation coefficients were not significantly different from 0 (mean $r$=0.031, SE=0.024; $Z$=145; $p$=0.1429) suggesting that pursuing an option with higher or lower information gain was not associated with higher or lower choice reaction times as predicted by a choice difficulty or conflict account of dACC function. We then entered choice reaction time as additional regressor in GLM4 alongside the across-option standard deviation (i.e., the standard deviation of expected reward values of the three options at time of the first free choice) as an additional proxy for choice difficulty (if the across-option standard deviation is small, choosing among options is harder compared to greater values of an across-option standard deviation; GLM4diff). Moreover, in the same GLM4diff, we entered a switch-stay regressor as proxy for switching behavior (i.e., coded 0 if choices on the first free-choice trial were the same as previous forced trial choices and 1 otherwise). We then used vmPFC and dACC ROIs from GLM1 to analyze the data. Results essentially replicate the above findings with significant activity in dACC which positively correlated with Information Gain ($t$(19)=2.4, $p$=0.027), while we found no significant activity in vmPFC

($t$(19)=−0.757, $p$=0.544; *Supplementary file 10*). No activity in dACC ($t$(19)=0.7498, $p$=0.4625) and vmPFC ROIs ($t$(19)=−0.4423, $p$=0.6633) was observed in the negative contrast.

Finally, in our task, the frequency of choosing the most informative option was higher than the frequency of choosing the two other alternatives (in the unequal condition—when participants were forced to sample options a different number of times, see Materials and methods; mean=64.6%, SD=18%). It is possible, therefore, that in our task, the default behavior was selecting the most informative options, and the switch behavior (or moving away from a default option) was selecting less informative (but potentially more rewarding) options. If this is correct, regions associated with exploration or switching behaviors (e.g., frontopolar cortex; *Daw et al., 2006*; *Zajkowski et al., 2017*) should be activated when participants select a non-default option (i.e., not choosing the most informative option).

We conducted a one sample t-test on the beta weights estimated for *GLM5* which consists of a regressor modeling choice onset associated with the most informative options (Default), and another

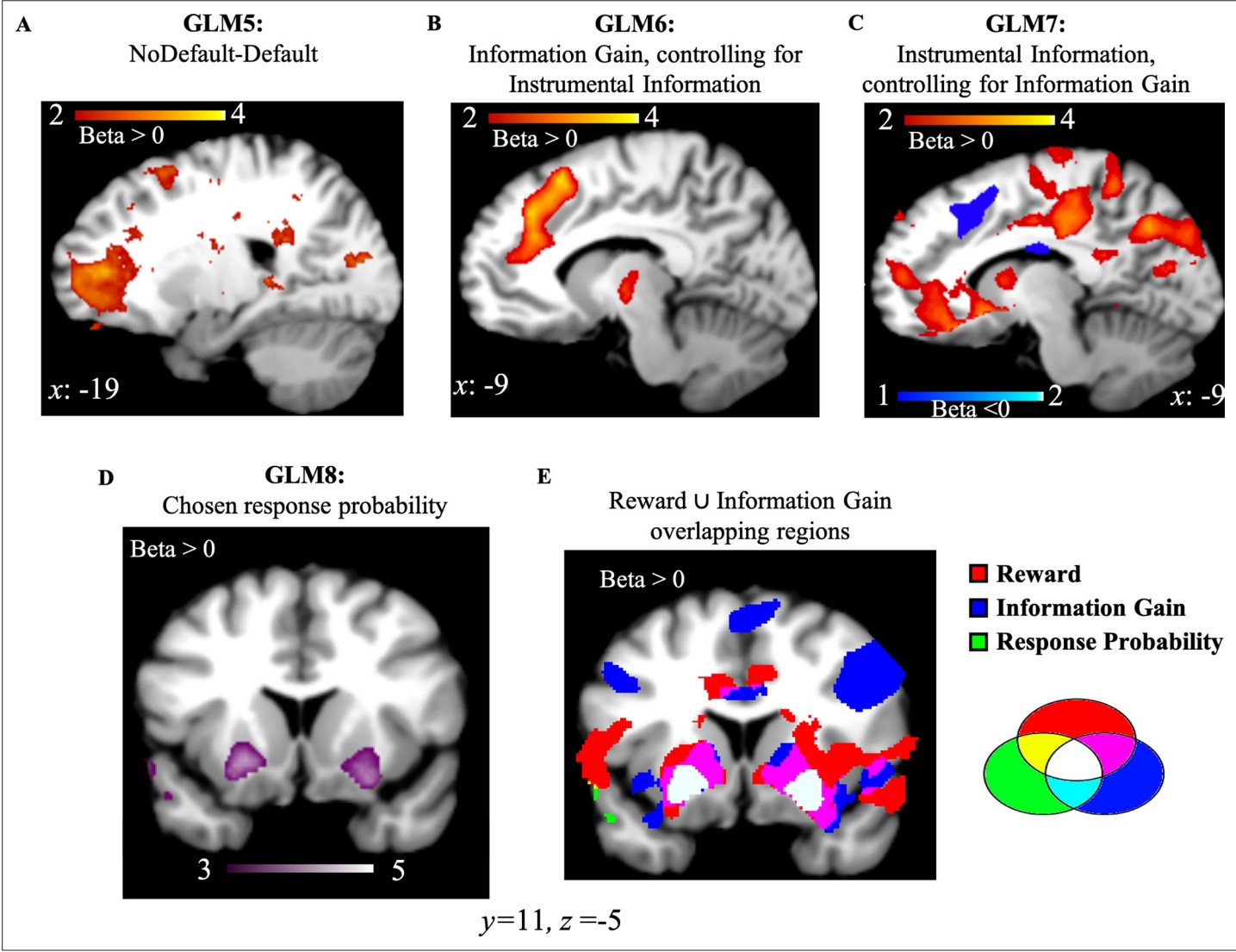

**Figure 5.** NoDefault vs. default behavior, instrumental information and combination of reward and information signals in subcortical regions. (**A**) Activity in the frontopolar region—a region often associated with exploration—correlated with NoDefault behavior (not choosing the most informative options)—Default behavior (choosing most informative options). (**B**) Activity in dACC correlated with Information Gain after controlling for the variance explained by the instrumental value of information. (**C**) Activity in vmPFC and dACC correlated with the instrumental value of information after accounting for the variance explained by Information Gain. (**D**) Activity in the ventral putamen (striatum region) correlated with response probabilities derived from the RL model. (**E**) RelReward, Information Gain, and response probabilities overlap in the striatum region (in white). Activity scale represents z-score. dACC, dorsal anterior cingulate cortex; RL, reinforcement Learning; vmPFC, ventromedial prefrontal cortex.

regressor modeling choice onset associated with the other two options (NoDefault). Activity in fronto-polar cortex was positively correlated with NoDefault (NoDefault – Default; *Figure 5A*; FWE *p*=0.019, voxel extent=148, peak voxel coordinates (–18, 52, 6), *t*(19)=5.71). We additionally analyzed the data using the dACC ROI from GLM1. Results showed no activity in the selected ROI (*t*(19)=0.664, *p*=0.514). First, this suggests that moving away from a default option in our task is associated with not choosing the most informative options. In other words, seeking information was decorrelated from moving away from a default option. Second, it additionally suggests that dACC activity elicited in our task is associated with the value of information, and not with moving away from a default option.

We would like to acknowledge, however, that while dACC activity associated to Information Gain in our task is not affected by proxies of exploratory decisions (e.g., switch-stay analysis and Default vs. NoDefault analysis), our task cannot dissociate decisions to explore to gain information (i.e., directed exploration) and Information Gain. This is because Information Gain and directed exploration in our task describe the same thing—picking the option about which least is known.

## Activity in dACC signals both the non-instrumental and instrumental value of information

In the previous sections, we showed that the value of information was independently encoded in dACC after accounting for reward, choice difficulty, and switching strategy. However, in our task, different motives may drive participants to seek information (*Sharot and Sunstein, 2020*). Information can be sought for its usefulness (i.e., instrumental information): the acquired information can help with the goal of maximizing points by the end of each game. Alternatively, information can be sought for its non-instrumental benefits including novelty, curiosity, or uncertainty reduction. Here, we tested whether the value of information independently encoded in dACC relates to the instrumental value of information, to its non-instrumental value, or to both.

We computed the instrumental value of information (Instrumental Information) by implementing a Bayesian learner and estimating the Bayes optimal long-term value for the option chosen by participants on the first free-choice trial (Materials and methods). We first entered Instrumental Information and Information Gain in a mixed logistic regression predicting first free choices (in the Unequal Information Condition; Materials and methods) with Instrumental Information and Information Gain as fixed effects, subjects as random intercepts and 0 + Instrumental Information + Information Gain | subjects as random slopes. Choices equal 1 when choosing the most informative options (i.e., the option never selected during the forced-choice task), and 0 when choosing options selected four times during the forced-choice task. We found a positive effect of Information Gain (beta coefficient=71.7±16.76 (SE), *z*=4.28, *p*=10$^{-5}$) and a negative effect of Instrumental Information (beta coefficient=–2.3±0.463 (SE), *z*=–4.98, *p*<10$^{-6}$) on most informative choices.

We then entered Instrumental Information and Information Gain as parametric modulators into two independent GLMs. We investigated the effects of Information Gain after controlling for (orthogonalized with respect to) Instrumental Information (*GLM6*) and the effects of Instrumental Information after controlling for (orthogonalized with respect to) Information Gain (*GLM7*; Materials and methods). Next, we ran an ROI analysis based on dACC and vmPFC coordinates observed in GLM1 (as both GLMs 6 and 7 investigate different forms of information, the ROI from GLM1 was selected to avoid double-dipping). Results showed that activity in dACC was positively correlated with Information Gain (after controlling for Instrumental Information; *t*(19)=5.56, *p*<10$^{-4}$) while no significant vmPFC activity was found, *t*(19)=–1.69, *p*=0.108. On the contrary, Instrumental Information (after controlling for Information Gain) positively correlated with vmPFC (*t*(19)=4.83, *p*<10$^{-4}$; *Figure 5B*) and negatively with dACC (*t*(19)=–2.92, *p*=0.009) (*Figure 5C*). These results suggest that dACC encodes both instrumental information signals (with a negative correlation) and non-instrumental information signals (what is left after accounting for the variance explained by Instrumental Information in the Information Gain signal), with the non-instrumental value of information solely encoded in this region while the instrumental value of information encoded in both dACC and vmPFC.

## Reward and information signals combine in the striatum

While distinct brain regions independently encode reward and information, these values appear to converge at the level of the basal ganglia. In a final analysis (*GLM8*), we entered choice probabilities derived from the RL model (where reward and information combine into a common option value;

*Equation 4* in Materials and methods) as a single parametric modulator, and we observed positively-correlated activity in bilateral dorsal putamen (striatum region; right: FWE *p*<0.01, voxel extent=238, peak voxel coordinates (22, 16, −6), *t*(19)=5.59; left: FWE *p*<0.01, voxel extent=583, peak voxel coordinates (–26, 8, –10), *t*(19)=5.89; *Figure 4D*. The negative contrast revealed no activity at uncorr *p*<0.001. Additionally, dorsal putamen overlaps with voxels passing a threshold of *p*<0.001 for overall brain effects of both RelReward and Information Gain from GLMs 3 and 4 (*Figure 4E*).

## Discussion

Information and reward are key behavioral drives. Here, we show that dedicated and independent value systems encode these variables in the human PFC. When the shared variance between reward and information was taken into account, dACC and vmPFC distinctly encoded information value and relative reward value of the chosen option, respectively. These value signals were then combined in subcortical regions that could implement choices. These findings are direct empirical evidence for a dedicated information value system in human PFC independent of reward value.

Activity in the brain suggests that the opportunity to gain information relies on similar neural circuitry as the opportunity to gain rewards (*Bromberg-Martin and Hikosaka, 2009*; *Kang et al., 2009*; *Kobayashi and Hsu, 2019*; *Charpentier et al., 2018*; *Smith et al., 2016*; *Bromberg-Martin and Hikosaka, 2011*; *Tricomi and Fiez, 2012*; *Gruber et al., 2014*; *Jessup and O'Doherty, 2014*; *Blanchard et al., 2015*) even when information has no instrumental benefits (*Tricomi and Fiez, 2012*; *Gruber et al., 2014*; *Charpentier et al., 2018*). Here, we show that the overlapping activity in PFC between these two adaptive signals elicited by our task design is only observed if their shared variance is not taken into account. In particular, in two independent GLMs—one with relative reward value and the other one with information value as a single parametric modulator—activity associated with reward and information activated both vmPFC and dACC. This overlapping activity might be explained by the fact that information signals are partly characterized by reward-related attributes such as valence and instrumentality *Kobayashi et al., 2019*; *Sharot and Sunstein, 2020*, while reward signals also contain informative attributes (e.g., winning $50 on a lottery allows the recipient to gain the reward amount but also information about the lottery itself; *Wilson et al., 2014*; *Smith et al., 2016*).

When eliminating the variance shared between reward and information as a possible explanation of activity, we showed that dACC activity correlated with information value but *not* with immediate reward value, while vmPFC activity correlated with reward value but *not* with information value. This was true even when controlling for covariates including alternative definitions of reward, switch strategy, and choice difficulty. Our finding that activity in dACC positively correlates with the information value of the chosen option suggests the existence of a dedicated system for information in the human PFC independent of the reward system. Our control analysis additionally shows that dACC encodes both the non-instrumental and the instrumental value of information. However, the instrumental value of information also elicits activity in reward regions in line with previous work on a common neural code between reward and information (*Kobayashi and Hsu, 2019*). The fact that neural substrates underlying information gain do not always overlap with those involved in optimizing reward suggests that, in fact, information and reward are not the same type of signals. In other words, information is valuable on its own, independent of its rewarding attributes.

These findings support theoretical accounts such as active inference (*Friston, 2010*; *Friston, 2003*; *Friston, 2005*) and certain RL models (e.g., upper confidence bound; *Auer et al., 2002*; *Wilson et al., 2014*; *Cogliati Dezza et al., 2017*) which predict independent computations for information value (epistemic value) and reward value (extrinsic value) in the human brain. Consistent with our findings, the activity of single neurons in the monkey orbitofrontal cortex independently and orthogonally reflects the output of the two value systems (*Blanchard et al., 2015*). Therefore, our results may highlight a general coding scheme that the brain adopts during decision-making evaluation.

Moreover, our results are in line with recent findings in monkey literature that identified populations of neurons in ACC which selectively encode the non-instrumental value of information (*White et al., 2019*) and are involved in tracking how each piece of information would reduce uncertainty about future actions (*Hunt et al., 2018*). Additionally, they are also consistent with computational models of PFC which predict that dACC activity can be primarily explained as indexing prospective information about an option independent of reward value (*Alexander and Brown, 2011*; *Alexander*

*and Brown, 2018*; *Behrens et al., 2007*). DACC has often been associated with conflict (*Botvinick et al., 2001*) and uncertainty (*Silvetti et al., 2013*), and recent findings suggest that activity in this region corresponds to unsigned prediction errors, or 'surprise' (*Vassena et al., 2020*). Our results enhance this perspective by showing that the activity observed in dACC during decision-making can be explained as representing the subjective representation of decision variables (i.e., the information value signal) elicited in uncertain or novel environments.

It is worth highlighting that other regions might be involved in processing information-related components of the value signal not elicited by our task. In particular, rostrolateral PFC signals the change in relative uncertainty associated with the exploration of novel and uncertain environments (*Badre et al., 2012*; *Tomov et al., 2020*). Neural recordings in monkeys also showed an interconnected cortico-basal ganglia network that resolves uncertainty during information-seeking (*White et al., 2019*). Taken together, these findings, among others (*Charpentier and Cogliati Dezza, 2021*), highlight an intricate and dedicated network for information, independent of reward. Further research is therefore necessary to map this independent network in the human brain and understand to what extent this network relies on neural computations so far associated with reward processing (e.g., dopaminergic modulations; *Bromberg-Martin and Hikosaka, 2009*; *Bromberg-Martin and Hikosaka, 2011*; *Vellani et al., 2021*).

Our finding that vmpFC positively correlates with the relative reward value of the chosen option is in line with previous research that identifies vmPFC as a region involved in value computation and reward processing (*Smith and Delgado, 2015*). VmPFC appears not only to code reward-related signals (*Chib et al., 2009*; *Kim et al., 2011*; *Hampton et al., 2006*) but to specifically encode the relative reward value of the chosen option (*Boorman et al., 2009*), in line with the results of our study.

Our results further suggest that these independent value systems interact in the striatum, consistent with its hypothesized role in representing expected policies (*Friston et al., 2015*) and information-related cues (*Bromberg-Martin and Hikosaka, 2009*; *Charpentier et al., 2018*; *Kobayashi and Hsu, 2019*; *Bromberg-Martin and Monosov, 2020*). The convergence of reward and information signals in the striatum region is also consistent with the identification of basal ganglia as a core mechanism that supports stimulus-response associations in guiding actions (*Samejima et al., 2005*) as well as recent findings demonstrating distinct corticostriatal connectivity for affective and informative properties of a reward signal (*Smith et al., 2016*). Moreover, activity in this region was computed from the softmax probability derived from our RL model, consistent with previous modeling work that identified the basal ganglia as the output of the probability distribution expressed by the softmax function (*Humphries et al., 2012*).

Taken together, by showing the existence of independent value systems in the human PFC, this study provides the empirical evidence in support of a theoretical work aimed at developing a unifying framework for interpreting brain functions. Additionally, we individuated a dedicated value system for information, independent of reward. Overall, our results suggest a new perspective on how to look at decision-making processes in the human brain under realistic scenarios, with potential implications for the interpretation of PFC activity in both healthy and clinical populations.

## Materials and methods

### Participants

Twenty-one right-handed, neurologically healthy young adults were recruited for this study (12 women; aged 19–29 years, mean age=23.24). Of these, one participant was excluded from the analysis due to problems in the registration of the structural $T_1$ weighted MPRAGE sequence. The sample size was based on previous studies (e.g., *Kolling et al., 2012*; *Boorman et al., 2013*; *Shenhav et al., 2014*). Participants also presented normal color vision and absence of psychoactive treatment. The entire group belonged to the Belgian Flemish-speaking community. The experiment was approved by the Ethical Committee of the Ghent University Hospital and conducted according to the Declaration of Helsinki. Informed consent was obtained from all participants prior to the experiment.

### Procedure

Participants performed a gambling task where on each trial choices were made among three decks of cards (*Cogliati Dezza et al., 2017*; *Figure 1*). The gambling task consisted of 128 games. Each

game contains two phases: a *forced-choice task* where participants selected options highlighted by the computer for six consecutive trials, and a *free-choice task* where participants produced their own choices in order to maximize the total gain obtained at the end of the experiment (from 1 to 6 trials, exponentially inversely distributed such that subjects were most frequently allowed to make six free choices). The free-choice trial length or horizon was not cued to participants. Therefore, in each game, participants were not aware of the free choice horizon. We have already shown that the choice horizon does not affect participants' choices on this task when the horizon is not cued (*Cogliati Dezza et al., 2017*; *Cogliati Dezza et al., 2019*). In the forced-choice task, participants were forced to either choose each deck two times (*equal information condition*), or to choose one deck four times, another deck two times, and zero times for the remaining deck (*unequal information condition*). The use of the forced-choice task allows us to orthogonalize available information and reward delivered to participants in the first free-choice trial. For this reason, the focus of our fMRI analyses is on the first free-choice of each game (resulting in 128 trials for the fMRI analyses). We adopted, however, trial-by-trial fMRI analyses (treating equal information condition and unequal information condition altogether) to have a better estimate of the neural activity over the overall performance and modeling method to independently compute reward and information signals.

On each trial, the payoff was generated from a Gaussian distribution with a generative mean between 10 and 70 points and standard deviation of 8 points. The generative mean for each deck was set to a base value of either 30 or 50 points and adjusted independently by ±0, 4, 12, or 20 points with equal probability, to avoid the possibility that participants might be able to discern the generative mean for a deck after a single observation. The generative mean for each option was stable within a game but varied across games. In 50% of the games, the three options had the same generative mean (e.g., 50, 50, and 50), while they had different means in the other half of the games. In 25% of these latter games, the means differed so that two options had the same generative mean with high values and the third option had a different generative mean with low values (e.g., 70, 70, and 30). In 75% of these latter games, two options had the same generative mean with low values and the third option had a different generative mean with high values (e.g., 30, 30, and 30).

Participants' payoff on each trial ranged between 1 and 100 points and the total number of points was summed and converted into a monetary payoff at the end of the experimental session (0.01 euros every 60 points). Participants were told that during the forced-choice task, they may sample options at different rates, and that the decks of cards did not change during each game, but were replaced by new decks at the beginning of each new game. However, they were not informed of the details of the reward manipulation or of the underlying generative distribution adopted during the experiment. Participants underwent a training session outside the scanner in order to make the task structure familiar to them.

The forced-choice task lasted about 8 s and was followed by a blank screen, for a variable jittered time window (1–7 s). The temporal jitter allows to obtain neuroimaging data at the onset of the first free-choice trial and right before the option was selected (decision window). After participants performed the first free-choice trial, a blank screen was again presented for a variable jittered time window (1–6 s) before the feedback, indicating the number of points earned, was given for 0.5 s and another blank screen was shown to them for a variable jittered time window. As the first free-choice trial was the main trial of interest for the fMRI analysis, subsequent free-choice trials were not jittered.

## Image acquisition

Data were acquired using a 3T Magnetom Trio MRI scanner (Siemens), with a 32-channel radio-frequency head coil. In an initial scanning sequence, a structural $T_1$ weighted MPRAGE sequence was collected (176 high-resolution slices, TR=1550 ms, TE=2.39, slice thickness=0.9 mm, voxel size=0.9×0.9×0.9 mm$^3$, FoV=220 mm, and flip angle=9°). During the behavioral task, functional images were acquired using a $T_2$* weighted EPI sequence (33 slices per volume, TR=2000 ms, TE=30 ms, no inter-slice gap, voxel size=3×3×3 mm$^3$, FoV=192 mm, and flip angle=80°). On average, 1500 volumes per participant were collected during the entire task. The task lasted approximately 1 hr split into 4 runs of about 15 min each.

## Behavioral analysis

### Expected reward value and information value

To estimate participants' expected reward value and information value, we adopted a previously implemented version of a RL model that learns reward values and information gained about each deck during the previous experience—the gamma-knowledge Reinforcement Learning model (gkRL; *Cogliati Dezza et al., 2017*; *Cogliati Dezza et al., 2019*). This model was already validated for this task and it was better able to explain participants' behavior compared to other RL models (*Friston, 2010*).

Expected reward values were learned by gkRL adopting on each trial a simple δ learning rule (*Rescorla and Wagner, 1972*):

$$Q_{t+1,\,j}\left(c\right) =\ Q_{t,j}\left(c\right) + \alpha\ \times\ \delta_{t,j} \tag{1}$$

where $Q_{t,j}\left(c\right)$ is the expected reward value for deck c (=Left, Central, or Right) at trial t and game j and $\delta_{t,j} =\ R_{t,j}\left(c\right) - Q_{t,j}\left(c\right)$ is the *prediction error*, which quantifies the discrepancy between the previous predicted reward values and the actual outcome obtained at trial t and game j.

Information was computed as follows:

$$I_{t,\,j}\left(c\right) = \left(\sum_{1}^{t} i_{t,j}\left(c\right)\right)^{\gamma}$$

$$\text{where, } i_{t,j}\left(c\right) = \begin{cases} 0, & choice \neq c \\ 1, & choice = c \end{cases} \tag{2}$$

$I_{t,\,j}\left(c\right)$, is the amount of information associated with the deck c at trial t and game j. $I_{t,\,j}\left(c\right)$, is computed by including an exponential term γ that defines the degree of nonlinearity in the amount of observations obtained from options after each observation. In other words, γ governs the shape of the information value function: values lower than 1 indicate a concave value function, while values larger than 1 indicate a convex value function. γ is constrained to be >0. Each time deck c is selected, $i_{t,j}\left(c\right)$ takes value of 1, and 0 otherwise. On each trial, the new value of $i_{t,j}\left(c\right)$ is summed to the previous $i_{t-1,1:\,j}\left(c\right)$ estimate and the resulting value is elevated to γ, resulting in $I_{t,\,j}\left(c\right)$.

Before selecting the appropriate option, gkRL subtracts the information gained $I_{t,\,j}\left(c\right)$ from the expected reward value $Q_{t+1,\,j}\left(c\right)$:

$$V_{t,\,j}\left(c\right) =\ Q_{t+1,\,j}\left(c\right) + \left(-I_{t,\,j}\left(c\right) * \omega\right) \tag{3}$$

$V_{t,\,j}\left(c\right)$ is the final value associated with deck c. Here, information accumulated during the past trials scales values $V_{t,\,j}\left(c\right)$ so that increasing the number of observations of one option decreases its final value. The parameter ω constitutes a 'mixing' parameter governing the relative importance of information value and reward value in generating behavior.

In order to generate choice probabilities based on expected reward and information values (i.e., final choice value), the model uses a softmax choice function (*Daw and Doya, 2006*). The softmax rule is expressed as:

$$P\left(c|V_{t,\,j}\left(c_i\right)\right)\ = \frac{\exp\left(\beta \times V_{t,\,j}\left(c\right)\right)}{\sum_{i} \exp\left(\beta \times V_{t,\,j}\left(c_i\right)\right)} \tag{4}$$

where β is the inverse temperature that determines the degree to which choices are directed toward the highest rewarded option. By minimizing the negative log likelihood of $P\left(c|V_{t,\,j}\left(c_i\right)\right)$ model parameters α, β, and w, g, were estimated for participants' choices made during the first free-choice trials. The fitting procedure was performed using MATLAB and Statistics Toolbox Release 2020a function *fminsearch* and its accuracy tested using parameter recovery analysis. As for the fitting procedure, the recovery procedure was run on first free-choice trials. The results of this fit procedure are reported in the *Supplementary file 1*. Model parameters were then used to compute the value of $Q_{t+1,\,j}\left(c\right)$ and $I_{t,\,j}\left(c\right)$ for each participant.

## Instrumental value of information

In order to approximate the instrumental utility of options in our task, we turn to Bayesian modeling. In the simplest case, a decision-maker's choice when confronted with multiple options depends on its beliefs about the relative values of those options. This requires the decision-maker to estimate, based on prior experience, relevant parameters such as the mean value and variance of each option. On one hand, the mean and variance of an option can be estimated through direct experience with that option through repeated sampling. However, subjects may also estimate long-term reward contingencies as well: even if an option has a specific mean reward during one game in our task, subjects may learn an estimate of the range of rewards that options can have even before sampling from any options. Similarly, although subjects may learn an estimate of the variance for a specific option during the forced-choice period, over many games subjects may learn that options *in general* have a variance around a specific value.

To model this, we developed a Bayesian learner that estimates, during each game, the probability distribution over reward and variance for each specific option in that game, and, over the entire experiment, estimates the global distribution over mean reward and variance based on observed rewards from all options. A learner's belief about an option can be modeled as a joint probability distribution over likely values for the mean reward $\mu$ and standard deviation $\sigma$. In order to reduce computational demands when conducting forward searches with the model (see below), sigma values were modeled as the integers from 1 to 25 and the range of rewards was modeled in 10-point increments from 5 to 95. Prior to any exposure to the task, the probability distributions over $\mu$ and $\sigma$ were initialized as a uniform distribution.

To model training received by each subject prior to participation in the experiment, the Bayesian learner was simulated on forced choices from 10 random games generated from the same routine used to generate trials during the experiment. After each choice was displayed, the global probability distribution over $\mu$ and $\sigma$ was adjusted using Bayesian updating:

$$P\left(\mu, \sigma | R\right) \propto P\left(\mu, \sigma\right) * N\left(R | \mu, \sigma\right) \tag{5}$$

where N() is the probability of observing a reward for a normal distribution with a given mean and variance.

Following the initial training period, the model performed the experiment using games experienced by the subjects themselves, that is during the forced-choice period, the model made the same choices and observed the same point values seen during the experiment. To model option-specific estimates, the model maintained three probability distributions over $\mu$ and $\sigma$ corresponding to each option, essentially a local instantiation of the global probability distribution described above. The option-specific distributions were reset to the global prior distribution before each new game, and were updated only after an outcome was observed for that option using the same updating rule described in *Equation 5*.

The Bayesian learner described above learns to estimate the probability distribution over the mean and variance for each option during the forced-choice component of the experiment. If the learner's only concern in the free-choice phase is to maximize reward for the next choice, it would select the option with the highest expected value. However, in our task, subjects are instructed to maximize their total return for a variable number of trials with the same set of options. In some circumstances, it is better to select from under-sampled decks that may ultimately have a higher value than the current best estimate.

To model this, we implemented a forward tree search algorithm (*Ross et al., 2022*; *Ghavamzadeh et al., 2015*) which considers all choices and possible outcomes (states) reachable from the current state, updates the posterior probability distribution for each subsequent state as described above, and repeats this from the new state until a fixed number of steps have been searched. By conducting an exhaustive tree search to a given search depth, it is possible to determine the Bayes optimal choice at the first free-choice trial in our experiment.

In practice, however, it is usually unfeasible to perform an exhaustive search for any but the simplest applications (limited branching factor, limited horizon). In our experiment, the outcome of a choice was an integer from 1 to 100 (# of points), and the model could select from three different options, yielding a branching factor of 300. The maximum number of free-choice trials available on a given game was 6, meaning that a full search would consider 3^8 possible states at the terminal leaves of

the tree. In order to reduce the time needed to perform a forward tree search of depth 6, we applied a coarse discretization to the possible values of $\mu$ and $\sigma$ (i.e., sigma values were modeled as the integers from 1 to 25 and the range of rewards was modeled in 10-point increments from 5 to 95). Although the coarse discretization resulted in somewhat less precise estimates of the distribution over $\mu$ and $\sigma$, this had minimal effect on our calculation of instrumental information (see below): for a single subject, instrumental utility over all trials when values for $\mu$ were discretized into 10 bins correlated at 0.936 ($p<0.001$) and 0.948 ($p<0.001$) when $\mu$ was discretized into 20 and 40 bins. We additionally pruned the search tree during runtime such that any branch that had a probability less than 0.001 of being observed was removed from further consideration.

The value of a state was modeled as the number of points received for reaching that state, plus the maximum expected value of subsequent states that could be reached. Thus, the value of leaf states was simply the expected value of the probability distribution over means (numerically integrated over $\sigma$), while the value of the preceding state was that state's value plus the maximum expected value of possible leaf states:

$$Value(state_t, \mu_t, \sigma_t) = R(state_t) + max\,Value(state_{t+1}, \mu_{t+1}, \sigma_{t+1}) \tag{6}$$

Recursively applying *Equation 6* from the leaf states to the first free-choice trial allows us to approximate the Bayes optimal long-term value for each option (i.e., Bayes Instrumental Value). The Bayes Instrumental Value corresponds to the overall expected reward value of choosing, which includes both reward and information benefit. As the instrumental value of information is the difference between the overall expected reward value of choosing which includes both reward and information benefit (i.e., Bayes Instrumental Value) and the reward value obtained from an option without receiving information (i.e., Reward Value without information), the latter was also computed. To do so, the Bayesian procedure was implemented by constraining the model to not update its belief distribution based on the information provided on the first free-choice trial. Next, expected instrumental value (Instrumental Information) for each option on the first free-choice trial following the forced-choice trials specific to that game was computed as:

$$Instrumental\;Information_{c,j} =$$
$$Bayes\;Instrumental\;Value_{c,j} - Reward\;Value\;without\;information_{c,j} \tag{7}$$

On an additional note, as subjects were not aware of the reward distributions adopted in the task—therefore they might develop different beliefs—the above procedure may not reflect each individual's subjective estimate, rather it reflects an objective estimate of the instrumental value of information.

### fMRI analysis

The first four volumes of each functional run were discarded to allow for steady-state magnetization. The data were preprocessed with SPM12 (Wellcome Department of Imaging Neuroscience, Institute of Neurology, London, UK). Functional images were motion corrected (by realigning to the first image of the run). The structural $T_1$ image was coregistered to the functional mean image for normalization purposes. Functional images normalized to a standardized (MNI) template (Montreal Neurological Institute) and spatially smoothed with a Gaussian kernel of 8 mm full width half maximum.

All the fMRI analyses focus on the time window associated with the onset of the first free trials prior to the choice was actually made (see Procedure). The rationale for our model-based analysis of fMRI data is as follows (also summarized in *Supplementary file 4*). First, in order to link participants' behavior with neural activity, *GLM0* was created with a regressor modeling choice onset associated with highest rewarded options (Highest Reward), and another regressor modeling choice onset associated with lower rewarded options (Lower Reward). Activity related to Highest Reward was then subtracted from the activity associated with Lower Reward (giving a value of 1 and –1, respectively) at the second level. Next, in order to identify regions with activity related to reward and information, we computed the relative value of the chosen deck (RelReward) and the (negative) value of gkrl model-derived information gained from the chosen option (Information Gain). RelReward was computed by subtracting the average expected reward values for the unchosen decks from the expected reward values of the chosen deck $c$ from the gkrl model $Q_{t+1,j}^R(c=1) = Q_{t+1,j}(c=1) - mean\,(Q_{t+1,j}(c=2), Q_{t+1,j}(c=3))$. We adopted a standard computation of relative reward values (*Shenhav et al., 2014*). It has already been

shown that vmPFC represents reward values following the above computation. However, we compare these computations with expected reward values and alternative reward computations (Results). Information Gain was computed as $- I_{t,j}(c)$. The negative value $I_{t,j}(c)$ relates to the information to be gained about each deck by participants. In particular, $- I_{t,j}(c)$ represents the functional form on how information value depends on the number of samples, rather than its absolute term. We have already shown that humans represent information value as computed by our model compared to alternative computations when performing the behavioral task adopted in this study (**Cogliati Dezza et al., 2017**). To note, the parameter $\omega$ is not added to the computation as it describes how much information is relevant with respect to reward, rather the information value on itself (**Supplementary file 11**). Next, we entered RelReward and Information Gain as parametric modulators into GLMS with a single regressor modeling the onset of the first free-choice trial as a 0 duration stick function. In *GLM1*, RelReward was included as a single parametric modulator. In *GLM2*, Information Gain was included as a single parametric modulator. In *GLM3*, two parametric modulators were included in the order: Information Gain, RelReward. In *GLM4*, the same two parametric modulators were included, with the order reversed, that is, RelReward, Information Gain. The intent of GLMs 3 and 4 was to allow us to investigate the effects of the second parametric modulator after accounting for variance that can be explained by the first parametric modulator. In SPM12, this is accomplished by enabling modulator orthogonalization (Wellcome Department of Imaging Neuroscience, Institute of Neurology, London, UK). Under ideal circumstances, results of analyzing the final parametric modulator in a sequence (orthogonalized with respect to all others) should be highly similar to analyses in which no serial orthogonalization is performed (**Mumford et al., 2015**). Practically, however, we have observed differences in the strength of our results with and without serial orthogonalization. While GLMs that include both information and reward regressors may be able to dissociate information and reward signals without the analysis stream we describe here, we elected to adopt serial orthogonalization in order to ensure variance that could be attributed to the final parameter was instead allocated elsewhere.

Additional GLMs were then used for the control analyses: *GLM0exprel* and *GLM0relexp* with ExpReward and RelReward as parametric modulators; and *GLM0rew* where the maximum value of 3 decks (Max Value), the minimum value of the 3 decks (Min Value), the averaged value of the 3 decks (Averaged Value), and RelReward competed for variance. *GLM1bis* with expected reward value (ExpReward) as single parametric modulator; *GLM3bis* with the relative value of information $RI_{t,j}(c) = = I_{t,j}(c=1) - mean\left(I_{t,j}(c=2), I_{t,j}(c=3)\right)$, Information Gain, first choice reaction time (RT) and RelReward as parametric modulators; *GLM4bis* with two parametric modulators ExpReward and Information Gain; *GLM4rew* with additional computations of reward, RelReward and Information Gain as parametric modulators; *GLM4diff* with across option standard deviation aSD=$sd\left(Q_{t+1,j}(c=1), Q_{t+1,j}(c=2), Q_{t+1,j}(c=3)\right)$, RT, switch-stay (i.e., coded 0 if choices on the first free-choice trial where the same as previous trial choices and 1 otherwise), RelReward and Information Gain as parametric modulators; *GLM5* which comprises of a regressor modeling choice onset associated with the most informative options (Default), and another regressor modeling choice onset associated with the other two options (NoDefault); *GLM6* and *7* with Instrumental Information and Information Gain as parametric modulators.

To determine the regions associated with Reward and Information Gain, beta weights for the first (single modulator GLMS) or second (two modulator GLMS) parametric modulators were entered into a second level (random effects) paired-sample t-test. In order to determine activity related to the combination of information and reward value, *GLM8* was created with the softmax probability of the chosen option $P(c/V_{t,j}(c_i))$ modeling the onsets of first free-choices.

Activity for these GLMs is reported either in the Result section or in **Supplementary file 5**.

In order to denoise the fMRI signal, 24 nuisance motion regressors were added to the GLMs where the standard realignment parameters were nonlinearly expanded incorporating their temporal derivatives and the corresponding squared regressors (**Friston et al., 1996**). Furthermore, in GLMS with two parametric modulators, regressors were standardized to avoid the possibility that parameter estimates were affected by different scaling of the models' regressors alongside with the variance they might explain (**Erdeniz et al., 2013**). During the second level analyses, we corrected for multiple comparisons in order to avoid the false positive risk (**Chumbley and Friston, 2009**). We corrected at the cluster level using both FDR and FWE. Both corrections gave similar statistical results therefore we reported only FWE corrections.

## Acknowledgements

Funded by FRS-fNRS, BOF and FWO (ICD), FWO-Flanders Odysseus II Award #G.OC44.13N (W.A.), and AC was partly supported by an Advanced Grant (RADICAL) from the European Research Council.

## Additional information

### Funding

| Funder | Grant reference number | Author |
|---|---|---|
| FWO-Flanders Odysseus 2 | G.OC44.13N | William H Alexander |
| F.R.S.-fNRS | | Irene Cogliati Dezza |
| FWO | | Irene Cogliati Dezza |
| European Research Council | | Axel Cleeremans |

The funders had no role in study design, data collection and interpretation, or the decision to submit the work for publication.

### Author contributions

Irene Cogliati Dezza, Conceptualization, Data curation, Formal analysis, Funding acquisition, Software, Visualization, Writing - original draft, Writing - review and editing; Axel Cleeremans, Writing - review and editing; William H Alexander, Conceptualization, Formal analysis, Funding acquisition, Methodology, Supervision

### Author ORCIDs

Irene Cogliati Dezza http://orcid.org/0000-0002-1212-4751
William H Alexander http://orcid.org/0000-0002-3723-4789

### Ethics

Human subjects: The experiment was approved by the Ethical Committee of the Ghent University Hospital and conducted according to the Declaration of Helsinki. Informed consent was obtained from all participants prior to the experiment.

### Decision letter and Author response

Decision letter https://doi.org/10.7554/eLife.66358.sa1
Author response https://doi.org/10.7554/eLife.66358.sa2

## Additional files

### Supplementary files

• Supplementary file 1. Model estimated parameters from participants' behavior. The table shows parameter estimates after fitting the model to participants' data. Group mean and standard deviation are also reported for each parameter.

• Supplementary file 2. Individual BICs. The table shows individual BIC for both standard RL and gkRL.

• Supplementary file 3. Correlation of covariates with relative reward value. The table shows correlation coefficients between relative reward value and the covariates for each subject.

• Supplementary file 4. GLMs for fMRI data. The table shows the 17 GLMs adopted in the fMRI data analysis all referring to activity associated with the onset of the first-free-choice trial. GLM0 and 5 are the univariate analyses, whereas the other GLMs relate with the model-based analysis.

• Supplementary file 5. Brain activity no reported in the text. The table shows brain activity not reported in the main text. PCC: Posterior Cingulate Cortex; mOFC: medial Orbitofrontal Cortex; aInsula: anterior Insula.

• Supplementary file 6. Brain activity in GLM1bis. The figure shows brain activity in GLM1bis

- Supplementary file 7. Brain activity in GLM4bis. The figure shows brain activity in GLM4bis
- Supplementary file 8. Brain activity in GLM4rew. The figure shows brain activity in GLM4rew.
- Supplementary file 9. Brain activity in GLM3bis. The figure shows brain activity in GLM3bis
- Supplementary file 10. Brain activity in GLM4diff. The figure shows brain activity in GLM4diff
- Supplementary file 11. Information gain and omega parameter.
- Transparent reporting form

## Data availability

Data will be available on OSF upon publication at https://osf.io/e3rp6/.

The following dataset was generated:

| Author(s) | Year | Dataset title | Dataset URL | Database and Identifier |
| --- | --- | --- | --- | --- |
| Dezza CogliatiI, Cleeremans A, Alexander W | 2022 | Independent and interacting value system for reward and information in the human brain | https://osf.io/e3rp6/ | Open Science Framework, OSF.IO/E3RP6 |

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
