## [Editor Report]

The paper proposes independent and dedicated reward value and information value systems that drive choice. The paper uses a combination of computational modeling and fMRI to provide evidence for these systems in the medial frontal cortex, respectively situated them in vmPFC and dACC.

---

## [Decision Letter]

**Decision letter after peer review:**

[Editors’ note: the authors submitted for reconsideration following the decision after peer review. What follows is the decision letter after the first round of review.]

Thank you for submitting your work entitled "Independent and interacting value systems for reward and information in the human prefrontal cortex" for consideration by *eLife*. Your article has been reviewed by 3 peer reviewers, and the evaluation has been overseen by a Reviewing Editor and a Senior Editor. The reviewers have opted to remain anonymous.

Our decision has been reached after consultation between the reviewers. Based on these discussions and the individual reviews below, we regret to inform you that your work will not be considered further for publication in *eLife*.

The study reports evidence from fMRI for independent brain systems that value reward versus information. An elegant free choice task is used during fMRI to show that people value both reward and information. Then, using model-based fMRI and a method of deconfounding information and value, a dissociation is reported between the dACC and vmPFC tracking information and reward, respectively.

The reviewers found this to be an important topic and were in largely in agreement that this dissociation would be an advance of interest to the field. However, the reviewers also identified a substantial number of serious concerns that raised doubt regarding these observations and their interpretation. Some of these are addressable with relatively straightforward additional analyses. However, others might also be addressable, but the path forward is less clear. At the very least, they would require considerable additional work and possibly collection of new data. Thus, after considerable discussion, it was decided that the work required to address these serious concerns was extensive enough that it would be beyond that expected of a revision at *eLife*. As a result, the consensus was to reject the paper. However, given the promise of the paper, the reviewers and editors also agreed that were the paper to be extensively reworked in order to address all of the points raised, then we would be willing to consider the paper again at *eLife* as a new submission.

I have attached the original reviews to this letter, as the reviewers provide detailed points and suggestions there. However, I wish to highlight a few points that the reviewers emphasized in discussion and that led to our decision.

Dissociating instrumental versus non-instrumental utility. To support the main conclusion that dACC does code for information value, not reward (i.e. dACC and vmPFC constitute an independent value system), one would need to show that dACC does not code for long-term reward. However, the key control analysis for ACC between information utility and inherent value is not convincing because of the assumptions it makes about people's beliefs (see comments by R1). However, addressing this might require post-hoc assessments of participant's beliefs or collection of new data. One reviewer suggested an analysis of behavior along these lines:

-describe the task more clearly so that we can understand the generative process,

-find a way to calculate the objective instrumental value of information in the task (or the value that participants should have believed it had, given their instructions).

-fit a model to behavior to estimate subjective information value

-check how much of the subjective value of information can be accounted for its objective instrumental value.

Regardless of the approach taken, it was agreed that addressing this point is essential for the conclusions to be convincing.

Behavioral Fits. R3 raises some detailed concerns regarding the model fits that are serious and might impact interpretation of the model-based ROI analysis. As such, these are quite important to address. During discussion, one reviewer suggested that the problem might be that the omega and γ parameters are trading off and behavior can be fit equally well by both. This is resulting in extreme and variable fits across participants (you can see this in an apparent negative correlation between your omega and γ parameters; Spearman's rho = -0.57, p = 0.008). Other parameters are showing similar issues (α and β, for example). Among these is omega and β (correlation of about -.6), which will make separating reward and information difficult. This reviewer suggests that you consider analyzing the data in terms of the model's total estimated subjective value (i.e. -omega*I), which incorporates the effects of both the omega and the γ parameters. They also suggest that you consider reformulating the choice function to separate reward terms from information terms, for example, applying terms like Β*Q – Omega*I. Regardless of whether you take these suggestions, the model fitting and recoverability of the parameters appears to be a significant issue to address. At a minimum, some recoverability analysis and analysis of the correspondence between model's and subject's behavior will be required.

- Support for two independent systems. Several comments from reviewers highlighted issues that diminish support for the core separable system finding. These are well summarized in R2's comments, but they center around the definition of reward and information terms, correlations between information and reward, no test of for a double dissociation, and the lack of control for other correlated variables.

The reviews contain many suggestions of ways these issues might be addressed. However, in discussion, the reviewers suggested the alternate regressors for reward be included as:

1) second highest option's value (V2) – avg(value chosen, V3)

2) entropy over the three options' values (V1,V2,V3)

In addition to the following covariates:

1) minimum value of the three decks

2) maximum value of the three decks

3) average value of the three decks

4) RT

They also emphasized that at least two models of brain activity should be considered. One that includes raw value (regressors for a chosen option's expected reward and information) and one that includes relative value (relative excepted reward and information). These last approaches are detailed in the comments from reviewers.

- Novelty. R3's comments raised a concern regarding the novelty of the contribution on first reading. However, after discussion, this reviewer came around on this point and was persuaded of its novelty. Nonetheless, placing the present findings the context of prior work locating dissociations between dACC and other brain regions is important. This reviewer has provided important guidance along these lines.

*Reviewer #1:*

Cogliati Dezza and colleagues report fMRI findings showing independent systems for valuing reward vs information. They use an elegant task in which participants freely choose between three decks of cards, after having gone through a sequence of forced choice trials which that control their sampling of the decks and prior information about their reward values. They operationally define information gain in terms of the number of times an option has been previously sampled (under the assumption that fewer samples = more information gain from the next sample) and show that people value both immediate reward and its informational gain. They interestingly show that simple univariate fMRI analysis of either information or reward would be confounded, but report that a multivariate analysis of activity related to the deck being chosen shows that vmPFC activations primarily track reward, dACC tracks information, and striatum tracks the choice probability reflecting its total value.

The key report that information and reward may be valued by separate neural systems in frontal cortex is clearly important and valuable for the field. This would be in contrast to recent studies which primarily reported areas that have similar responses for both information and reward value (Charpentier et al., PNAS 2018; Kobayashi and Hsu, PNAS 2019). This idea is plausible, appealing, and has strong implications for brain function. However, I have some concerns about whether the manuscript is fully convincing.

1. Figure 1 that sets up the paper would benefit from a clear organization where it shows in parallel the two hypotheses, the findings of univariate analysis (i.e. similar under both hypotheses), and crucially, also shows the simulated findings of the improved analysis+experiment advocated in this paper, in the same format (i.e. clearly different under the two hypotheses, presumably by showing only a non-zero red bar for the information system and only a non-zero blue bar for the reward system, and that the estimates of the two effects aren't negatively correlated).

This last simulation seems crucial since their current simulations show the flaws of bad approaches but do not show the validity of their current approach, which needs to be proven to know that we can trust the key findings in Figure 4.

Similarly, I would suggest retitling Figure 3 something like "APPARENT symmetrical opposition…" to make to clear that they are saying that the findings shown in Figure 3A,B are invalid. Otherwise nothing in this figure tells the reader that they are not supposed to trust them.

2. I greatly appreciate their control analysis on a very important point: how much of the information value in behavior and brain activity in this task is due to the information's instrumental utility in obtaining points from future choices (the classic exploration-exploitation tradeoff) vs. its non-instrumental utility (e.g. satisfying a sense of curiosity). It is important to understand which (or both) the task is tapping into, since these could be mediated by different neural information systems. If this dissociation is not possible, it would be important to discuss this limitation carefully in the introduction and interpretation.

I am not sure I understand the current control analysis, and am not fully confident this can be dissociated with this task. The logic of the control analysis is that if dACC is involved in long-term reward maximization its activity should be lower before choices of an unsampled deck in contexts when the two sampled decks gave relatively high rewards, presumably because the unsampled deck is less worth exploring. I think I see where they are coming from, but this is only obviously true if people believe each deck's true mean reward is drawn independently from the true mean rewards of the other two decks. However, this may not be the true setup of the task or how people understand it.

The paper's methods do not describe the generative distributions in enough detail for me to understand them (the methods describe how the final payoff was generated, but not how the numbers from the 6 forced choices were generated, or how the numbers from the forced choices relate to the final payoff). But in their previous interesting behavior paper they used different High Reward and Low Reward contexts where all decks had true means that were High or Low. If they did the same in this paper then I am not sure their control analysis would make sense, since the deck values would be positively correlated.

Broadly speaking, their logic seems to be that if people value information for its instrumental utility there should be a Reward x Information interaction so that they value information more for decks with high expected reward (potential future targets for exploitation) than low expected reward (not worth exploring). I am not sure if this would be strong evidence about whether information value is instrumental because it has been reported that people may also value non-instrumental information in relation to expected reward (e.g. Kobayashi et al., Nat Hum Behav 2019). But if they do want to use this logic, wouldn't it be simpler to just add an interaction term to their behavioral and neural models and test whether it is significant?

Also, their current findings seem to imply a puzzling situation where participants presumably do make use of the information at least in part for its instrumental value in helping them make future choices (as shown a similar task by Wilson et al), but their brain activity in areas sensitive to information only reflects non-instrumental utility. I am not sure how to interpret this.

However they address these, I have two suggestions.

First, whatever test they do to dissociate instrumental vs non-instrumental utility, it would be ideal if they can do it on both behavior and neural activity. That way they can resolve the puzzle because they can find which type of utility is reflected in behavior and then test how the same utility is reflected in brain activity.

Second, they apparently manipulated the time horizon (giving people from 1 to 6 free choices). If so, this nice manipulation could potentially test this, right? Time horizon has been shown by Wilson et al. to regulate exploration vs exploitation. The instrumental value of information gained by exploration should be higher for long time horizons, whereas the non-instrumental value of information (e.g. from satisfying curiosity) has no obvious reason to scale with time horizon.

3. They analyze the first free-choice trial because "information and reward are orthogonalized by the experiment design", but is this true? They say Wilson et al. showed this in their two option task but it is unclear it applies to this three option task. Particularly because they do not analyze activity associated with all decks, but rather in terms of the chosen deck, and participants chose decks based on both info and reward. Depending on a person's choice strategy the info and reward of the chosen deck may not be orthogonal, even though the info and reward considered over all possible decks they could choose are orthogonal.

Basically, it would be valuable to prove their point by including supplemental scatterplots with tests for correlation between whichever information and reward regressors they use in the behavioral model and in the neural model. If they do have some correlation, they should show their analysis handles them properly.

4. I am a little confused by how they compare information value and reward value. In the model from their previous paper and used in this paper they say, roughly speaking, that the value of a deck = (expected reward value from the deck) + w*(expected information value from the deck). So, I thought that they would use the reward and information value from the chosen deck as their regressors. But they subtract the reward values of other decks from the reward term. I am not sure why. They cite studies saying that "VmPFC appears not only to code reward-related signals but to specifically encode the relative reward value of the chosen option", but if so, then if they want to test if vmPFC codes information-related signals shouldn't they also test if it encodes the relative information value? (Or does this not matter because It,j(c==1) is linearly related to It,j(c==1) – mean(It,j(c==2),It,j(c==3)) in this task so the findings would be the same in either case?). And conversely, if dACC encodes information value of a deck shouldn't they test if it encodes the reward value of that deck, instead of relative reward value? It's also confusing that in this paper and their previous paper they argue that

Basically I want to make sure the difference in vmPFC vs. dACC coding is truly due to reward vs information coding rather than relative vs non-relative coding.

*Reviewer #2:*

This is an interesting and well-motivated study, using a clever experimental design and model-based fMRI. The authors argue their results support two independent value systems in the prefrontal cortex, one for reward and one for information. Based on separate GLMs, it is argued for a double dissociation, with relative reward (but not information gain) coded in vmPFC and information gain (but not relative reward) encoded in dACC. While I am sympathetic to this characterization, I have concerns about several of the key analyses conducted, and their interpretation that, I believe, call into question the main conclusions of the study.

1. Correlations between relative reward and information gain terms. The methods state: "Because information and reward are expected to be partially correlated, the intent of GLMs was to allow us to investigate the effects of the second parametric modulator after accounting for variance that can be explained by the first parametric modulator." I had thought the purpose of the task design was to orthogonalize reward and information. What is the correlation between regressors and what is driving the correlation? Are these terms correlated because people often switch to choose options that have high expected reward but have also not been sampled as frequently, in the unequal information trials? If these terms were in fact experimentally dissociated, it should be possible to allow them to compete for variance within the same GLM without any orthogonalization. This is an important issue to address in order to support the central claims in the paper.

2. If dACC is specifically encoding the "information value" then there should be no effect of the relative value on equal information condition trials. By design, these trials should appropriately control for any information sampling difference between options. What is the effect of the negative relative value in the dACC on these equal information trials (or similar terms as defined in my next point)?

3. Relative value definition. In the analyses presented, the negative effect of relative reward in dACC is going in the predicted direction (negatively) and looks marginal. This may result from how the relative reward term was defined. What does the effect in dACC look like if relative reward is instead defined as V2 – V1, V2 – avg(V2,V3), or even entropy(V1,V2,V3)? A similar question applies to the vmPFC analyses. Did the relative value defined as the Vc – avg(V2, V3) explain VMPFC activity better than Vc – V2 or even -entropy(V1,V2,V3)?

4. Double dissociation. The authors are effectively arguing that their data supports a double dissociation between VMPFC encoding relative reward but not information gain and dACC for information gain but not relative reward, but this is not directly tested. This claim can be supported by a statistical interaction between region and variable type. To test this hypothesis appropriately, the authors should therefore perform an F-test on the whole brain or a rmANOVA on the two ROIs, using the β coefficients resulting from a single GLM.

5. Information gain or directed exploration (or both)? Based on the equal and unequal information conditions that were all modeled in the GLMs, it seems likely that the information gain would be high (0 or 2 samples in the unequal information trials) only when subjects choose the more uncertain option, which is equivalent to directed exploration (provided the relative reward is controlled for). Conversely, it would be low when subjects choose the better-known option and are therefore exploiting known information, and intermediate in the equal information condition. Thus, would it be fair to say that the activity in dACC could equally be driven by directed exploration decisions (also equivalent to switching away from a better known "default" option), rather than necessarily "coding for information gain" per se? One way to address this is to show that the information gain is still reflected when subjects choose the more sampled options, or to include an additional explore/exploit regressor in the GLM.

6. Background literature in the introduction. Some of the assertions concerning vmPFC and dACC function in the introduction could be further queried.

For example, the introduction states: "In general, vmPFC activity appears to reflect the relative reward value of immediate, easily-obtained, or certain outcomes, while dACC activity signals delay, difficulty, or uncertainty in realizing prospective outcomes." While I do agree that some evidence supports a distinction between current outcomes in vmPFC and prospective outcomes in dACC, there is considerable evidence that vmPFC only reflects the relative chosen value for trials in which a comparison is more difficult and integration amongst attributes is required. For example, in both fMRI and MEG studies, the vmPFC effect of value difference goes away for "no-brainer" decisions in which both the reward probability and the magnitude are greater (dominant) for one of the options, rendering integration unnecessary and the decision easier – see Hunt et al., Nature Neurosci. 2012, PLoS CB, 2013; Jocham et al., Nature Neurosci., 2014. In addition, vmPFC is typically found to reflect relative subjective value in inter-temporal choice studies (e.g. Kable and Glmicher, Nature Neurosci., 2007; Nicolle et al., Neuron., 2011, etc.). Interestingly, vmPFC does not show relative value effects for physical effort-based decisions, whereas the dACC does (see Prevost et al., J. Neurosci. 2010; Lim et al., Klein-FLugge et al. J. Neurosci., 2016; Harris and Lim, J. Neurosci., 2016). Thus, vmPFC relative value effects are observed for difficult decisions requiring integration and comparison, including for uncertain and delayed outcomes, but potentially not when integrating effort-based action costs. This is also consistent with lesion work in humans and monkeys (Rudebeck et al., J. Neurosci., 2008; Camille et al., J. Neurosci., 2011).

In addition, there is considerable evidence from single unit recording to support the interpretation that ACC neurons do not only encode information gain or information value but integrate multiple attributes into a value (or value-like) effect, even in the absence of any possibility for exploration and, thus, information gain. For example, Kennerley et al., JoCN 2009, Nature Neurosci. 2011 have shown that ACC neurons, in particular, multiplex across multiple variables including reward amount, probability, and effort such that neurons encode each attribute aligned to the same valence (e.g. positive value neurons encode reward amount and probability positively and effort negatively, and negative value neurons do the opposite), consistent with an integrated value effect, even in the absence of any information gain. This literature could be incorporated into the Introduction, or the assumptions toned down some.

7. Experimental details. Some key details appear to be omitted in the manuscript or are only defined in the Methods but are important to understanding the Results. For instance:

– The information gained is defined as follows: "negative value It, j(c) relates to the information to be gained about each deck by participants". However, it is unclear for which choice (deck) this is referring to. I assume it is for the chosen deck on the first free choice trial? Or is it the sum/average over the decks? This should be made clearer.

– It would be helpful it the relative reward and Information gain regressors were defined in the main text.

– What details were subjects told about the true generative distributions for reward payouts.

– What do subjects know about the time horizon, which is important for modulating directed exploration? This could assist in interpreting the dACC effect in particular. Indeed, manipulating the time horizon was a key part of the Wilson et al. original task design and would assist in decoupling the reward and information, if it is manipulated here.

– How are the ROIs defined? If it is based on Figure 2F, the activation called dACC appears to be localized to pre-SMA. This should therefore be re-labelled or a new ROI should be defined.

8. Brain-behavior relationships. If dACC is specifically important for information gain computations, then it would follow that those subjects who show more of a tendency for directed exploration behaviorally would also show a greater neural effect of information gain in ACC and vice versa for vmPFC and relying on relative reward values. Is there any behavioral evidence that can further support the claims about their specific functions?

9. Reporting of all GLM results. It would be helpful to produce a Table showing all results from all GLMs. For example, where else in the brain showed an effect of choice probability, etc.?

*Reviewer #3:*

The manuscript describes an fMRI study that tests whether vmPFC and dACC independently code for relative reward and information gain, respectively, suggesting an independent value system. The paper argues that the two variables have been confounded in previous studies, leading researchers to advocate for a single value system in which vmPFC and dACC serve functionally opposed roles, with vmPFC coding for positive outcome values (e.g. reward) and dACC coding for negative outcome values (e.g. difficulty, costs or uncertainty). The study deploys a forced-choice and a free-choice version of a multi-armed bandit task in which participants select between three decks to maximize reward. It also includes an RL model (gkRL) that is fit to participant's behavior. Variables of the fitted model are used as GLM regressors to identify regions with activity related to relative reward, as well as information gain. The study finds symmetrical activity in vmPFC and dACC (e.g. vmPFC correlates positively, dACC negatively with reward) if correlations between relative reward and information gain are not taken into account. In contrast, the study finds that vmPFC and dACC independently code for relative reward and information gain, respectively, if confounds between reward and information gain are accounted for in the analysis. The manuscript concludes that these findings support an independent value system as opposed to dual value system of PFC.

I found this paper to be well-written and consider it to be an interesting addition to a rich literature on neural correlates of information. However, I got the impression that the manuscript oversells the novelty of its results and fails to situate itself in the existing literature (Comments 1-2). Moreover, some of the key references appear to be misrepresented in favor of the study's conclusions (Comment 3). While most of the methodology appears sound (though see Comment 6 in the statistical comments section), the RL model yields questionable model fits (Comment 4). The latter is troubling given that reported analyses rely on regressors retrieved from the RL model. Finally, the study may be missing a proper comparison to alternative models (Comment 5).

Below, I elaborate on concerns that will hopefully be useful in reworking this paper for a future submission.

1. Novelty of the findings: A main contribution of the paper is the finding that vmPFC and dACC code independently for reward and information gain, respectively, when accounting for a confound between reward and information. However, it should be noted that several other decision-making studies found a similar distinction between vmPFC and dACC by orthogonalizing reward value and uncertainty (e.g. Blair et al. 2006, Daw et al., 2006, Marsh et al. 2007, FitzGerald et al., 2009; Kim et al., 2014). In light of this work, the novelty distinction drawn in this manuscript appears exaggerated. Rather than advocating for functionally opposing roles between dACC and vmPFC, these articles support interpretations similar to the Dual Systems View. While the present manuscript includes a citation of such prior work ("other studies have reported dissociations between dACC and vmPFC during value-based decision-making [citations]", lines 52-53), the true distinction remains nebulous.

2. Relevance of the studied confound: I commend this study for addressing a confound between reward and information in multi-armed bandit tasks of this type. However, the manuscript overstates the generality of this confound, as a motivation for this study. A reader of the manuscript might get the impression that most of the relevant literature is affected by this confound, when in fact the confound only applies to a subset of value-based decision-making studies in which information can be gained through choice. It is unclear how the confound applies to decision-making studies in which values for stimuli cannot be learned through choice (e.g. preference-based decisions). A large proportion of decision-making research fits this description. It is not clear how value would be confounded with information in this domain. Finally, the study focuses on the *relative* value of options, whereas many studies do not (see the references above).

3. Mischaracterization of the literature: In Table 1, and throughout the text, the manuscript lists studies in which vmPFC and dACC are shown to be positively or negatively correlated with decision variables. It is then implied that these studies advocate a "Single Value System", i.e. that regions such as vmPFC and dACC constitute a "single distributed system that performs a cost-benefit analysis" (line 36). Yet, most of the cited work does not advocate for this interpretation. For instance, cited work on foraging theory (e.g. Rushworth et al., 2012) makes the opposite claim, suggesting that vmPFC and dACC are involved in different forms of decision-making. Furthermore, the cited study by Shenhav et al. (2016) neither explicitly promotes the interpretation that vmPFC and dACC show "symmetrically opposed activity", nor does it suggest that they are in "functional opposition" in value-based choice (lines 53-58). Instead, the authors of Shenhav et al. (2016) suggest that their findings support the view that vmPFC and dACC are associated with distinct roles in decision-making, such as vmPFC being associated with reward, and dACC being associated with uncertainty or conflict. Another cited study from Shenhav et al. (2019) also does not advocate for "emerging views regarding the symmetrically opposing roles of dACC and vmPFC in value-based choice" (line 336). Instead, the cited article draws a strong distinction between (attentionally-insensitive) reward encoding in vmPFC vs. (attention-dependent) conflict coding in dACC.

4. Model fits: The study's main fMRI results (e.g. that dACC independently codes for information gain) are based on regressors extracted from variables of the fitted RL model (e.g. information gain). Yet, more than a third of the participants yields model fits that could lead one to question the validity of the GLM regressors, in particular the regressor for information gain. One of the fitted model parameters, labeled "γ", exponentiates the history of previous observations to compute the amount of information associated with a deck. In line 420 it is written that "[γ] is constrained to be > 0". However, γ appears to be 0 for three out of twenty participants (6, 11, 14, see Table S2). If I understand the model correctly, then information associated with any deck (line 416) would evaluate to 1 for those participants, irrespective of which decks they observed. That is, the model fit suggests that these participants show no sensitivity to information across decks. A second parameter, omega, appears to scale how much information gain weighs into the final value for a deck (line 425). Again, if I understand correctly, the lower omega the less the RL agent weighs information gain when computing the final value associated with a deck. Thus, if omega evaluates close to 0, the agent would barely consider information gain when choosing a deck based on its final value. Given this, I am surprised that omega varies vastly in magnitude across participants, ranging from 0.26 to 48.66. Four out of twenty participants (1, 3, 19, 20, see Table S2) have fitted values for omega smaller than 1. This makes one question whether these participants actually use information to guide their decision making, relative to other participants. Relatedly, I was a bit surprised that the GLM implements – I_{t,j}(c) as regressor for information gain, instead of – omega * I_{t,j}(c). If omega is small, then the former regressor may not be a good proxy for information gain that is actually used to guide decision-making. Altogether, these model fits (seven out of twenty participants) may undermine the validity of "information associated with a deck" when used as a regressor for information gain in the fMRI analysis (at least in GLMs 2, 3 and 4).

5. Model comparison: In light of the previous comment, I wonder whether the gkRL model is best suited to explain behavior in this task, and, thus, whether it serves as a proper foundation for the intended fMRI analysis. As the manuscript notes in lines 408-409, the "model was already validated for this task and it was better able to explain participants' behavior compared to other RL models". However, this evaluation was conducted in a different study with (presumably) a different sample of participants. The manuscript should nevertheless provide a convincing argument for deploying this model to this study, as well as for using corresponding regressors in the fMRI analysis. This could be achieved by demonstrating that the model does a better job in explaining behavior compared to a standard RL model (without the information term) ‘for each of these participants’.

[Editors’ note: further revisions were suggested prior to acceptance, as described below.]

Thank you for resubmitting your work entitled "Independent and Interacting Value Systems for Reward and Information in the Human Brain" for further consideration by *eLife*. Your revised article has been evaluated by Michael Frank (Senior Editor) and a Reviewing Editor and the three reviewers who previously reviewed your paper.

The reviewers felt that the revised manuscript has been improved. Indeed, in some cases, the reviewers felt that revisions were extensive and commendable; for example, in addressing whether the information value is instrumental or non-instrumental.

However, there were several other fundamental points that were missed in the revision. The reviewers and reviewing editor discussed these points at length, and it was decided that most could be addressed in some form but simply were not. These include issues related to the placement of these results in the broader literature and characterization of previous studies, the definition of relative reward, the incorporation of the omega parameter in the information term, matching of covariates in the GLM, and a number of gaps in the added analyses, such as the switching versus default behavior. It was noted that some other issues like the orthogonalization approach or the confounding of the information term with other variables were also not addressed, and there was some discussion of whether these are addressable. Nonetheless, these outstanding issues were all viewed as important.

Thus, the consensus was to send this back for another revision. However, we must emphasize that if these points are not addressed in revision, the manuscript may not be considered acceptable for publication at *eLife*. I have included the full and detailed reviews from each reviewer in this letter so you can address them directly in your revision.

*Reviewer #1:*

The revised manuscript addresses some concerns raised by the reviews, by adding separate analyses that examine the effect of absolute versus relative reward and information gain, an analysis of parameter recovery and a formal comparison to a standard RL model. However, the revised manuscript still misses to address a number of critical issues:

1. Novelty: The revised manuscript makes an attempt of distinguishing its claims from prior work. However, the reader may still be confused about how the authors situate their findings in contrast to prior work. The manuscript seeks to promote an "independent value system for information in human PFC", in contrast to single value system view that "vmPFC positively and dACC are negatively contributing to the net-value computation". The authors state that "[e]ven for studies in which dACC and vmPFC activity is dissociated activity in vmPFC is generally linked to reward value, while activity in dACC is often interpreted as indexing negative or non-rewarding attributes of a choice (including ambiguity, difficulty, negative reward value, cost and effort)". This statement requires further elaboration. Why can a non-rewarding attribute not be considered independent of reward, thus promoting a dual-systems view? That is, the reader might wonder: are the non-rewarding attributes of choice in the cited studies necessarily in opposition to reward, contributing to the same "net-value computation" and suggesting a symmetrical opposition? For some of these attributes, this is not necessarily the case (see Comment 2 for a specific example).

2. Mischaracterization of literature: The revised manuscript continues to mischaracterize the existing literature, suggesting that the studies of Shenhav et al. interpret ACC and vmPFC as participating in a single reward system-although the revision appears to partly avoid this characterization by dropping the Shenhav et al. (2018) reference. Regardless, there are several reasons for why one should not interpret these studies as advocating for and/or interpreting their findings as supportive of a single-value system:

– As the authors of this manuscript state in their response, the Expected Value of Control (EVC; 2013) framework suggests that certain forms of conflict or uncertainty may be considered when computing EVC. However, the 2016 and 2018 studies do consider forms of conflict and uncertainty that are not explicitly taken into account when computing the expected value of a control signal. For instance, in the 2018 study, EVC is expressed a function of reward, not uncertainty or conflict (Equation 3):

EV_t = [Pr(Mcorrect) * Reward(MCorrect)] + [Pr(Ccorrect) * Reward(CCorrect)]

– Irrespective of the way in which the EVC is computed, EVC theory neither suggests that dACC activity is reflective of a single value, nor does it explicitly propose that dACC activity correlates with EVC. Shenhav et al. (2013, 2016) communicate this in their work.

– It is worth noting that Shenhav et al. (2014, 2018b, 2019; see references below) make the case that dACC vs. vmPFC are engaged in different kinds of valuation (with dACC being more tied to uncertainty and vmPFC being more tied to reward). The authors may want to consider this work when situating their study in the existing literature (see Comment 1):

Shenhav, A. and Buckner, R.L. (2014). Neural correlates of dueling affective reactions to win-win choices. Proceedings of the National Academy of Sciences 111(30): 10978-10983

Shenhav, A., Dean Wolf, C.K., and Karmarkar, U.R. (2018). The evil of banality: When choosing between the mundane feels like choosing between the worst. Journal of Experimental Psychology: General 147(12): 1892-1904.

Shenhav, A. and Karmarkar, U.R. (2019). Dissociable components of the reward circuit are involved in appraisal versus choice. Scientific Reports 9(1958): 1-12.

– Finally, the statements made in Shenhav et al. do simply not express a single-value system view, as erroneously suggested by the authors of this manuscript. For convenience of the authors, reviewers and editors, I attached excerpts from the Discussion sections of the relevant 2016 and 2018 articles below.

That is, the manuscript continues to mischaracterize these studies when stating that they interpret ACC and vmPFC as a single value reward system. I respectfully ask the authors to consider these points in their current and future work.

3. Model fits: I commend the authors for including an analysis of parameter recoverability as well as a model comparison with a standard RL model. However, the model comparison should indicate which trials were included in the analysis (all first free choices, as used for parameter estimation?). As indicated in my previous review, the analysis should also report the outcome of this model comparison, separately for each participant (i.e. a table with BIC values for gkRL and standardRL, for each participant), so that the reader can evaluate whether gkRL is generally better (not just on average).

4. Subjective information gain: I apologize if I missed this, but the authors' response doesn't seem to address the following issue raised earlier: The analysis refrains from taking into account the *subjective* information gain that is actually factored into the model's choice. As re-iterated by the editor, I was a bit surprised that the GLM implements – I_{t,j}(c) as regressor for information gain, instead of – omega * I_{t,j}(c). If omega is small, then the former regressor may not be a good proxy for information gain that is actually used to guide decision-making. This issue was implicitly raised in comment 8 of Reviewer 2, to which the authors replied that "all the neural activity relates to subjects' behavior". The claimed brain-behavior relationship would be more internally valid if the participant-specific parameter omega is considered in the analysis.

5. Consideration of Covariates: The authors state that "Standard Deviation and Chosen-Second were highly correlated with RelReward" and "[f]or this reason, these covariates were not considered any further". Does this mean that these covariates are confounded with relative reward, and that the authors cannot dissociate whether vmPFC encodes Standard Deviation (as the author's proxy for decision entropy) or Chosen-Second? Also, why aren't all covariates included when examining the effect of information gain (both relative and absolute) on both vmPFC and dACC activity, as well as the effect of expected reward? It would seem appropriate to include at least non-correlating covariates in all analyses to allow for a proper comparison across GLMs. Furthermore, it would seem reasonable to include RT as a covariate despite it not being correlated with information gain. First, information gain is not the only factor being considered. Second, a lack of statistical correlation with information gain doesn't imply that RT cannot explain additional variance when included in the GLM.

Excerpts of Discussion from

Shenhav, A., Straccia, M. A., Botvinick, M. M., and Cohen, J. D. (2016). Dorsal anterior cingulate and ventromedial prefrontal cortex have inverse roles in both foraging and economic choice. Cognitive, Affective, and Behavioral Neuroscience, 16(6), 1127-1139:

"Although these findings collectively paint a picture inconsistent with a foraging account of dACC, it is important to note that the difficulty-related activations we observed lend themselves to a number of possible interpretations. These include accounts of dACC as monitoring for cognitive demands such as conflict/uncertainty (Botvinick, Braver, Barch, Carter, and Cohen, 2001; Cavanagh and Frank, 2014), error likelihood (Brown and Braver, 2005), and deviations from predicted response-outcome associations (Alexander and Brown, 2011); indicating the aversiveness of exerting the associated cognitive effort (Botvinick, 2007); explicitly comparing between candidate actions (e.g., Hare, Schultz, Camerer, O'Doherty, and Rangel, 2011); and regulating online control processes (Dosenbach et al., 2006; Posner, Petersen, Fox, and Raichle, 1988; Power and Petersen, 2013). In line with a number of these accounts, we recently proposed that dACC integrates control-relevant values (including factors such as reward, conflict, and error likelihood) in order to make adjustments to candidate control signals (Shenhav, Botvinick, and Cohen, 2013). In this setting, one of multiple potentially relevant control signals is the decision threshold for the current and future trials, adjustments of which have been found to be triggered by current trial conflict (i.e., difficulty) and mediated by dACC and surrounding regions (Cavanagh and Frank, 2014; Cavanagh et al., 2011; Danielmeier, Eichele, Forstmann, Tittgemeyer, and Ullsperger, 2011; Frank et al., 2015; Kerns et al., 2004). Our findings also do not rule out the possibility that dACC activity will in other instances track the likelihood of switching rather than sticking with one's current strategy – as has been observed in numerous studies of default override (see Shenhav et al., 2013) – over and above signals related to choice difficulty. However, our results do suggest that a more parsimonious interpretation of such findings would first focus on the demands or aversiveness of exerting control to override a bias, rather than on the reward value of the state being switched to.

Another recent study has questioned the necessity of dACC for foraging valuation by showing that this region does not track an analogous value signal in a delay of gratification paradigm involving recurring stay/switch decisions (McGuire and Kable, 2015). Instead, this study found that vmPFC played the most prominent evaluative role for these decisions (tracking the value of persisting toward the delayed reward). The authors concluded that vmPFC may therefore mediate evaluations in both foraging and traditional economic choice. Our study tested this assumption directly within the same foraging task that was previously used to suggest otherwise. We confirmed that vmPFC tracked relative chosen value similarly in both task stages. As is the case for our (inverse) findings in dACC, there are a number of possible explanations for this correlate of vmPFC activity, over and above the salient possibility that these activations reflect the output of a choice comparison process (Boorman et al., 2009; Boorman et al., 2013; Hunt et al., 2012). First, it may be that vmPFC activity is in fact tracking ease of choice or some utility associated therewith (cf. Boorman et al., 2009), such as the reward value associated with increased cognitive fluency (Winkielman, Schwarz, Fazendeiro, and Reber, 2003). Similarly, it may be the case that this region is tracking confidence in one's decision (possibly in conjunction with value-based comparison), as has been reported previously (De Martino, Fleming, Garrett, and Dolan, 2013; Lebreton, Abitbol, Daunizeau, and Pessiglione, 2015). A final possibility is that this region is not tracking ease or confidence per se, but a subtle byproduct thereof: decreased time spent on task. Specifically, it is possible that a greater proportion of the imposed delay period in this study and in KBMR was filled with task-unrelated thought when participants engaged with an easier choice, leading to greater representation of regions of the so-called "task-negative" or default mode network (Buckner, Andrews-Hanna, and Schacter, 2008), including vmPFC. While this is difficult to rule out in the current study, our finding of similar patterns of vmPFC activity in our previous experiment (which omitted an imposed delay) counts against this hypothesis. These possibilities notwithstanding, our results at least affirm McGuire and Kable's conclusion that the vmPFC's role during foraging choices does not differ fundamentally from its role during traditional economic choices."

Excerpts of Discussion from

Shenhav, A., Straccia, M. A., Musslick, S., Cohen, J. D., and Botvinick, M. M. (2018). Dissociable neural mechanisms track evidence accumulation for selection of attention versus action. Nature communications, 9:

"Our findings within dACC are consistent with previous proposals that this region signals demands for cognitive control (e.g., conflict, error likelihood [34,56,57]) and that these demands may be differentially encoded across different populations within dACC [44,45]. Most notably, our findings are broadly consistent with the recent proposal that dACC signals such demands in a hierarchical manner [29,32,40] (cf. Refs. [14,58]). Specifically, it has been suggested that dACC contains a topographic representation of potential control demands, with more caudal regions reflecting demands at the level of individual motor responses and more rostral regions reflecting demands at increasing levels of abstraction (e.g., at the level of effector-agnostic response options). According to this framework, it is reasonable to assume that this rostrocaudal axis might encode uncertainty regarding which attribute to attend more rostrally than uncertainty regarding which response to select. Under the added assumption that our participants were heavily biased towards attending the high-reward attribute and became increasingly likely to attend the low-reward attribute as its coherence increased (cf. Figure 2) – potentially narrowing their relative likelihood of attending either stimulus and thereby increasing uncertainty over which attribute to attend – our findings could be interpreted as further evidence for such an axis of uncertainty. However, such an interpretation remains speculative in the absence of additional measures of attentional allocation (e.g., eyetracking within a task that uses spatially segregated attributes). Our findings may also be consistent with a more recent proposal that a similar axis within dACC tracks the likelihood of responses and outcomes (e.g., error likelihood) at similarly increasing levels of abstraction [33]. Collectively these accounts of the current findings are consistent with our theory that regions of dACC integrate information regarding the costs and benefits of control allocation (including traditional signals of control demand) in order to adaptively adjust control allocation28,34.

The dACC signals we observed are also consistent with evaluation processes unrelated to control per se, indicating for instance the costs of maintaining the current course of action in caudal dACC and the value of pursuing an alternate course of action (cf. foraging) in rostral dACC [18,19]. The connection between rostral dACC activity and choices to follow evidence for the low-reward attribute can be seen as further support for such an account (though this could similarly reflect adjustments of attentional allocation). Our current study is limited in adjudicating between these two accounts because increasing evidence in support of an alternative attentional target in our task (i.e., increased coherence of the low-reward attribute) necessarily leads to greater uncertainty regarding whether to continue to focus on the high-reward attribute. However, given that evidence for foraging-specific value signals in dACC remains inconsistent [34,37,59], an interpretation of our findings that appeals to cognitive costs or demands may be more parsimonious. That said, future studies are required to substantiate the current interpretation by demonstrating that the dACC's response to the would-be tempting alternative (the high-coherence low-reward attribute) decreases when the relative coherence and reward of the alternate attribute are such that the decision to switch one's target of attention is easy (cf. Ref. [31]).

In contrast to dACC, where activity tracked how little evidence was available to support the chosen response (i.e., to discriminate between the correct and incorrect response), vmPFC instead tracked the evidence in favor of the chosen response, in a manner proportional to the reward expected for information about each attribute. This finding is broadly consistent with previous findings in the value-based decision making literature, where vmPFC is often associated with the value of the chosen option and/or its relationship to the value of the unchosen option [60,61]. The fact that vmPFC's weights on these attributes were not proportional to the weight each attribute was given in the final decision suggests that vmPFC may have played less of a role in determining how this information was used to guide a response, than in providing an overall estimate of expected reward. In addition to any incidental influence it may have on the perceptual decision on a given trial, this reward estimate could provide a learning signal about the task context more generally (e.g., overall reward rate [24,48,49] or confidence in one's performance [50,51]), consistent with our observation that this region encodes elements of reward expected from a previous trial. While our findings are suggestive, the degree to which vmPFC guides and/or is guided by decisions regarding what to attend deserves further examination within studies that measure attention allocation while systematically varying reward as well as the degree of control one has over one's outcomes (versus, for instance, being instructed what to attend). It will also be worth directly contrasting vmPFC correlates of attribute evidence when attention is guided by reward (as in the current study) versus instruction (e.g., Ref. [3])."

*Reviewer #2:*

The authors propose that there are "dedicated" and "independent" reward value and information value systems driving choice in the medial frontal cortex, respectively situated in vmPFC and dACC. This is a major claim that would be highly significant in understanding the neural basis of decision making, if true, and therefore commendable. In their revision, the authors clarified several important points. However, I am afraid that their response to my and the other reviewers' previous concerns has not convinced me that their data clearly support this claim. While some of my previous comments were addressed, others concerns were mostly unaddressed.

The authors have designed a task that builds on a similar task from Wilson et al. (2014) that set out to experimentally dissociate reward difference and information difference between choice options. However, rather than use those terms for the fMRI analysis and/or focus separately on conditions in which these terms are experimentally dissociable (such as considering equal and unequal information trials separately), the way in which the relative reward and information terms are defined for the fMRI analysis introduces a significant confound between the relative reward and information gain terms (as the authors themselves point out in the manuscript and Figure 1). Apparently, this is done because the design does may not be sufficiently well powered to focus on each condition separately. The approach adopted to try to deal with this confound is instead to estimate separate GLMs (3 and 4) and orthogonalise one term with respect to the other in reverse order. This approach does not appear to "control for the confound", but instead assigns all shared variance to the un-orthogonalised term. That is, the technique does not affect the results of the orthogonalized regressor but instead assigns any shared variance to the unorthogonalised regressor (see e.g. Andrade et al., 1999). Thus, in their analysis of relative reward, the relative reward regressor is given explanatory power that might derive from information value, and in the analysis for information value, the reverse is true. Notwithstanding the interesting results from their simulations, this approach makes the interpretation of the β coefficients and the claim of independence challenging, particularly if one of the scenarios included in the simulations (independent reward value and information systems, or single value system) does not accurately capture the true underlying processes in the brain. More generally, because the two terms are highly correlated, the model-based approach across all trials requires that their computational model can precisely identify each term of interest accurately and independently. This is the same problem noted in other previous studies of exploration/exploitation that experimentally dissociating relative reward and relative information was supposed to address (at least as I understood it).

Relative Reward Definition

As I stated in my previous review, the precise definition of relative reward is important for determining what computation is driving the signals measured in vmPFC and dACC. Consideration of alternative terms was appropriately applied to analysis of vmPFC (following reviewer 1 and my suggestions), but not to analysis of dACC activity. Specifically, my suggestions to compare different relative value terms, specifically V2-E(V2,V3) and entropy(V1,V2,V3) (definitions that are based on dACC effects in past studies) for dACC activity were not performed (or reported). I suspect one of these would show an effect in dACC, but it's not possible to tell because they did not address the question or show the results. Importantly, as I pointed out previously, there is evidence in their Figure 4E for a marginal? negative effect of the relative reward term they did test in the dACC ROI. This suggests that getting the term's formulation correct could easily result in a significant negative effect. If there were an effect of either of these alternative terms in dACC, it would go against one of the central claims of the paper. In the authors' reply, rather than test these alternatives, they seemed to simply say they did not know what my comment had meant.

Information Value Confounds

In their task design the information term (I(c)) is directly proportional to the number of times an item is chosen during forced choice trials. Therefore, this term is highly correlated with the novelty/familiarity of that option. The authors clarified that they did not cue to subject about the horizon (as done in a previous task by Wilson et al. 2014). The authors state cueing the horizon did not matter in prior behavioral studies of their task (which is somewhat surprising if subjects are engaging in directed exploration), but in any case this means that the number of samples is the only variable determining the "information value". Had the horizon been cued, then one could at least test whether dACC activity increased with the planning horizon, since there would be an opportunity to benefit from exploration when this was greater than 1, and this manipulation would be useful because it would mean I(c) was not solely determined by the number of times an option had been chosen in forced choice trials and therefore highly correlated with familiarity/novelty of a deck.

Importantly, the I(c) term also appears to be confounded by the choice of the subject. As I noted in my previous review comment, because the I(c) term is only modeled at the first free choice, its inverse (which is what is claimed to be reflected in dACC activity) will be highest whenever the subject explores. Therefore, the activity in dACC could instead be related to greater average activity on exploratory compared to exploitative choices, as has been suggested previously, without any encoding of "information value" per se. For this reason I suggested including the choice (exploratory vs exploitative or switching away from the most selected "default" option) as a covariate of no interest to account for this possibility. This comment was not directly addressed. Instead, the authors make a somewhat convoluted argument that switching away from a default in their task corresponds to not exploring, since explore choices were the more frequent free choices in their task (though this would not be the case for the current game, where the forced choices would likely determine the current "default" option). If the dACC is in fact encoding the "information value" then this effect should remain despite inclusion of the explore/exploit choice, meaning it should be present on other trials when they do not choose to explore. This concern was not addressed, although it is unclear if their experimental design provides sufficient independent variance to test this on exploit trials.

As I pointed out in my original review, if the dACC shows no effect of relative value (or the alternative terms described above) as claimed, then there should be no effect of the relative value on the "equal information" trials. The authors do report no significant effects at a p<0.001 whole-brain threshold on these trials, which they acknowledge may be underpowered. It would be more convincing to show this analysis in the dACC ROI that is used for other analyses. A significant effect on these trials that experimentally dissociate relative reward from information would also run counter to one of the main claims in the paper.

For the above reasons, I do not believe my previous concerns were addressed and that the data support some of the central claims of the manuscript.

References:

Andrade A, Paradis AL, Rouquette S, Poline JB (1999) Ambiguous results in functional neuroimaging data analysis due to covariate correlation. Neuroimage 10:483-486.

*Reviewer #3:*

This is a greatly improved manuscript. The authors have done substantial work thoroughly revising their manuscript including carefully addressing almost all of my original comments. These have made the paper considerably more methodologically solid and clarified their presentation, but have even allowed them to uncover additional results that strengthen the paper, such as providing evidence that ACC activity related to information is especially related to the non-instrumental value of information.

I do have a few questions and concerns for the authors to address regarding the methodological details of some of their newly added analyses (comments 1-4). I also have some suggestions to improve their presentation of the results to help readers better understand them (comments 5-6 and most minor comments).

1. The authors have included a new analysis that nicely addresses my comments about the need to distinguish between neural signals encoding Instrumental vs Non-Instrumental Information. I greatly appreciate them developing the quite painstaking and elaborate computations necessary to estimate the instrumental value of information in this task! Furthermore, this led them to report a very interesting result that supports their big picture message, that ACC activity is most related to their estimate of Information Gain from their gkRL model of behavior, while vmPFC is most related to their estimate of Instrumental Information. This is very striking, and if true, is an important contribution to our understanding of how neural systems value information.

I have a few comments about this:

- Overall, the whole approach is explained very clearly. It appears methodologically solid, strongly motivated, and well-founded in Bayesian decision theory.

There is one missing step that needs to be explained, though: the final step for how they use the Bayes-optimal long term values for each option (which I will call "Bayes Instrumental Value") to compute the Instrumental Information which they used as a regressor. I don't quite understand how they did this. Reading the methods, it sounds almost as if they set Instrumental Information = Bayes Instrumental Value. If so, I don't think that would be correct, since then a large component of "Instrumental Information" would actually just be the ordinary expected gain of reward from choosing an option (e.g. based on prior beliefs about option reward distributions and on the mean experienced rewards from that option during the current game), not specifically the value of the information obtained from that option (i.e. the improvement in future rewards due to gaining knowledge to make better decisions on future trials). I worry that the authors might have done this, since their analysis suggests that Instrumental Information may be positively correlated with experienced rewards (e.g. it was correlated with similar activations in vmPFC as RelReward). It is also odd that "The percentage of trials in which most informative choices had positive Instrumental Information was ~ 22%" since surely the instrumental value of information in this task should almost always be at least somewhat positive and should only be zero if participants know they are at the end of the time horizon, which should never occur on first free choice trials.

In general, a simple way to compute the instrumental value of information (VOI) in decision problems is to take the difference between the total reward value of choosing an option including the benefit of its information (i.e. the Bayes Instrumental Value that the authors have already computed) minus the reward value the option would have had if it provided the decision maker with the same reward but did not provide any information (which I the authors may be able to calculate by re-running their Bayesian procedure with a small modification, such as constraining the model to not update its within-game belief distributions based on the outcome of the first free choice trial, thus effectively ignoring the information from the first free-choice trial while still receiving its reward). Thus, we would have a simple decomposition of value like this:

Bayes Instrumental Value = Reward Value Without Information + Instrumental Value of Information

- Relatedly, I was surprised by their fit to behavior that predicts choices based on Instrumental Information and Information Gain. Given the strong effects of expected reward on choice (Figure 2D), I naively would have expected them to include an additional term for reward value, in order to decompose behavior into three parts, "Reward Value Without Information" representing simple reward seeking independent of information, "Instrumental Value of Information" representing the component of information seeking driven by the instrumental demands of the task, and "Information Gain" representing the variable the authors believe corresponds to the non-instrumental value of information (or total value of information) reflecting the subjective preferences of individuals. The way I suggested above to compute Instrumental Information would allow this kind of decomposition.

- I suggest adding a caveat in the methods or discussion to mention that the Bayesian learner may not reflect each individual's subjective estimate of the instrumental value of information, since the task was designed to have some ambiguity allowing the participants to have different beliefs about the task, by not fully explaining the task's generative distribution.

2. The difference between the confounded analysis (Figure 3C) and non-confounded analysis (Figure 4E) is very striking and clear for vmPFC but is not quite as clear for ACC. It looks like the non-confounded analysis of ACC produces slightly less positive β for information and slightly less negative β for reward (from about -0.5 to about -0.35, though I cannot compare this data precisely because these plots do not have ticks on the y-axis). While is true the effect went from significant to non-significant, I am not sure how much this change supports the black and white interpretation in this paper that ACC BOLD signal reflects information completely independently of reward. It might support a less black and white (but still scientifically important and valuable) distinction that ACC primarily reflects information but may still weakly reflect reward. I understand it may not be possible to 'prove a negative' here, but it would be nice if the authors could show stronger evidence about this in ACC, e.g. by breaking down the 3-way ANOVA to ask whether removing the confound had significant effects in each area individually (vmPFC and ACC). Otherwise, it might be better to state the conclusions about ACC and information vs. reward in a less strictly dichotomous/independent way.

3. The analysis of switching strategy is a valuable control, and I agree with the authors on their interesting point that in this task the default strategy would appear to be information seeking rather than exploitation. Can the authors report the results for ACC? The paper seems to describe this section as providing evidence about what ACC encodes, so I was expecting them to report no significant effect of Switch-Default in ACC, but I only see results reported for frontopolar cortex.

4. The analysis of choice difficulty is also a very welcome control. My one suggestion for the authors to consider is also correlating chosen Information Gain with a direct index of choice difficulty according to the gkRL model of behavior that they fit to each participant (e.g. setting choice difficulty to be the difference between V of the best deck with V of the next-best deck). This would be more convincing to choice difficulty enthusiasts, because reaction times can be influenced by many factors in addition to choice difficulty, and previous papers arguing for dACC role in choice difficulty (e.g. Shenhav et al., Cogn Affect Behav Neurosci 2016) define choice difficulty in terms of model-derived value differences (or log odds of choosing an option). The gkRL model is put forward as the best model of behavior in this task in the paper, so it would presumably give a more direct estimate of choice difficulty than reaction time.

5. One of the main messages of the paper is that correlations between reward and information may cause ACC and vmPFC to appear to have opposing activity under univariate analysis, even if ACC and vmPFC actually have independent activity under multivariate analysis.

On this point, the authors nicely addressed my first comment by greatly improving and clarifying Figure 1 showing their simulation results about this confound. The authors also addressed my third comment, which was also on this topic.

However, part of the way they explain this in the text could do with some re-phrasing, since in its current form it could be confusing for readers. Specifically, they make some statements as if information and reward were uncorrelated ("Both Wilson et al. and our task orthogonalize reward and information by adding different task conditions." and "the use of the forced-choice task allows to orthogonalize available information and reward delivered to participants in the first free choice trial") but make other statements as if they were correlated ("However, in order to better estimate the neural activity over the overall performance we adopted a trial-by-trial model-based fMRI analyses. Since reward value and information value are correlated due to subjects' choices during the experiment, this introduces an information-reward confound in our analysis.").

After puzzling over this and carefully re-reading the Wilson et al. paper I understand what they are saying. Here is what I think they want to say. On the first free-choice trial, if you consider all available options, an option's information and that option's reward are uncorrelated. However, if you only consider the chosen option (as most of their analyses do) that option's information and reward are negatively correlated.

If this is the case, the authors need to clarify their explanations, especially specifying when "information" and "reward" refer to all options or only to the chosen option. For example, instead of "orthogonalize available information and reward delivered to participants in the first free choice trial" I think it would be more clear and accurate to say "orthogonalize experienced reward value and experienced information for all available options on the first free-choice trial".

Relatedly, in my opinion this whole issue would be clearer if the authors took my earlier suggestion of simply showing a scatterplot of the two variables they claim are causing the problem due to their correlation (presumably "chosen RelReward" vs "chosen Information Gain"), for instance in Figure 1 or in a supplementary figure. This would be even more powerful if they put it next to a plot of the analogous variables from considering all options showing a lack of correlation (e.g. "all options RelReward" vs "all options Information Gain") thus demonstrating that the problematic correlation is induced by focusing the analysis on the chosen option. Then readers can clearly see what two variables they are talking about, how they are correlated, and why the correlation emerges. It seems critical to show this correlation prominently to the readers, since the authors are holding up this correlation, and the need to control for it, as a central message of the paper.

6. The paper has a very important discussion about how broadly applicable their approach to non-confounded analysis of decision variables is to the neuroscience literature. There is one part that seems especially important to me but may be hard for readers to understand from the current explanation:

"Furthermore, we acknowledge that this confound may not explicitly emerge in every decision-making studies such as preference-based choice. However, the confound in those decision types is reflected in subjects' previous experiences (e.g., the expression of a preference for one type of food over another are consistent, and therefore the subject reliably selects that food type over others on a regular basis). The control of this confound is even more tricky as it is "baked in" by prior experiences rather than learned over the course of an experimental session. Additionally, our results suggest that other decision dimensions involved in most decision-making tasks (e.g., effort and motivation, cost, affective valence, or social interaction) may also be confounded in the same manner."

Here is what I think the authors are saying: In their task, people made decisions based on both reward and other variables (i.e. information), which induced a correlation between the chosen option's reward value and its other variables. If the authors had analyzed neural activity related to the chosen option only in terms of reward value, they would have thought that ACC and vmPFC had negatively related activity to each other, and had opposite relationships to reward value. In effect, any brain area that purely encoded any one of the chosen option's variables (e.g. information) would appear to have activity somewhat related to all of the other decision-relevant variables. Thankfully, the authors were able to avoid this problem. However, crucially, they were only able to avoid this problem because their task was designed to manipulate the information variable. This let them model its effects on behavior, estimate the information gain on each trial, and do multivariate analysis to disentangle it from reward. Therefore, the same type of confound could be lurking in other experiments that analyze neural activity related to chosen options. This potential confound may be especially hazardous in tasks that are complex or that attempt to be ecologically-valid, since they may have less knowledge or control of the precise variables that participants are using to make their decisions, and hence less ability to uncover those hidden decision-relevant variables and disentangle them from reward value.

If this is what they are saying, then I agree that this is a very important point. I suggest giving a more in-depth explanation to make this clear. Also, I think it would be good to state that this confound only applies to analyses of options that have correlated decision-relevant variables (e.g. due to the task design, or due to the analysis including options based on the participant's choice, such as analysis of the chosen option or the unchosen option).

[Editors’ note: further revisions were suggested prior to acceptance, as described below.]

Thank you for resubmitting your work entitled "Independent and Interacting Value Systems for Reward and Information in the Human Brain" for further consideration by *eLife*. Your revised article has been evaluated by Michael Frank (Senior Editor) and a Reviewing Editor and three reviewers.

The revised manuscript has been greatly improved and after some discussion the reviewers agreed that the concerns had been addressed in a responsive revision. However, two points arose in the revision that, after consultation, the reviewers still felt should be addressed, at the very least with clarification and clear discussion of their strengths and weaknesses in the text. These were:

a) Reviewers were confused about why instrumental information was negative on the majority of trials, given that a Bayesian optimal learner was used. The concern is elaborated by a reviewer, as follows:

"I am still puzzled why they say the most informative deck only has positive Instrumental Information on 20% of trials and often has negative Instrumental Information.

The rebuttal is completely correct to point out that instrumental information can be negative. However, this happens when the decision maker is suboptimal at using information so it performs better without some information (e.g. the paper they cite uses examples like "not knowing whether a client is guilty could improve a solicitor's performance"). In this paper the formula is supposed to be using a Bayes optimal learner, that knows the structure of their task in more detail than the actual participants. A Bayes optimal learner should make optimal use of information, and should never be hurt by information, so instrumental information should always be positive or at least non-negative, right?

The rebuttal says that instrumental information can be negative in a specific condition in this task with unequal reward, unequal information, and the most informative option has lower reward value than the other options. However, I do not see why this should be the case. Providing information should never hurt a Bayesian with an accurate model of the task. Also, the specific condition should only be a fraction of trials, it is surprising if something occurring on those trials could account for the fact that 80% of the trials, they report have zero or negative instrumental information. Finally, it is strange that instrumental information would be negative on those trials because they are ones where one deck has not been sampled and has low expected reward, so it seems to me that there is high instrumental value in learning from information that it has a low value. This information is needed for the participant to learn that the deck has a low value and should be avoided going forward, otherwise if the participant did not learn anything from this information, they would be likely to choose it on their second choice. All of this makes me suspect a bug in the code."

This issue is important to resolve. You might just consider working through the computation in a simple example the rebuttal letter.

b) The omega*I versus I distinction was still confusing to reviewers. This concern is detailed by the reviewer, as follows:

"my concern stems from part 3 of the rebuttal, the incorporation of the omega parameter in the information term. The authors report that if they follow our advice to replace -I with -I*omega then their main result in ACC is not significant anymore (at the second level of analysis used to test if activity across participants is significantly affected by information value). Roughly, as I interpret it, -I controls the functional form of how information value depends on samples, and omega controls its magnitude, so both are needed to model information value and its effect on choice. And my analysis suggests that unfortunately -I is not a reliable predictor across individuals of either -I*omega or the effect of information on choice. […] if -I*omega is a more accurate way to quantify information value, does this mean that their current results using -I are wrong or fail to support their major claims?"

After discussion, the reviewers were satisfied that I can be a reasonable regressor but felt it important to clarify in the paper that this regressor reflects the functional form of how information value depends on the number of samples, but not the absolute magnitude of info value (e.g. its monetary worth) or its effect on the probability of choosing the most informative option. Further, it would be helpful to justify why it would be reasonable to expect ACC BOLD signals to be related to this type of relative/scaled value of info rather than its absolute monetary value. So, the distinction between these regressors should be acknowledged, along with the null result for the omega*I regressor.

c) Beyond these two points, the reviewers had suggestions for some additional clarifications.

- If the reviewers' understanding is correct, that the current orthogonalization procedure controls for shared variance is really only applicable when compared to GLMs that only model reward or information, or model them in separate GLMs, rather than the many that model them in the same GLM. This could be clarified in the text.

- For the sake of completeness and transparency, it would be helpful to include some of the findings of control analyses, including (1) whole-brain maps for some of the control GLMs tested, even if they go in the supplement or online; and (2) for ROI analyses of vmPFC and dACC, the effect sizes, t-statistics, and p-values of the other independent variables included in the respective GLMs. As examples, it would be informative to present the -SD, switch – stay, and explore – exploit effects in dACC and vmPFC (pg 12) and the negative effect of GLM 8 (choice probability). Currently only the relative reward and information gain terms from these GLMs are reported for these ROI analyses. Finally, I recommend reporting the analysis of relative reward on equal information trials in the dACC ROI.

- It should be clarified that information gain is confounded with decisions to explore in this task design and the accompanying analysis. As acknowledged in the authors' reply: "Certainly if we limited ourselves to considering only those trials in which it is unambiguous as to whether the subjects were exploring or exploiting via their choices, the explore/exploit covariate the reviewer requests would eliminate the effect of I(c) simply because I(c) and directed exploration describe the same thing (picking the option about which least is known)." As the reviewer understood it at least, based on the title and abstract, the paper claims there are dedicated and independent information and reward signals in PFC. Other existing theories in the literature have argued that the role of the dACC is to decide to explore or to switch between exploratory and exploitative bouts (e.g. Karlson et al., Science, 2012). While these are clearly related, this is not synonymous with an "information system" that encodes the expected information gain of any decision at hand, or perhaps even when no decision is made at all. Other studies have shown it is possible to successfully dissociate related relative uncertainty terms from the decision to explore (e.g. Badre et al., Neuron, 2011; Trudel et al., Nature Human Behavior, 2021). Interestingly this latter study showed that the dACC (and vmPFC) signal related to relative uncertainty does in fact depend on whether subjects are in an exploratory or exploitative modes, defined largely based on the planning horizon. Thus, in the view of reviewers, these are not the same thing, and it should be acknowledged that this task design cannot in fact dissociate these ideas.

- It should be clarified whether the switch regressor was coded with respect to the previous free choice or the previous forced choice?

- The authors make good points about their conclusions in paragraphs 2 and 3 of their discussion. I would suggest some rephrasing about the ways they refer to the previous literature, though, as the current phrasing (perhaps unintentionally) makes it sound like they are opposing certain previous work, when their results actually seem quite compatible with it. For example, they cite previous work suggesting that information and reward have similar neural circuitry, then say that this "apparent" overlap in neural substrates in only observed when a confounded analysis is used that does not account for their shared variance. This sounds like they are implying that the previous findings of overlap were simply due to confounds. This doesn't make sense to me. First, the reason reward and information are intermixed in this paper, and need to be deconfounded by an analysis approach, is due to the particular experimental design where people can choose based on reward, information, or a combination of the two, leading to correlations between these variables. This is an important point since this is a common task design relevant in real life. However, most of the studies they cite do not have designs that would fall prey to this (e.g. Kang et al. and Gruber et al. where no rewards were provided from the task used for fMRI, Iigaya et al. where choices only affected information and not reward, etc.). Second, the current paper does report overlap between information and reward as well (in the striatum), and several of the papers they cite as reporting "apparent" overlap also reported it in nearby regions of striatum, or in midbrain regions or neural populations which provide input to striatum and are thought to be a major contributor to its BOLD signals in tasks like this (e.g. Kobayashi and Hsu, Charpentier et al., Bromberg-Martin and Hikosaka, Tricomi and Fiez, etc.).

So rather than this paper's findings suggesting that aspect of previous work was confounded, instead this paper replicates and supports that previous work. I would rephrase these paragraphs to make it clear that only certain experimental tasks need to be corrected for shared variance (and that it is in the context of such tasks where their finding of ACC vs vmPFC differences is interesting), and that their results are in agreement with some areas having overlap.

- line 464 to the end of the paragraph, these statements seem a little off so I would suggest changing or removing them. I am not sure how this work speaks to multidimensional versus "pure" value encoding in striatum, at least in the way indicated by the cited studies. Those studies argued that single neurons encode multiple types of value-like variables or both value and non-value variables, while this study can't distinguish with its current approach (e.g. do some striatal cells signal Information Gain while others signal Relative Reward, or are those mixed; something like an fMRI-adaptation design could give evidence on this but that is not the goal of the current study). Also, one of the cited works recorded dopamine neurons not striatal neurons, though I suppose they may influence striatal BOLD as I mentioned above. Finally, this study and the cited study do not speak to the question of whether the basal ganglia use a softmax function since both of them simply built a softmax into their analysis without attempting to test softmax against competing functions (which is fine on its own, since identifying the function was not the goal of those studies, so they should be interpreted as such).

---

## [Author Response]

[Editors’ note: the authors resubmitted a revised version of the paper for consideration. What follows is the authors’ response to the first round of review.]

The study reports evidence from fMRI for independent brain systems that value reward versus information. An elegant free choice task is used during fMRI to show that people value both reward and information. Then, using model-based fMRI and a method of deconfounding information and value, a dissociation is reported between the dACC and vmPFC tracking information and reward, respectively.The reviewers found this to be an important topic and were in largely in agreement that this dissociation would be an advance of interest to the field. However, the reviewers also identified a substantial number of serious concerns that raised doubt regarding these observations and their interpretation. Some of these are addressable with relatively straightforward additional analyses. However, others might also be addressable, but the path forward is less clear. At the very least, they would require considerable additional work and possibly collection of new data. Thus, after considerable discussion, it was decided that the work required to address these serious concerns was extensive enough that it would be beyond that expected of a revision at eLife. As a result, the consensus was to reject the paper. However, given the promise of the paper, the reviewers and editors also agreed that were the paper to be extensively reworked in order to address all of the points raised, then we would be willing to consider the paper again at eLife as a new submission.I have attached the original reviews to this letter, as the reviewers provide detailed points and suggestions there. However, I wish to highlight a few points that the reviewers emphasized in discussion and that led to our decision.Dissociating instrumental versus non-instrumental utility. To support the main conclusion that dACC does code for information value, not reward (i.e. dACC and vmPFC constitute an independent value system), one would need to show that dACC does not code for long-term reward. However, the key control analysis for ACC between information utility and inherent value is not convincing because of the assumptions it makes about people's beliefs (see comments by R1). However, addressing this might require post-hoc assessments of participant's beliefs or collection of new data. One reviewer suggested an analysis of behavior along these lines:-describe the task more clearly so that we can understand the generative process,-find a way to calculate the objective instrumental value of information in the task (or the value that participants should have believed it had, given their instructions).-fit a model to behavior to estimate subjective information value-check how much of the subjective value of information can be accounted for its objective instrumental value.Regardless of the approach taken, it was agreed that addressing this point is essential for the conclusions to be convincing.

We thank the reviewers and reviewing editor for this comment. This is indeed a crucial point. We have now included an additional analysis to account for this concern (p. 12). We computed the instrumental value of information (Instrumental Information) by implementing a Bayesian learner and estimating the Bayes optimal long-term value for the chosen option (pg. 22). We then entered Instrumental Information and Information Gain as parametric modulators into two independent GLMs.

We investigated the effects of Information Gain after controlling for Instrumental Information and, the effects of Instrumental Information after controlling for Information Gain. Results showed that dACC activity positively correlated with Information Gain, while vmPFC with Instrumental Information. These results suggest that activity in dACC was strictly related to the non-instrumental value of information, while activity associated with the instrumental value of information was expressed in reward regions consistent with Kobayashi and Hsu. PNAS. 2019.

Behavioral Fits. R3 raises some detailed concerns regarding the model fits that are serious and might impact interpretation of the model-based ROI analysis. As such, these are quite important to address. During discussion, one reviewer suggested that the problem might be that the omega and γ parameters are trading off and behavior can be fit equally well by both. This is resulting in extreme and variable fits across participants (you can see this in an apparent negative correlation between your omega and γ parameters; Spearman's rho = -0.57, p = 0.008). Other parameters are showing similar issues (α and β, for example). Among these is omega and β (correlation of about -.6), which will make separating reward and information difficult. This reviewer suggests that you consider analyzing the data in terms of the model's total estimated subjective value (i.e. -omega*I), which incorporates the effects of both the omega and the γ parameters. They also suggest that you consider reformulating the choice function to separate reward terms from information terms, for example, applying terms like Β*Q – Omega*I. Regardless of whether you take these suggestions, the model fitting and recoverability of the parameters appears to be a significant issue to address. At a minimum, some recoverability analysis and analysis of the correspondence between model's and subject's behavior will be required.

We thank the reviewers and reviewing editor for this comment. R3’sconcern might relate to extremely small values of γ parameter which when rounding these values to 3 digits results in values equal to zero for some participants. We agree with the reviewer – if γ is equal to 0, the expression evaluates to 1, indicating indifference amongst decks for information. In our revision, we have now added the log of the these values to show that the estimates for γ, while very small, are not exactly 0 (Table S2, pg. 46). Nonetheless, the point remains that for very small values of γ, the information value difference for decks with different numbers of samples is negligible. This would suggest some subjects were relatively insensitive to information differences amongst options. On its own, this does not seem too far-fetched. Nor is it impossible that the relative weighting of information value vs reward value (captured by omega) could be < 1 – subjects could in principle be sensitive to differences in information value, but simply weigh it less than reward value.

We have now added a parameter recovery analysis (pg. 37) based on Willson and Collins. *eLife*. 2019. In particular, we simulated data from gkRL using the parameters obtained from the fitting procedure (*true parameters*), and we fit the model to those simulated data to obtain the estimated parameters (*fit parameters*). We then ran a correlation for each pair of parameters. This revealed high correlation coefficients for α (r=0.8, *p* < 10^-3^), β (r=0.8, *p* < 10^-3^), omega (r=0.6, *p*=0.006) and γ (r=0.6, *p*=0.002). Although parameters are highly correlated in our model, we were able to recover all the parameters with high accuracy.

It is worth mentioning that in our previous paper (Cogliati Dezza etal. 2019) we tried to fit a model as Β*Q – Omega*I. However, the recoverability of the parameters of such a model was less accurate (it adds an extra parameter for the same number of data points) and the model was not better able to explain participants’ data.

To link model’s and participants’ behavior we simulated the gkRL using the estimated parameters. We then performed a logistic regression for each simulation predicting gkRL choices (exploitative choices=1; nonexploitative choices – or exploration = 0) with reward and number of samples as fixed effects. Logistic regression was fitted for each individual simulation and β coefficients tested against zero. Reward and number of samples significantly influenced gkRL choices (both p < 10^-5^) as observed in participants (pg. 38).

Next, we ran a model comparison analysis to estimate the degree by which gkRL explains choice behavior in our sample compared to a standard RL model- where only reward predictions influence choices. Negative log likelihoods obtained during the fitting procedure were used to compute model evidence (the probability of obtaining the observed data given a particular model). We adopted an approximation to the (log) model evidence, namely the Bayesian Information Criterion (BIC) and we compared its estimate across different models (fixed-effect comparison). Additionally, we used a random-effects procedure to perform Bayesian model selection at group level. We approximated model evidence as – BIC/2. Model comparison showed that the gkRL model was better able to explain participants’ choice behavior compared to a standard RL (xp_gkRL_=1, BIC_gkRL_=14354; xp_standardRL_=0, BIC _standardRL_=16990). These results suggest that the gkRL model was better able to explain participants’ choices compared to a standard RL. This result replicates our previous findings in which we showed that when humans play with our task gkRL was better able to explain their behavior compared to different classes of models (Cogliati Dezza et al. 2017, 2019).

As we were able to recover model parameters with high accuracy and to replicate participants’ behavior in this task, we decided to adopt gkRL model for our fmri analyses.

- Support for two independent systems. Several comments from reviewers highlighted issues that diminish support for the core separable system finding. These are well summarized in R2's comments, but they center around the definition of reward and information terms, correlations between information and reward, no test of for a double dissociation, and the lack of control for other correlated variables.The reviews contain many suggestions of ways these issues might be addressed. However, in discussion, the reviewers suggested the alternate regressors for reward be included as:1) second highest option's value (V2) – avg(value chosen, V3)2) entropy over the three options' values (V1,V2,V3)In addition to the following covariates:1) minimum value of the three decks2) maximum value of the three decks3) average value of the three decks4) RT

We thank the reviewers and reviewing editor for this comment. We address each point individually.

Definition of reward and information terms

We defined the reward term based on previous studies. Since vmPFC is commonly associated with the relative reward value of an option (e.g. Shenhav et al. 2014 Nat Neurosci) we entered this estimate into our analysis. We have now added two additional glms with Expected Reward (controlling for Relative Reward) and Relative Reward (controlling for Expected Reward) as parametric modulators (pg. 39). Activity in vmPFC positively correlated with relative reward after controlling for expected value (FWE *p* < 0.001, voxel extent = 1829, peak voxel coordinates (-6, 52, 14), t (19) = 7.21). However, vmPFC activity associated with Expected Reward disappeared after controlling for its relative estimate suggesting that vmPFC encodes the relative reward value of an option as previously suggested (e.g. Shenhav et al. 2014 Nat Neurosci). The information term was chosen based on our previous studies in which we showed that humans represent the value of information as implemented by our model compared to alternative computations when performing the behavioral task adopted in this study (Cogliati Dezza et al. 2017, 2019). We also ran GLM3 and GLM4 with relative information instead of Information Gain. The analyses did not reveal any significant activity probably due to the high correlations between Relative Information Gain and Relative Reward (for most subjects r > 0.6).

Correlation between information and reward

We acknowledge this part was not well explained in the previous version of the manuscript. Both Wilson et al. and our task orthogonalize reward and information by adding different task conditions. “In the forced-choice task, participants were forced to either choose each deck 2 times (*equal information condition*), or to choose one deck 4 times, another deck 2 times, and 0 times for the remaining deck (*unequal information condition*).” “…the use of the forced-choice task allows to orthogonalize available information and reward delivered to participants in the first free choice trial”. However, to have a better estimate of the neural activity over the overall performance we adopted trial-by-trial fMRI analyses. This introduces an information-reward confound in our analysis. Indeed, logistic regression of subjects’ behavior on the first trial shows that -9 choices were driven both by the reward (3.22, t(1,19) = 12.4, *p* < 10) -6 and information (-3.68, t(1,19) = -7.84, *p* < 10 ) experienced during the learning phase. This cofound was expressed at the neural level where the β estimates for GLM1 and GLM2 were negatively correlated across subjects for both vmPFC and dACC clusters. We addressed this confound in GLM3 and 4 by investigating activity associated with reward (information) after controlling for information (reward). We have now better explained this point in the current version of the manuscript (pg. 7, 19).

Double dissociation. We respectfully submit that the emphasis on doubly dissociating dACC and vmPFC activity is incorrect; indeed, the double dissociation between dACC and vmPFC is already present in GLM1and2 when information and reward value are not controlled for. Our argument is that the effects of reward on dACC (and information on vmPFC) are the result of the confound between information and reward, and that by controlling for this confound, these effects should disappear. Essentially, our claim is that the strength of the interaction between ROI (dACC/vmPFC) and Value (Reward/Information) depends on a third factor: whether the confound between reward and information is controlled for. Specifically, we hypothesized a weaker interaction, and this hypothesis was supported by our results from the previous version of this manuscript in which effects of reward/info in dACC/vmPFC respectively disappeared after controlling for the reward/information confound.

However, we acknowledge that these tests did not directly answer the question of whether the ROI/Value interaction strength depended on whether the cofound we identify was controlled or not. In order to do so, we conducted a 3-way ANOVA with ROI (dACC, vmpFC), Value Type (Info, Reward), Analysis type (not-confounded, confounded) and we tested the 3-way interaction term (pg. 11). Results showed a significant 3 way interaction F(1,19) = 19.74, p = 0.0003, demonstrating that controlling for the confound between information and reward modulates the ROI/Value interaction in line with our hypothesis.

Covariates. We investigated the role that additional covariates can play in modulating neural activity (pg. 39). Based on reviewers’ suggestions we individuated 5 additional covariates: the maximum value of 3 decks (Max Value), the minimum value of the 3 decks (Min Value), the reward value variation for the chosen option (Standard Deviation), the averaged value of the 3 decks (Averaged Value), the value of the chosen option minus the value of the best second option (Chosen-Second). The choice of computing Standard Deviation instead of Entropy as the reviewer suggests is as follows: To compute Entropy, option probabilities are required. In our model, however, option probabilities are the output of the softmax function which combines both reward and information values. Therefore, we could not use these probabilities to compute Entropy as they would carry both reward and information signals. However, Standard Deviation has been extensively used as an alternative computation for information about reward distributions. We first computed the correlations with RelReward. Standard Deviation and Chosen-Second were highly correlated with RelReward (most subjects showed r>0.6 for Standard Deviation and r>0.8 for Chosen-Second, Table S5 pg. 49). Given these high correlation coefficients, the variance explained by Standard Deviation/Chosen-Second and RelReward might overlap. For this reason, these covariates were not considered any further. For the remaining variables, we entered them as parametric modulators in a GLM together with RelReward allowing them to compete for variance. Using the vmPFC ROI identified in GLM2 (based on information value) we computed the averaged β estimates for each covariate. Results showed that the betas for RelReward were significantly higher than the betas for Max Value (p <-3 0.05), Min Value (p < 0.05), Averaged Value (p < 10) suggesting that the majority of the variance in vmPFC was accounted for by RelReward. This analysis justifies the use of our reward computation compared to alternative computations.

One additional note as reported in the main text, RT did not correlate with Information Gain (pg. 13), therefore our decision to not add RT as additional covariate in the analysis.

They also emphasized that at least two models of brain activity should be considered. One that includes raw value (regressors for a chosen option's expected reward and information) and one that includes relative value (relative excepted reward and information). These last approaches are detailed in the comments from reviewers.

We have now included additional glms where expected reward value where entered instead of the relative estimates. These analyses demonstrate that dACC encoded the value of information also after controlling for the variance explained by expected reward value (ExpReward; pg. 39) and that vmPFC encodes the relative value of information instead of the expected reward value (pg. 39).

- Novelty. R3's comments raised a concern regarding the novelty of the contribution on first reading. However, after discussion, this reviewer came around on this point and was persuaded of its novelty. Nonetheless, placing the present findings the context of prior work locating dissociations between dACC and other brain regions is important. This reviewer has provided important guidance along these lines.

The point of our paper was not to show that dACC and vmPFC encode different decision variables, as it has already been suggested (e.g., Daw et al. 2006, Behrens et al. 2007, Hogan et al. 2019). We used the symmetrical opposition between dACC and vmPFC as a tool to investigate the existence of a dedicated and independent value system for information in human PFC. The symmetrical opposite pattern is observed in a wide range of value-based decision-making contexts, including foraging (Kolling, Behrens et al. 2012, Shenhav, Straccia et al. 2016), risk (Kolling, Wittmann et al. 2014), intertemporal (Boorman, Rushworth et al. 2013, Wittmann, Kolling et al. 2016) and effort-based choice (Skvortsova, Palminteri et al. 2014, Arulpragasam, Cooper et al. 2018). We acknowledge that other studies have reported dissociations between dACC and vmPFC during value-based decision-making (Daw, O'Doherty et al. 2006; Behrens, Woolrich et al. 2007; Hogan, Galaro et al. 2019; Rushworth, Kolling et al. 2012; Marsh, Blair et al. 2007; Kim, Shin et al. 2014; Shenhav, Straccia et al. 2014). However, even when activity in dACC and vmPFC is dissociated, activity in vmPFC is generally linked to reward value, while activity in dACC is often interpreted as indexing negative or non-rewarding attributes of a choice including ambiguity (Silvetti, Seurinck et al. 2013), difficulty (Shenhav, Straccia et al. 2014), negative reward value (Skvortsova, Palminteri et al. 2014), cost and effort (Hillman and Bikey 2012). Model simulations further demonstrate that functional opposition between reward and information system in value-based choices may be observed even in absence of a clear symmetric opposition of activity. The interpretation of dACC and vmPFC as opposing one another therefore includes both symmetrically-opposed activity, as well as a more general functional opposition in value-based decision making. Model simulations showed that this symmetrical opposition is observed in both a model which consists of independent value systems (when not controlling for correlations) and a model which consists of a single value system. Therefore, investigating this frequently observed symmetric opposition after controlling for potential correlations between reward and information could give us a clue on whether information and reward are indeed encoded by independent value systems. We have better clarified this point in the manuscript (pg. 4)

Reviewer #1:Cogliati Dezza and colleagues report fMRI findings showing independent systems for valuing reward vs information. They use an elegant task in which participants freely choose between three decks of cards, after having gone through a sequence of forced choice trials which that control their sampling of the decks and prior information about their reward values. They operationally define information gain in terms of the number of times an option has been previously sampled (under the assumption that fewer samples = more information gain from the next sample) and show that people value both immediate reward and its informational gain. They interestingly show that simple univariate fMRI analysis of either information or reward would be confounded, but report that a multivariate analysis of activity related to the deck being chosen shows that vmPFC activations primarily track reward, dACC tracks information, and striatum tracks the choice probability reflecting its total value.The key report that information and reward may be valued by separate neural systems in frontal cortex is clearly important and valuable for the field. This would be in contrast to recent studies which primarily reported areas that have similar responses for both information and reward value (Charpentier et al., PNAS 2018; Kobayashi and Hsu, PNAS 2019). This idea is plausible, appealing, and has strong implications for brain function. However, I have some concerns about whether the manuscript is fully convincing.1. Figure 1 that sets up the paper would benefit from a clear organization where it shows in parallel the two hypotheses, the findings of univariate analysis (i.e. similar under both hypotheses), and crucially, also shows the simulated findings of the improved analysis+experiment advocated in this paper, in the same format (i.e. clearly different under the two hypotheses, presumably by showing only a non-zero red bar for the information system and only a non-zero blue bar for the reward system, and that the estimates of the two effects aren't negatively correlated).This last simulation seems crucial since their current simulations show the flaws of bad approaches but do not show the validity of their current approach, which needs to be proven to know that we can trust the key findings in Figure 4.Similarly, I would suggest retitling Figure 3 something like "APPARENT symmetrical opposition…" to make to clear that they are saying that the findings shown in Figure 3A,B are invalid. Otherwise, nothing in this figure tells the reader that they are not supposed to trust them.

We thank the reviewer for this comment, and we agree that the figure did not clearly illustrate our hypotheses. We have revised the figure to show how simulated activity from a single value and a dual value model responds to our analysis approach (pg. 5). While both models show apparent symmetrically-opposed activity when regressors are not orthogonalized, serial orthogonalization of regressors specifically abolishes this pattern only in the dual value system model (pg. 34-35).

We have also updated the title of Figure 3 as suggested by the reviewer.

2. I greatly appreciate their control analysis on a very important point: how much of the information value in behavior and brain activity in this task is due to the information's instrumental utility in obtaining points from future choices (the classic exploration-exploitation tradeoff) vs. its non-instrumental utility (e.g. satisfying a sense of curiosity). It is important to understand which (or both) the task is tapping into, since these could be mediated by different neural information systems. If this dissociation is not possible, it would be important to discuss this limitation carefully in the introduction and interpretation.I am not sure I understand the current control analysis, and am not fully confident this can be dissociated with this task. The logic of the control analysis is that if dACC is involved in long-term reward maximization its activity should be lower before choices of an unsampled deck in contexts when the two sampled decks gave relatively high rewards, presumably because the unsampled deck is less worth exploring. I think I see where they are coming from, but this is only obviously true if people believe each deck's true mean reward is drawn independently from the true mean rewards of the other two decks. However, this may not be the true setup of the task or how people understand it.The paper's methods do not describe the generative distributions in enough detail for me to understand them (the methods describe how the final payoff was generated, but not how the numbers from the 6 forced choices were generated, or how the numbers from the forced choices relate to the final payoff). But in their previous interesting behavior paper they used different High Reward and Low Reward contexts where all decks had true means that were High or Low. If they did the same in this paper then I am not sure their control analysis would make sense, since the deck values would be positively correlated.Broadly speaking, their logic seems to be that if people value information for its instrumental utility there should be a Reward x Information interaction so that they value information more for decks with high expected reward (potential future targets for exploitation) than low expected reward (not worth exploring). I am not sure if this would be strong evidence about whether information value is instrumental because it has been reported that people may also value non-instrumental information in relation to expected reward (e.g. Kobayashi et al., Nat Hum Behav 2019). But if they do want to use this logic, wouldn't it be simpler to just add an interaction term to their behavioral and neural models and test whether it is significant?Also, their current findings seem to imply a puzzling situation where participants presumably do make use of the information at least in part for its instrumental value in helping them make future choices (as shown a similar task by Wilson et al), but their brain activity in areas sensitive to information only reflects non-instrumental utility. I am not sure how to interpret this.However they address these, I have two suggestions.First, whatever test they do to dissociate instrumental vs non-instrumental utility, it would be ideal if they can do it on both behavior and neural activity. That way they can resolve the puzzle because they can find which type of utility is reflected in behavior and then test how the same utility is reflected in brain activity.Second, they apparently manipulated the time horizon (giving people from 1 to 6 free choices). If so, this nice manipulation could potentially test this, right? Time horizon has been shown by Wilson et al. to regulate exploration vs exploitation. The instrumental value of information gained by exploration should be higher for long time horizons, whereas the non-instrumental value of information (e.g. from satisfying curiosity) has no obvious reason to scale with time horizon.

These are indeed crucial points. We address them separately below.

Horizon

In our task, we did manipulate the free choice horizon. However, participants were not cued on the horizon length prior to the start of each game (pg. 19) as in Wilson et al. We have already shown that (uncued) horizon length does not affect choice in our task implementation (Cogliati Dezza et al. 2017, 2019). Therefore, although the reviewer’s suggestion is accurate (and it provides an interesting way instrumental and non-instrumental information can be dissociated in sequential decision-making tasks), it cannot be implemented here.

Behavior

We computed the instrumental value of information (Instrumental Information) by implementing a Bayesian learner and estimating the Bayes optimal long-term value for the chosen option (pg. 22). We then entered Instrumental Information and Information Gain in a logistic regression predicting first free choices with Instrumental Information and Information Gain as fixed effects and subjects as random intercepts and 0+Instrumental Information + Information Gain|subjects as random slope (pg. 13). Choices equal 1 for most informative options, and 0 for options selected 4 times during the forced choice task. We found a positive effect of Information Gain (β coefficient = 72.07 ± 16.91 (SE), z = 4.26, *p* = 10^-5^) and a negative effect of Instrumental Information (β coefficient=-2.25 ± 0.464 (SE), z = -4.85, *p* < 10^-6^) on most informative choices. The percentage of trials in which most informative choices had positive Instrumental Information was ~ 22% suggesting that in most of the trials informative options were not selected based on instrumental utility.

Brain activity

We entered Instrumental Information and Information Gain as parametric modulators into two independent GLMs (pg. 13). We investigated the effects of Information Gain after controlling for Instrumental Information and, the effects of Instrumental Information after controlling for Information Gain. Results showed that dACC activity positively correlated with Information Gain (FWE *p* < 0.001, voxel extent = 1750, peak voxel coordinates (10, 26, 56), t(19) = 8.19), while vmPFC with Instrumental Information (FWE *p* < 0.001, voxel extent = 557, peak voxel coordinates (20, 20, -10), t(19) = 5.62). These results suggest that activity in dACC signals the non-instrumental value of information, while activity associated with the instrumental value of information was expressed in reward regions consistent with Kobayashi and Hsu. PNAS. 2019.

Generative distributions

On each trial, the payoff was generated from a Gaussian distribution with a generative mean between 10 and 70 points and standard deviation of 8 points. On each game, the generative mean for each deck was set to a base value of either 30 or 50 points and adjusted independently by +/- 0, 4, 12, or 20 points with equal probability, to avoid the possibility that participants might be able to discern the generative mean for a deck after a single observation. The generative mean for each option was stable within a game, but varied across games. This reward distribution was the same for forced and free choice trials. The only difference was that in the forced-choice trials participants were not free to choose. Participants’ payoff on each trial ranged between 1 and 100 points and the total number of points was summed and converted into a monetary payoff at the end of the experimental session (0.01 euros every 60 points). Rewards experienced during the forced-choice task were not added to participants’ final payoff. We have now added this information at pg. 19

3. They analyze the first free-choice trial because "information and reward are orthogonalized by the experiment design", but is this true? They say Wilson et al. showed this in their two option task but it is unclear it applies to this three option task. Particularly because they do not analyze activity associated with all decks, but rather in terms of the chosen deck, and participants chose decks based on both info and reward. Depending on a person's choice strategy the info and reward of the chosen deck may not be orthogonal, even though the info and reward considered over all possible decks they could choose are orthogonal.Basically, it would be valuable to prove their point by including supplemental scatterplots with tests for correlation between whichever information and reward regressors they use in the behavioral model and in the neural model. If they do have some correlation, they should show their analysis handles them properly.

We acknowledge this part was not well explained in the previous version of the manuscript. Both Wilson et al. and our task orthogonalize reward and information by adding different task conditions. “In the forced-choice task, participants were forced to either choose each deck 2 times (*equal information condition*), or to choose one deck 4 times, another deck 2 times, and 0 times for the remaining deck (*unequal information condition*).” “…the use of the forced-choice task allows to orthogonalize available information and reward delivered to participants in the first free choice trial”. However, in order to better estimate the neural activity over the overall performance we adopted a trial-by-trial model-based fMRI analyses. Since reward value and information value are correlated due to subjects’ choices during the experiment, this introduces an information reward confound in our analysis. Indeed, logistic regression of subjects’ behavior on the first trial shows that choices were driven both by the reward (3.22, t(1,19) = 12.4, *p* < 10^-9^) and information (-3.68, t(1,19)=-7.84, *p* < 10^-6^) experienced during the learning phase. This cofound was expressed at the neural level where the β estimates for GLM1 and GLM2 were negatively correlated across subjects for both vmPFC and dACC clusters. We then took care of this confound in GLM3 and 4 by investigating activity associated with reward (information) after controlling for information (reward). We have now better explained this point in the current version of the manuscript (pg. 7, 19).

4. I am a little confused by how they compare information value and reward value. In the model from their previous paper and used in this paper they say, roughly speaking, that the value of a deck = (expected reward value from the deck) + w*(expected information value from the deck). So, I thought that they would use the reward and information value from the chosen deck as their regressors. But they subtract the reward values of other decks from the reward term. I am not sure why. They cite studies saying that "VmPFC appears not only to code reward-related signals but to specifically encode the relative reward value of the chosen option", but if so, then if they want to test if vmPFC codes information-related signals shouldn't they also test if it encodes the relative information value? (Or does this not matter because It,j(c==1) is linearly related to It,j(c==1) – mean(It,j(c==2),It,j(c==3)) in this task so the findings would be the same in either case?). And conversely, if dACC encodes information value of a deck shouldn't they test if it encodes the reward value of that deck, instead of relative reward value? It's also confusing that in this paper and their previous paper they argue that.Basically I want to make sure the difference in vmPFC vs. dACC coding is truly due to reward vs information coding rather than relative vs non-relative coding.

This is indeed a really important point. First of all, as our goal wasto dissociate activity in vmPFC and dACC and since vmPFC is commonly associated with the relative reward value of an option (e.g. Shenhav et al. 2014 Nat Neurosci) we entered this estimate into our analysis. We have now added two additional glms with Expected Reward (controlling for Relative Reward) and Relative Reward (controlling for Expected Reward) as parametric modulators (pg. 39). Activity in vmPFC positively correlated with relative reward after controlling for expected value (FWE *p* < 0.001, voxel extent = 1829, peak voxel coordinates (-6, 52, 14), t (19) = 7.21). However, vmPFC activity associated with Expected Reward disappeared after controlling for its relative estimate suggesting that vmPFC encodes the relative reward value of an option as previously suggested (e.g. Shenhav et al. 2014 Nat Neurosci).

To control for ‘relativeness’, we have now included an additional analysis in which the Reward regressor was defined as expected reward values, instead of relative reward values in GLM3 and 4 (pg. 39). We observed that activity in dACC was still associated with Information Gain after accounting for variance related to expected reward values, suggesting that the coding wasn’t related to ‘relativeness’ but to reward itself.

We also ran GLM3 and GLM4 with relative information instead of Information Gain. The analyses did not reveal any significant activity probably due to the high correlations between relative reward and relative information (for most subjects r > 0.6).

Reviewer #2:This is an interesting and well-motivated study, using a clever experimental design and model-based fMRI. The authors argue their results support two independent value systems in the prefrontal cortex, one for reward and one for information. Based on separate GLMs, it is argued for a double dissociation, with relative reward (but not information gain) coded in vmPFC and information gain (but not relative reward) encoded in dACC. While I am sympathetic to this characterization, I have concerns about several of the key analyses conducted, and their interpretation that, I believe, call into question the main conclusions of the study.1. Correlations between relative reward and information gain terms. The methods state: "Because information and reward are expected to be partially correlated, the intent of GLMs was to allow us to investigate the effects of the second parametric modulator after accounting for variance that can be explained by the first parametric modulator." I had thought the purpose of the task design was to orthogonalize reward and information. What is the correlation between regressors and what is driving the correlation? Are these terms correlated because people often switch to choose options that have high expected reward but have also not been sampled as frequently, in the unequal information trials? If these terms were in fact experimentally dissociated, it should be possible to allow them to compete for variance within the same GLM without any orthogonalization. This is an important issue to address in order to support the central claims in the paper.

We thank the reviewer for this comment. We acknowledge this part was not well explained in the previous version of the manuscript. Both Wilson et al. and our task orthogonalize reward and information by adding different task conditions. “In the forced-choice task, participants were forced to either choose each deck 2 times (*equal information condition*), or to choose one deck 4 times, another deck 2 times, and 0 times for the remaining deck (*unequal information condition*).” “…the use of the forced-choice task allows to orthogonalize available information and reward delivered to participants in the first free choice trial”.

However, to have a better estimate of the neural activity over the overall performance we adopted trial-by-trial model-based fMRI analyses. This introduces an information-reward confound in our analysis because (1) regressors for reward value and information value are applied to every trial and (2) reward and information values are correlated due to subjects’ choices. Indeed logistic regression of subjects’ behavior on the first trial shows that choices were driven both by the reward (3.22, t(1,19) = 12.4, *p*< 10^-9^) and information (-3.68, t(1,19) = -7.84, *p* < 10^-6^) experienced during the learning phase. This confound was expressed at the neural level where the β estimates for GLM1 and GLM2 were negatively correlated across subjects for both vmPFC and dACC clusters.

To account for this confound we could either select only those trials in which reward and information are orthogonal by design (e.g., unequal info condition for dACC activity or equal information condition for vmPFC activity) or we could implement GLM3 and 4 where activity associated with reward (information) was controlled by activity associated to information (reward). Implementing GLM3and4 would result in increased power compared to the first analysis (because it considers 128 trials instead of ~60 trials, see also the answer to point 2). Therefore, we decided to go for this option.

We have now better explained this point in the current version of the manuscript (pg. 7, 19).

2. If dACC is specifically encoding the "information value" then there should be no effect of the relative value on equal information condition trials. By design, these trials should appropriately control for any information sampling difference between options. What is the effect of the negative relative value in the dACC on these equal information trials (or similar terms as defined in my next point)?

We thank the reviewer for this comment. Indeed, dACC activity should show up only in unequal trials. We ran the analyses suggested by the reviewer (i.e., GLM1and2 on equal and unequal trials separately) however all the contrasts gave no significant activity at uncorr <0.001. The number of trials in each condition is around 63 which may result in not enough power to test what the reviewer asked. However, our GLM3and4 do the same job as computing GLM1and2 separately for task conditions. To better strengthen the claim dACC computes the value of information we have introduced two additional control analyses: (a) controlling for reward value (pg. 39; details below) and (b) controlling for instrumental value of information (pg. 12; details below). The results of these analyses all point towards a role of dACC in encoding the non-instrumental value of information.

a) We have now included an additional glm where Reward regressor was defined as expected reward values, instead of relative reward values in GLM4. We observed that activity in dACC was still associated with Information Gain after taking out the variance related to expected reward values.

b) We computed the instrumental value of information (Instrumental Information) by implementing a Bayesian learner and estimating the Bayes optimal long-term value for the chosen option (pg. 22). We then entered Instrumental Information and Information Gain as parametric modulators into two independent GLMs. We investigated the effects of Information Gain after controlling for Instrumental Information and, the effects of Instrumental Information after controlling for Information Gain. Results showed that dACC activity positively correlated with Information Gain (FWE *p* < 0.001, voxel extent = 1750, peak voxel coordinates (10, 26, 56), t(19) = 8.19), while vmPFC with Instrumental Information (FWE *p* <0.001, voxel extent = 557, peak voxel coordinates (20, 20, -10), t(19) = 5.62). These results suggest that activity in dACC signals the non-instrumental value of information.

3. Relative value definition. In the analyses presented, the negative effect of relative reward in dACC is going in the predicted direction (negatively) and looks marginal. This may result from how the relative reward term was defined. What does the effect in dACC look like if relative reward is instead defined as V2 – V1, V2 – avg(V2,V3), or even entropy(V1,V2,V3)? A similar question applies to the vmPFC analyses. Did the relative value defined as the Vc – avg(V2, V3) explain VMPFC activity better than Vc – V2 or even -entropy(V1,V2,V3)?

We are not sure to which analyses the reviewer refers to. In our univariate analyses (Figure 2G) the reviewer is correct that the negative effect of reward is marginal. Our subsequent model-based analyses, however, indicate that the effect of reward on dACC activity (without controlling for confounds between reward value and information value) significantly correlated with the relative reward value of the chosen option in GLM1 with FWE p = 0.001. Therefore, the effect does not seem to be marginal. In GLM3, instead, no dACC activity was associated with the relative reward value of the chosen option (i.e., nothing survived at p uncorr 0.0010).

However, the reviewer is right in saying that the way we computed values in the reward dimension might influence our results.

We addressed this point in two ways. We first added an additional analysis accounting for different ways reward value could be computed and we checked which explained most of the variance in vmPFC (point a, pg. 39). This analysis showed that relative reward value accounted for the most of the variance in vmPFC. This would suggest that adding alternative computations of reward value in the glm might not change the results concerning dACC and the value of information. However, we also tested if this was true by entering expected reward value and the instrumental value of information, instead of relative reward value, into GLM4 (point b, pg. 39).

a) Following this suggestion, and those of other reviewers, we individuated 5 additional ways reward could be computed by vmPFC. This includes: the maximum value of 3 decks (Max Value), the minimum value of the 3 decks (Min Value), the reward value variation for the chosen option (Standard Deviation), the averaged value of the 3 decks (Averaged Value), the value of the chosen option minus the value of the best second option (Chosen-Second). The choice of computing Standard Deviation instead of Entropy (as suggested by the reviewer) is the following. To compute Entropy, option probabilities are required. In our model, however, option probabilities are the output of the softmax function which combines both reward and information values. Therefore, we could not use these probabilities to compute Entropy as they would carry both reward and information signals. However, Standard Deviation has been extensively used as an alternative computation for information about reward distributions. We first computed the correlations with RelReward. Standard Deviation and Chosen-Second were highly correlated with RelReward (most subjects showed r>0.6 for Standard Deviation and r>0.8 for Chosen-Second). Given these high correlation coefficients, the variance explained by Standard Deviation/Chosen-Second and RelReward might overlap. For this reason, these covariates were not considered any further. For the remaining variables, we entered them as parametric modulators in a GLM together with RelReward allowing them to compete for variance. Using the vmPFC ROI identified in GLM2 (based on information value) we computed the averaged β estimates for each covariate. Results showed that the betas for RelReward were significantly higher than the betas for Max Value (p < 0.05), Min Value (p < 0.05), Averaged Value (p < 10^-3^) suggesting that the majority of the variance in vmPFC was accounted for by RelReward. This analysis justifies the use of our reward computation compared to alternative computations.

b) When we replaced the relative reward value with expected reward values or instrumental information (which is another way to formalize the reward benefits of exploring an option) in GLM4, we observed activity in dACC associated with information value after controlling for the expected reward value. This suggests that no matter how the reward value is computed, dACC signals the non-instrumental value of information.

4. Double dissociation. The authors are effectively arguing that their data supports a double dissociation between VMPFC encoding relative reward but not information gain and dACC for information gain but not relative reward, but this is not directly tested. This claim can be supported by a statistical interaction between region and variable type. To test this hypothesis appropriately, the authors should therefore perform an F-test on the whole brain or a rmANOVA on the two ROIs, using the β coefficients resulting from a single GLM.

We respectfully submit that the emphasis on doubly-dissociating dACC and vmPFC activity is incorrect; indeed, the double dissociation between dACC and vmPFC is already present in GLM1and2 when information and reward value are not controlled for. Our argument is that the effects of reward on dACC (and information on vmPFC) are the result of the confound between information and reward, and that by controlling for this confound, these effects should disappear. Essentially, our claim is that the strength of the interaction between ROI (dACC/vmPFC) and Value (Reward/Information) depends on a third factor: whether the confound between reward and information is controlled for. Specifically, we hypothesized a weaker interaction, and this hypothesis was supported by our results from the previous version of this manuscript in which effects of reward/info in dACC/vmPFC respectively disappeared after controlling for the reward/information confound.

However, we acknowledge that these tests did not directly answer the question of whether the ROI/Value interaction strength depended on whether the cofound we identify was controlled or not. In order to do so, we conducted a 3-way ANOVA with ROI (dACC, vmpFC), Value Type (Info, Reward), Analysis type (not-confounded, confounded) and we tested the 3-way interaction term (pg. 11). Results showed a significant 3 way interaction F(1,19) = 19.74, p = 0.0003, demonstrating that controlling for the confound between information and reward modulates the ROI/Value interaction in line with our hypothesis.

5. Information gain or directed exploration (or both)? Based on the equal and unequal information conditions that were all modeled in the GLMs, it seems likely that the information gain would be high (0 or 2 samples in the unequal information trials) only when subjects choose the more uncertain option, which is equivalent to directed exploration (provided the relative reward is controlled for). Conversely, it would be low when subjects choose the better-known option and are therefore exploiting known information, and intermediate in the equal information condition. Thus, would it be fair to say that the activity in dACC could equally be driven by directed exploration decisions (also equivalent to switching away from a better known "default" option), rather than necessarily "coding for information gain" per se? One way to address this is to show that the information gain is still reflected when subjects choose the more sampled options, or to include an additional explore/exploit regressor in the GLM.

The reviewer raises an important point. Indeed, a possible explanation for dACC activity associated with the non-instrumental value of information is that dACC encodes exploration or the tendency to switch to alternative options. However, in our task the frequency of choosing the most informative option (i.e., exploration) was higher than the frequency of choosing the two other alternatives (in unequal condition, mean = 68.2%, SD = 16.8%), consistent with a default exploration strategy, and the switching strategy was to not choose the most informative options. If this is correct, regions associated with switching strategies (e.g., frontopolar cortex Daw et al. 2006) should be activated when participants did not select the most informative option. We conducted one sample ttest on the β weights estimated for a glm which comprises of a regressor modelling choice onset associated with the most informative options (Default), and another regressor modelling choice onset associated with the other two options (Switch). Activity in frontopolar cortex was positively correlated with Switch (Switch-Default; FWE *p* = 0.019, voxel extent = 148, peak voxel coordinates (-18, 52, 6), t(19) = 5.71). This suggests that adopting a switching strategy in our task was associated with not choosing the most informative options, and not with choosing highly informative options. Therefore, dACC seems to encode information gain rather than a tendency to switch towards alternative options. We have now added this analysis at (pg. 14)

On a side note, in our analyses we entered all free choice trials so that the information gain reflects all choices (whether is 0, 2 or 4). This suggests that dACC activity is modulated by the information gain regressor.

6. Background literature in the introduction. Some of the assertions concerning vmPFC and dACC function in the introduction could be further queried.For example, the introduction states: "In general, vmPFC activity appears to reflect the relative reward value of immediate, easily-obtained, or certain outcomes, while dACC activity signals delay, difficulty, or uncertainty in realizing prospective outcomes." While I do agree that some evidence supports a distinction between current outcomes in vmPFC and prospective outcomes in dACC, there is considerable evidence that vmPFC only reflects the relative chosen value for trials in which a comparison is more difficult and integration amongst attributes is required. For example, in both fMRI and MEG studies, the vmPFC effect of value difference goes away for "no-brainer" decisions in which both the reward probability and the magnitude are greater (dominant) for one of the options, rendering integration unnecessary and the decision easier – see Hunt et al., Nature Neurosci. 2012, PLoS CB, 2013; Jocham et al., Nature Neurosci., 2014. In addition, vmPFC is typically found to reflect relative subjective value in inter-temporal choice studies (e.g. Kable and Glmicher, Nature Neurosci., 2007; Nicolle et al., Neuron., 2011, etc.). Interestingly, vmPFC does not show relative value effects for physical effort-based decisions, whereas the dACC does (see Prevost et al., J. Neurosci. 2010; Lim et al., Klein-FLugge et al. J. Neurosci., 2016; Harris and Lim, J. Neurosci., 2016). Thus, vmPFC relative value effects are observed for difficult decisions requiring integration and comparison, including for uncertain and delayed outcomes, but potentially not when integrating effort-based action costs. This is also consistent with lesion work in humans and monkeys (Rudebeck et al., J. Neurosci., 2008; Camille et al., J. Neurosci., 2011).In addition, there is considerable evidence from single unit recording to support the interpretation that ACC neurons do not only encode information gain or information value but integrate multiple attributes into a value (or value-like) effect, even in the absence of any possibility for exploration and, thus, information gain. For example, Kennerley et al., JoCN 2009, Nature Neurosci. 2011 have shown that ACC neurons, in particular, multiplex across multiple variables including reward amount, probability, and effort such that neurons encode each attribute aligned to the same valence (e.g. positive value neurons encode reward amount and probability positively and effort negatively, and negative value neurons do the opposite), consistent with an integrated value effect, even in the absence of any information gain. This literature could be incorporated into the Introduction, or the assumptions toned down some.

We thank the reviewer for this comment. We have now better explained our claim in the introduction. In particular, we adopted the frequently observed symmetry between dACC and vmPFC only as a tool to test whether a dedicated and independent value system for information exists in human PFC. Simulations of two computational models, one with a single value system and one with independent value systems, demonstrate opposed activity for reward value and information gain when these two factors are confounded (pg 5, 34-35). Under the single value system model, symmetrically-opposed activity should still be observed after controlling for the confound. In contrast, under the independent value system model, removing the confound between information gain and reward value should abolish symmetric opposition.

We acknowledge however that although this symmetrical activity is observed across a wide range of value-based decision-making contexts, other studies have reported dissociations between dACC and vmPFC during value-based decision-making, or have assigned different roles for these regions in decision-making. We respectfully submit, however, that even when activity in dACC and vmPFC is dissociated, activity in vmPFC is generally linked to reward value, while activity in dACC is often interpreted as indexing negative or non-rewarding attributes of a choice (including ambiguity, difficulty, negative reward value, cost and effort). Our model simulations further demonstrate that functional opposition between reward and information system in value-based choices may be observed even in absence of a clear symmetric opposition of activity (pg 36). The interpretation of dACC and vmPFC as opposing one another therefore includes both symmetrically-opposed activity, as well as a more general functional opposition in value-based decision-making.

In the papers cited by the reviewer, vmPFC doesn’t not encode “difficult” attribute rather it encodes the relative reward value when the comparison is difficult. This doesn’t seem in contradiction to what we are claiming. Concerning the effort, this is in line with the role of dACC, but not vmPFC, in signaling effort. Additionally, ACC neurons seem to encode attributes associated to reward value, but not the reward value itself. Those attributes can be informative for the monkey. Lastly, we agree that the picture we are offering on the role of these two regions in decision making is not exhaustive. However, our goal was not to provide a new theory over vmPFC and dACC function. It was rather to use these two regions to show our point of independent value signals. Given that activity in these two regions is frequently observed in decision-making tasks and these activities are usually opposed to one and other, we decided to focus on the two regions to demonstrate our point.

7. Experimental details. Some key details appear to be omitted in the manuscript or are only defined in the Methods but are important to understanding the Results. For instance:– The information gained is defined as follows: "negative value It, j(c) relates to the information to be gained about each deck by participants". However, it is unclear for which choice (deck) this is referring to. I assume it is for the chosen deck on the first free choice trial? Or is it the sum/average over the decks? This should be made clearer.

It is indeed for the chosen option. As detailed at line 590 “In GLM2, a single parametric modulator was included using the negative value of gkrl model-derived information gained from the chosen option (It, j(c)) ”. We have now clarified some of the experimental details in the results as well to help the reader better understanding them (pg. 9).

– It would be helpful it the relative reward and Information gain regressors were defined in the main text.

We have now added their definition in the main text pg. 9.

– What details were subjects told about the true generative distributions for reward payouts.

Participants were told that during the forced-choice task they may sample options at different rates, and that the decks of cards did not change during each game, but were replaced by new decks at the beginning of each new game. However, they were not informed of the details of the reward manipulation, or the underlying generative distributions adopted in the experiment. We have now included additional details at pg. 19.

– What do subjects know about the time horizon, which is important for modulating directed exploration? This could assist in interpreting the dACC effect in particular. Indeed, manipulating the time horizon was a key part of the Wilson et al. original task design and would assist in decoupling the reward and information, if it is manipulated here.

In contrast to Wilson et al. ‘s task, the free choice trial length or horizon was not cued to participants. Therefore, on each game participants were not aware of the free choice horizon. We have already shown that the choice horizon does not affect participants’ choices on this task (Cogliati Dezza et 2017, 2019). Even if not relevant for the goal of this study, we have also shown that directed exploration can be individuated using 3 options (Cogliati Dezza et 2017). The decoupling of information and reward in our task is done by using model-based approach. We have now included additional details at pg. 19.

– How are the ROIs defined? If it is based on Figure 2F, the activation called dACC appears to be localized to pre-SMA. This should therefore be re-labelled or a new ROI should be defined.

We thank the reviewer for this comment. First, we acknowledge the results we report for dACC (including Figure 2f) overlap significantly with preSMA. We note this overlap in our revised manuscript at pg 7. It is frequently the case that fMRI studies of decision making/foraging/cognitive control/etc identify a medial ROI similar to those reported in this manuscript with significant intrusion in preSMA as ‘dACC;’ we therefore adopt that term to be consistent with the literature our current study addresses.

With respect to the ROIs used to extract β values for correlations, these were defined based on the clusters passing cluster-level correction (voxelwise threshold p=0.001) for each GLM. Note this does not bias our analyses (“double-dipping”) since we are correlating activity between ROIs rather than reporting the correlation of a ROI with a term used to identify the ROI (e.g., it would be double dipping if we were to correlate vmPFC/ACC activity with parameters for information and reward derived from model fits).

8. Brain-behavior relationships. If dACC is specifically important for information gain computations, then it would follow that those subjects who show more of a tendency for directed exploration behaviorally would also show a greater neural effect of information gain in ACC and vice versa for vmPFC and relying on relative reward values. Is there any behavioral evidence that can further support the claims about their specific functions?

We thank the reviewer for this comment. However, all the neural activity relates to subjects’ behavior. We obtained the parametric modulators by fitting our computational model to participants’ data and then we entered them into the fmri analyses. Therefore, both vmPFC and dACC activity are already related to behavior.

9. Reporting of all GLM results. It would be helpful to produce a Table showing all results from all GLMs. For example, where else in the brain showed an effect of choice probability, etc.?

We have now added a table in the supplementary reporting theactivity which is not reported in the main or supplementary text (Table S4, pg. 48).

Reviewer #3:The manuscript describes an fMRI study that tests whether vmPFC and dACC independently code for relative reward and information gain, respectively, suggesting an independent value system. The paper argues that the two variables have been confounded in previous studies, leading researchers to advocate for a single value system in which vmPFC and dACC serve functionally opposed roles, with vmPFC coding for positive outcome values (e.g. reward) and dACC coding for negative outcome values (e.g. difficulty, costs or uncertainty). The study deploys a forced-choice and a free-choice version of a multi-armed bandit task in which participants select between three decks to maximize reward. It also includes an RL model (gkRL) that is fit to participant's behavior. Variables of the fitted model are used as GLM regressors to identify regions with activity related to relative reward, as well as information gain. The study finds symmetrical activity in vmPFC and dACC (e.g. vmPFC correlates positively, dACC negatively with reward) if correlations between relative reward and information gain are not taken into account. In contrast, the study finds that vmPFC and dACC independently code for relative reward and information gain, respectively, if confounds between reward and information gain are accounted for in the analysis. The manuscript concludes that these findings support an independent value system as opposed to dual value system of PFC.I found this paper to be well-written and consider it to be an interesting addition to a rich literature on neural correlates of information. However, I got the impression that the manuscript oversells the novelty of its results and fails to situate itself in the existing literature (Comments 1-2). Moreover, some of the key references appear to be misrepresented in favor of the study's conclusions (Comment 3). While most of the methodology appears sound (though see Comment 6 in the statistical comments section), the RL model yields questionable model fits (Comment 4). The latter is troubling given that reported analyses rely on regressors retrieved from the RL model. Finally, the study may be missing a proper comparison to alternative models (Comment 5).Below, I elaborate on concerns that will hopefully be useful in reworking this paper for a future submission.1. Novelty of the findings: A main contribution of the paper is the finding that vmPFC and dACC code independently for reward and information gain, respectively, when accounting for a confound between reward and information. However, it should be noted that several other decision-making studies found a similar distinction between vmPFC and dACC by orthogonalizing reward value and uncertainty (e.g. Blair et al. 2006, Daw et al., 2006, Marsh et al. 2007, FitzGerald et al., 2009; Kim et al., 2014). In light of this work, the novelty distinction drawn in this manuscript appears exaggerated. Rather than advocating for functionally opposing roles between dACC and vmPFC, these articles support interpretations similar to the Dual Systems View. While the present manuscript includes a citation of such prior work ("other studies have reported dissociations between dACC and vmPFC during value-based decision-making [citations]", lines 52-53), the true distinction remains nebulous.

We thank the reviewer for this comment. While we acknowledge that previous studies have indeed dissociated vmPFC and dACC activity as the reviewer indicates, we respectfully disagree that these previous results negatively impact the novelty of our findings. The studies noted by the reviewer in this comment identify vmPFC as indexing reward in some form (as do we in this manuscript), and variously associate dACC with uncertainty, set size, conflict, or choice difficulty. Notably absent from these interpretations of dACC function is that of information value, i.e., while previous studies may identify effects consistent with a dual value systems view, none of them have explicitly framed their results in those terms. Indeed, our results directly challenge Daw et al’s finding of no ‘uncertainty bonus’ in their task. Thus, the unique contribution of this manuscript lies not only in dissociating vmPFC and dACC, but also providing an explanation why these regions may at times show opposed patterns of activity, and at other times not. Specifically, by questioning the validity of reward value as a singular framework for interpreting neural function.

Moreover, previous studies on human information-seeking suggested an overlap between the neural networks involved in information processing with those involved in reward processing (Kobayashi and Hsu, 2019, PNAS; Charpentier et al. 2018, PNAS). Here, we show that this is the case only for the instrumental value of information; our results suggest that the non-instrumental value of information relies on neural substrates independent of those commonly associated with reward processing. Lastly, evidence for a role of dACC in encoding the non-instrumental value of information has only been shown in monkey (White et al. 2019 Nat Com).

2. Relevance of the studied confound: I commend this study for addressing a confound between reward and information in multi-armed bandit tasks of this type. However, the manuscript overstates the generality of this confound, as a motivation for this study. A reader of the manuscript might get the impression that most of the relevant literature is affected by this confound, when in fact the confound only applies to a subset of value-based decision-making studies in which information can be gained through choice. It is unclear how the confound applies to decision-making studies in which values for stimuli cannot be learned through choice (e.g. preference-based decisions). A large proportion of decision-making research fits this description. It is not clear how value would be confounded with information in this domain. Finally, the study focuses on the *relative* value of options, whereas many studies do not (see the references above).

The reviewer raises an important point. First, we have now added an additional glm where reward was computed as expected reward value. This analysis shows the same symmetrical opposition between dACC and vmPFC (pg. 38).

Second, while we acknowledge that this confound may not emerge in every human decision, we submit that the fundamental tension between information and reward value is prevalent in decision-making. As the reviewer notes, preference-based decision-making (e.g., intertemporal choice, effort-based choice, food preferences) constitutes a large proportion of the literature on decision-making. Typically, subjective preferences are not learned during the course of an experiment but reflect a subject’s previous experience – presumably the expression of a preference for one type of food over another is consistent, and therefore the subject reliably selects that food type over others on a regular basis. In other words, the reward value of the food is confounded with its information value. Generally, one cannot control for this confound when eliciting preferences since those preferences are baked in by prior experience. Thus, while our study focuses on a bandit task, the dissociation between information value and reward value is of more general interest *especially* for studies in which the confound cannot be controlled for.

Lastly, our results suggest that other decision dimensions involved in most decision-making tasks (e.g., effort and motivation, cost, affective valence, or social interaction) may also be confounded in the same manner. Indeed, symmetrical opposition between dACC and vmPFC has been reported for a wide range of contexts involving decision variables such as effort, delay, and affective valence.

We have now clarified this point at pg. 18.

3. Mischaracterization of the literature: In Table 1, and throughout the text, the manuscript lists studies in which vmPFC and dACC are shown to be positively or negatively correlated with decision variables. It is then implied that these studies advocate a "Single Value System", i.e. that regions such as vmPFC and dACC constitute a "single distributed system that performs a cost-benefit analysis" (line 36). Yet, most of the cited work does not advocate for this interpretation. For instance, cited work on foraging theory (e.g. Rushworth et al., 2012) makes the opposite claim, suggesting that vmPFC and dACC are involved in different forms of decision-making. Furthermore, the cited study by Shenhav et al. (2016) neither explicitly promotes the interpretation that vmPFC and dACC show "symmetrically opposed activity", nor does it suggest that they are in "functional opposition" in value-based choice (lines 53-58). Instead, the authors of Shenhav et al. (2016) suggest that their findings support the view that vmPFC and dACC are associated with distinct roles in decision-making, such as vmPFC being associated with reward, and dACC being associated with uncertainty or conflict. Another cited study from Shenhav et al. (2019) also does not advocate for "emerging views regarding the symmetrically opposing roles of dACC and vmPFC in value-based choice" (line 336). Instead, the cited article draws a strong distinction between (attentionally-insensitive) reward encoding in vmPFC vs. (attention-dependent) conflict coding in dACC.

We respectfully submit that our interpretation of the literature does not mischaracterize it in the way the reviewer suggests. We acknowledge (and have indicated in our manuscript pg. 4) that not all studies we cite posit a single value system in the sense that the regions involved contribute to the calculation of a single value signal driving behavior. Nevertheless, the functional differences between regions are consistently interpreted in the framework of reward value, and the functional opposition between regions stems from attempting to identify the axis along which reward value is optimized. The different forms of decision making outlined in Rushworth et al. (2012) differentiate between immediate choices (vmPFC) or alternative actions (ACC) – essentially an opposition along a temporal axis for short-term and long-term reward value (see also Kolling et al. 2018 in which distinct regions are identified coding for prospective vs myopic value).

In Shenhav and colleagues’ Expected Value of Control (2013) framework, which informed (and is specifically invoked in) their 2016 and 2019 studies, uncertainty and conflict are explicitly identified as opponent to reward and value of information in the computation of a net (reward) value of a control signal. These studies are therefore direct examples of ACC and vmPFC activity being interpreted as participating in a single reward value system.

4. Model fits: The study's main fMRI results (e.g. that dACC independently codes for information gain) are based on regressors extracted from variables of the fitted RL model (e.g. information gain). Yet, more than a third of the participants yields model fits that could lead one to question the validity of the GLM regressors, in particular the regressor for information gain. One of the fitted model parameters, labeled "γ", exponentiates the history of previous observations to compute the amount of information associated with a deck. In line 420 it is written that "[γ] is constrained to be > 0". However, γ appears to be 0 for three out of twenty participants (6, 11, 14, see Table S2). If I understand the model correctly, then information associated with any deck (line 416) would evaluate to 1 for those participants, irrespective of which decks they observed. That is, the model fit suggests that these participants show no sensitivity to information across decks. A second parameter, omega, appears to scale how much information gain weighs into the final value for a deck (line 425). Again, if I understand correctly, the lower omega the less the RL agent weighs information gain when computing the final value associated with a deck. Thus, if omega evaluates close to 0, the agent would barely consider information gain when choosing a deck based on its final value. Given this, I am surprised that omega varies vastly in magnitude across participants, ranging from 0.26 to 48.66. Four out of twenty participants (1, 3, 19, 20, see Table S2) have fitted values for omega smaller than 1. This makes one question whether these participants actually use information to guide their decision making, relative to other participants. Relatedly, I was a bit surprised that the GLM implements – I_{t,j}(c) as regressor for information gain, instead of – omega * I_{t,j}(c). If omega is small, then the former regressor may not be a good proxy for information gain that is actually used to guide decision-making. Altogether, these model fits (seven out of twenty participants) may undermine the validity of "information associated with a deck" when used as a regressor for information gain in the fMRI analysis (at least in GLMs 2, 3 and 4).

We agree with the reviewer – if γ is equal to 0, the expression evaluates to 1, indicating indifference amongst decks for information. In our revision, we have now added the log of these values to show that the estimates for γ, while very small, are not exactly 0 (Table S4, pg. 46). Nonetheless, the point remains that for very small values of γ, the information value difference for decks with different numbers of samples is negligible. This would suggest some subjects were relatively insensitive to information differences amongst options. On its own, this does not seem too far-fetched. Nor is it impossible that the relative weighting of information value vs reward value (captured by omega) could be < 1 – subjects could in principle be sensitive to differences in information value, but simply weigh it less than reward value.

On an additional note, parameters were estimated by fitting the model to all first free choices, the 50% of which by design contains choices towards options equal in information. When we look at the unequal condition participants chose on average the informative option (the option selected 0 times during the forced choice-task) in about 65% of those trials suggesting that they prefer informative options. This effect however is softened when fitting the model to all first free choices (which is needed to have a better estimate of the 4 parameters). Lastly, logistic regression of subjects’ behavior on the first trial shows that indeed choices (i.e., exploitative choices) were driven both by the experienced reward of each option (3.22, t(1,19) = 12.4, *p* < 10^-9^) and amount of time each option was selected (-3.68, t(1,19) = -7.84, *p* < 10^-6^). The logistic regions was run for each subject individually and for all of them both reward and information coefficients were significant.

To answer the reviewer’s concern regarding the variability in our parameter estimates, we have now added a model comparison analysis to estimate the degree by which gkrl explained choice behavior in our sample compare to a standard RL model- where only reward predictions influence choices (pg. 38). Negative log likelihoods obtained during the fitting procedure were used to compute model evidence (the probability of obtaining the observed data given a particular model). We adopted an approximation to the (log) model evidence, namely the Bayesian Information Criterion (BIC) and we compared its estimate across different models (fixed-effect comparison). Additionally, we used random-effects procedure to perform Bayesian model selectin at group level where we computed an approximation of model evidence as –BIC/2. Model comparison showed that gkRL model was better able to explain participants’ choice behavior compared to a standard RL (xp_gkRL_=1, BIC_gkRL_=14354; xp_standardRL_ = 0, BIC_standardRL_ = 16990).

Overall, these results suggest that our participants make use of information when making a choice.

As last note, we have now added a parameter recovery analysis (pg. 37) based on Willson and Collins. *eLife*. 2019. In particular, we simulated data from gkRL using the parameters obtained from the fitting procedure (*true parameters*), and we fit the model to those simulated data to obtain the estimated parameters (*fit parameters*). We then ran a correlation for each pair of parameters. This revealed high correlation coefficients for α (r = 0.8, *p* < 10^-3^), β (r = 0.8, *p* < 10^-3^), omega (r = 0.6, *p* = 0.006) and γ (r = 0.6, *p* = 0.002).

5. Model comparison: In light of the previous comment, I wonder whether the gkRL model is best suited to explain behavior in this task, and, thus, whether it serves as a proper foundation for the intended fMRI analysis. As the manuscript notes in lines 408-409, the "model was already validated for this task and it was better able to explain participants' behavior compared to other RL models". However, this evaluation was conducted in a different study with (presumably) a different sample of participants. The manuscript should nevertheless provide a convincing argument for deploying this model to this study, as well as for using corresponding regressors in the fMRI analysis. This could be achieved by demonstrating that the model does a better job in explaining behavior compared to a standard RL model (without the information term) ‘for each of these participants’.

We thank the reviewer for this comment. The reviewer raises an important point. We have now included model comparison in our analyses (please see our previous comment).

[Editors’ note: what follows is the authors’ response to the second round of review.]

The reviewers felt that the revised manuscript has been improved. Indeed, in some cases, the reviewers felt that revisions were extensive and commendable; for example, in addressing whether the information value is instrumental or non-instrumental.However, there were several other fundamental points that were missed in the revision. The reviewers and reviewing editor discussed these points at length, and it was decided that most could be addressed in some form but simply were not. These include issues related to the placement of these results in the broader literature and characterization of previous studies, the definition of relative reward, the incorporation of the omega parameter in the information term, matching of covariates in the GLM, and a number of gaps in the added analyses, such as the switching versus default behavior. It was noted that some other issues like the orthogonalization approach or the confounding of the information term with other variables were also not addressed, and there was some discussion of whether these are addressable. Nonetheless, these outstanding issues were all viewed as important.Thus, the consensus was to send this back for another 1revision. However, we must emphasize that if these points are not addressed in revision, the manuscript may not be considered acceptable for publication at eLife. I have included the full and detailed reviews from each reviewer in this letter so you can address them directly in your revision.

We thank the editor for raising these issues which have helped strengthen the paper. We believe we have addressed them all in the revised manuscript.

1) The placement of these results in the broader literature and characterization of previous studies.

We have extensively revised introduction and discussion to better place the current findings within the broader literature.

In the introduction, we have included several findings which suggest a shared neural activity between reward and information signals in the human PFC (e.g., Jepma 2012; Charpentier 2018; Kobayashi 2019; Iigaya, 2020; Kaanders, 2020; Trudel 2021). In particular, recent advances in the neuroscience of information-seeking all seem to converge on a key principle: information is valuable, similar to primary or monetary rewards. Humans and other animals are oftentimes willing to incur a cost for obtaining information (Pierson and Goodman, 2014) (Blanchard, Hayden, and Bromberg-Martin, 2015; Charpentier, Bromberg-Martin, and Sharot, 2018), even if the information provides no or few instrumental benefits. This is reflected in the brain where the opportunity to gain information relies on similar neural circuitry as the opportunity to gain rewards (Bromberg-Martin and Hikosaka, 2009; Kang et al., 2009) (Blanchard et al., 2015; Bromberg-Martin and Hikosaka, 2011; Charpentier et al., 2018; Gruber, Gelman, and Ranganath, 2014; Jessup and O'Doherty, 2014; Kobayashi and Hsu, 2019; Smith, Rigney, and Delgado, 2016; Tricomi and Fiez, 2012). However, reward and information often explain similar variance. Information signals are partly characterized by reward-related attributes such as valence and instrumentality (Sharot and Sunstein, 2020), while reward signals also contain informative attributes (e.g., winning $50 on a lottery allows the recipient to gain the reward amount but also information about the lottery itself; Wilson et al. 2014). Because of this “shared variance” it is not surprising that the neural substrates underlying reward value processing frequently overlap with those involved in optimizing information. This raises an interesting question as to whether information and reward are really two distinct signals. In other words, is information merely a kind of reward that is processed in the same fashion as more typical rewards, or is the calculation of information value independent of reward value computations? We address this issue by using computational modeling, model-based fMRI analyses, and a novel experimental paradigm. Our results show that information is not processed in the same fashion as reward. Rather when we took into account the shared variance between reward and information, we observed a dedicated and independent value system for information in PFC. In particular, we showed that information value calculation occurred within dACC, but not in vmPFC, while reward value computation occurred within vmPFC, but not in dACC. Our control analysis additionally shows that dACC encodes both the non-instrumental and the instrumental value of information. However, the instrumental value of information also shows activity in reward regions in line with previous work on common neural code between reward and information (Kobayashi et al. 2019). The fact that neural substrates underlying information processing do not always overlap with those involved in optimizing reward suggests that, in fact, information and reward are not the same signals. In other words, information is valuable on its own, independently of its rewarding attributes. This is the first study that shows that alongside the often observed reliance of information on similar neural circuitries as those of reward, information is also processed independently of it. Understanding how the brain implements information-seeking is still in its early stages, and the results of our study can definitely help to advance the field. These findings supports theoretical accounts such as active inference (Friston, 2003, 2005, 2010) and certain RL models (e.g., upper confidence bound Auer, Cesa-Bianchi, and Fischer, 2002; Cogliati Dezza, Yu, Cleeremans, and Alexander, 2017; Wilson, Geana, White, Ludvig, and Cohen, 2014) which predict independent computations for information value (epistemic value) and reward value (extrinsic value) in the human brain. Moreover, our results are in line with recent findings in monkey literature that identified populations of neurons in ACC which selectively encodes the non-instrumental value of information (White et al., 2019) and involved in tracking how each piece of information would reduce uncertainty about future actions (Hunt et al., 2018). Additionally, our results are also consistent with computational models of PFC which predict that dACC activity can be primarily explained as indexing prospective information about an option independent of reward value (Alexander and Brown, 2011, 2018; Behrens, Woolrich, Walton, and Rushworth, 2007).

2) The definition of relative reward.

We have included now additional control analyses to test whether dACC (which correlates with information signals but not with reward signals when considering the shared variance between the two) shows information-related signals when other computations of reward are considered. We would like to stress that our task was not implemented to test all the possible reward computations which may take place in the brain, but rather to differentiate information signals from reward signals which are commonly observed in the regions implicated in processing both kinds of value. We note that in the studies discussed in our review of the literature it is largely unheard of that all possible definitions for a signal that could explain activity in a region are included in a single GLM. Instead, it is typical that multiple GLMs are created to answer specific questions about the nature of a signal observed. For example, in Shenhav et al., 2016, the authors constructed 8 GLMs to investigate the roles of dACC and vmPFC in foraging. Each GLM used a constrained set of regressors based on testing different calculations of relative value, and in only one case were additional definitions of value (search value and engage value) included. Absent from these analyses are several of the alternative definitions the reviewers have requested of us – min, max, expected value, average, 2^nd^ best, and choice entropy. While we agree with the reviewers that identifying the precise definition of reward that best describes brain activity is an interesting question, it seems unreasonable to hold our approach to a standard that is not observed by the field. Furthermore, adding arbitrarily many regressors increases the risk of committing a type II error; even if additional regressors are only marginally correlated with our regressor of interest, it could be the case that some combination of these additional regressors are collinear with the regressor of interest, thus potentially hiding a real effect.

Nevertheless, we have revised our analyses to include the following definitions of reward in a single GLM: min value of an option, the expected reward, averaged value, relative second (second best value -mean(value of the other two options)). Altogether these analyses suggest that even when controlling for several alternative reward computations, dACC still encodes information signals rather than reward signals. Details of this analysis, and the procedure for selecting reward definitions, are discussed in point 4 of this rebuttal letter.

3) The incorporation of the omega parameter in the information term.

We acknowledge we did not explain well-enough this point in the previous version of the manuscript. The parameter omega constitutes a ‘mixing’ parameter governing the relative importance of information value and reward value in generating behavior. The gkrl model could easily be rewritten with the omega modifying the reward term instead of the information term. Omega therefore is not part of the calculation of subjective information value itself, but only applies after subjective information value has been calculated to determine its influence on behavior when considered alongside reward values.

Nevertheless, we re-ran our analysis (GLM2) using the informationGain*omega regressor suggested by the reviewer. The inclusion of a constant multiplier on a regressor has no effect on the significance of that regressor’s β estimates. Significantly active voxels for our 1^st^ level analyses are identical whether the regressor is entered as only the InformationGain, or as InformationGain*Omega. However, because the regressor values were not standardized, the magnitude of the β estimates depended on omega, i.e., the β estimate of a voxel in the InformationGain*Omega analysis was simply the β estimate obtained in the InformationGain analysis multiplied by omega. Again, this has no effect on the significance of the 1^st^ level analysis since changing the scale of a regressor does not change its relationship with the dependent variable (e.g., age is still significantly related to the passage of time regardless whether time is measured in years, days, or hours). However, the scale of betas from the parameter omega affects significance at 2^nd^ level.

Indeed, 2^nd^ level analysis of our data using the InformationGain*omega regressor (again, values were not standardized) produced no significant activity in dACC. This is not necessarily surprising, since omega reflects the degree to which subjective information is used to inform behavior in conjunction with reward value, and not the calculation of subjective information itself. Another way of saying this is that omega only has a meaningful interpretation in the context of a comparison with reward. Yet another way of saying this is that omega indirectly reflects reward value, and since our manuscript is premised on dissociating reward from information value, we would not necessarily expect to observe reward-related activity in a region associated with information value.

To summarize:

1) The omega parameter in the gkrl model is not related to the calculation of subjective information value, but reflects the weighting of subjective information relative to reward value.

2) Regressing informationGain*omega against the BOLD signal does not influence significance at the 1^st^ level (see Author response image 1).

**Author response image 1. sa2fig1:** 

3) At the 2^nd^ level, no activity in ACC is observed, but we never hypothesized that ACC activity reflects the comparison of reward to information which is implicitly coded by the omega parameter.

In Author response image 1, we report 1^st^ level tmaps from Information Gain and Information Gain*omega from a randomly chosen subject. The figures shows only one color as activity for Information Gain and Information Gain*w perfectly overlap at 1^st^ level.Considering that we did not observe any activity related to Information Gain*w we did not investigate this any further. We have however added additional information in the Method section regarding the role of omega. The new text reads as follows: “The parameter omega constitutes a ‘mixing’ parameter governing the relative importance of information value and reward value in generating behavior.” “To note, the parameter omega is not added to the computation as it describes how much information is relevant respect to reward, rather the information value on itself.”

*4) Matching of covariates in the GLM*. We have now added results for GLM3 and 4 with covariates as described below.

Different reward definitions in dACC

The reviewers asked us to investigate different definitions of reward, and helpfully suggested several, including average reward, minimum reward, maximum reward, second-best reward, and the relative value of the second option minus the average of the other two. In our previous revision, we tested some of these definitions directly against RelativeReward, a definition of reward that has previously been linked to vmPFC function. Subsequently, the reviewers questioned why we did not include all definitions of reward in the same GLM.

Different ways of defining the same underlying concept (reward, in this case) may result in correlated values, potentially introducing problems in estimating a GLM (i.e., multicollinearity). In order to identify reward definitions that may be collinear, we calculated Variance Inflation Factors (Craney and Surles, 2002) for the suggested definitions of reward for each subject, and removed regressors with VIFs (averaged across subjects) above a threshold of 5 (interpreted as 80% of the variability in the regressor can be explained by the rest of the regressors).

We immediately experienced problems with this approach because, due to the high degree of collinearity, each definition of reward suggested by the reviewers (with the exception of the second-most-valuable reward) was almost perfectly predicted by the other definitions of reward, i.e., the VIF was infinite, or at least very, very large (keep in mind that a VIF of 5 is considered to be ‘high’). Rather than removing definitions at random, we introduced a small amount of normally-distributed random noise (mean = 0, σ = 0.01) to the regressor values and conducted our analysis again. Our intent in introducing noise was to reduce correlations amongst regressors sufficiently to make ordinal judgments about their VIFs.

Using this approach, we removed variables with the highest VIFS iteratively. In order, these were: the value of the chosen reward minus the second highest reward (VIF ≈ 2 X 10^5^, 99.9995% of variability explained by other regressors), and the maximum reward (VIF ≈ 320, 99.69% of variability explained by other regressors, after removing ChosenMinusSecond). After removing these two regressors, VIFs for the remaining regressors were all under 5 (average value VIF = 3.52, value of the second-best option VIF = 1.99, relative value VIF = 2.21, and minimum value VIF = 3.42).

We note that the AverageReward regressor had the highest VIF initially (VIF ≈ 3.6x 10^5^). However, we elected to retain this definition since behavior and brain activity have previously been linked to the overall level of reward across options (Cogliati Dezza et al., 2017; Kolling et al., 2012).

Our finding of very high VIFs for different definitions of reward suggests that, in our design, we are unable to dissociate reward definitions – the magnitude of the VIFs indicates that some of these regressors, being highly correlated, are essentially indistinguishable. It should be noted that differentiating amongst definitions of rewards was NOT a primary goal of this study. On a positive note, this analysis suggests that our conclusions regarding the dissociation of informationValue and rewardValue are not greatly impacted by the specific definition of reward; effects associated with a regressor with an extremely high VIF (i.e., one that shares practically all its variance with one or more other regressors) would be captured by the remaining, collinear regressors following its removal.

We have explained these findings in our revised manuscript on pgs 9-10. The new text reads as follow:

*“*For GLM4, similar results as those reported above were observed when ExpReward was entered in GLM4 instead of RelReward (GLM4bis: t(19) = 4.4, p < 10-3) and when accounting for covariates for reward i.e., Average Value, the value of the best second option- mean (value chosen, value third option) and Min Value. The rationale behind choosing these reward covariates was as follows: we calculated Variance Inflation Factors 36 for the possible definitions of reward (suggested by reviewers) for each subject, and removed regressors with VIFs (averaged across subjects) above a threshold of 5 (interpreted as 80% of the variability in the regressor can be explained by the rest of the regressors). Due to high multicollinearity amongst the possible reward definitions, VIFs for each definition were indistinguishable (≈ infinity). Rather than removing definitions at random, we introduced a small amount of normally-distributed random noise (mean = 0, σ = 0.01) to the regressor values and conducted our analysis again. Our intent in introducing noise was to reduce correlations amongst regressors sufficiently to make ordinal judgments about their VIFs.

Using this approach, we removed variables with the highest VIFS iteratively. In order, these were: the value of the chosen reward minus the second highest reward (VIF ≈ 2 X 105, 99.9995% of variability explained by other regressors), and the maximum reward (VIF ≈ 320, 99.69% of variability explained by other regressors, after removing Chosen-Second). After removing these two regressors, VIFs for the remaining regressors were all under 5 (average value VIF = 3.52, value of the second-best option VIF = 1.99, relative value VIF = 2.21, and minimum value VIF = 3.42). We note that the AverageReward regressor had the highest VIF initially (VIF ≈ 3.6x 105). However, we elected to retain this definition since behavior and brain activity have previously been linked to the overall level of reward across options 13,32. Next, we ran a ROI analysis based on dACC and vmPFC coordinates observed in GLM1. Results showed significant activity in dACC which positively correlates with Information Gain (t(19) = 2.73, p = 0.013), while we found no correlated activity in vmPFC as was observed in GLM2 (t(19) = – 1.5, p = 0.1503). ”

Covariates for GLM3

The new text reads as follow:

“For GLM3, similar results as those reported in the previous section were observed when accounting for covariates and an alternative definition of information. In particular, we entered the relative value of information (as it is possible that vmPFC computes the “relativeness” of the chosen options rather than its reward value) and first choice reaction time as covariates (GLM3bis). A ROI analysis based on dACC and vmPFC coordinates observed in GLM2 showed that activity in vmPFC remained positively correlated with RelReward (t(19) = 4.8, p < 0.001) after controlling for Information Gain. In contrast, no significant cluster was observed after removing variance associated with Information Gain (t(19) = -0.68, p = 0.505). ”

*5) Switching versus default behavior*. To answer reviewer’s concern on computation of switching strategy in dACC we have included an additional regressor in GLM4 accounting for this variance. The new text reads as follow:

“We then entered choice reaction time as additional regressor in GLM4 alongside the across-option standard deviation (i.e., the standard deviation of expected reward values of the 3 options at time of the first free choice) as an additional proxy for choice difficulty. Moreover, in the same GLM we entered a switch-stay regressor as proxy for switching behavior (i.e., coded 0 if choices on the first free choice trial where the same as previous trial choices and 1 otherwise). We then used vmPFC and dACC ROIs from GLM1 to analyze the data. Results essentially replicate the above findings with significant activity in dACC which positively correlated with Information Gain (t(19) = 2.4, p = 0.027), while we found no significant activity in vmPFC (t(19) = – 0.757, p = 0.544).” We have, however, kept the text describing default vs. switching behavior in our task as we think it gives additional information on this matter and it was also well appreciated by reviewer 3.

*6) Orthogonalization approach.* Reviewer 2 is correct and we acknowledge that we did not explain this point well enough in the previous version. We have made modifications to the Introduction to better explain our claim which we summarize below.

Reward and information often explain similar variance. Information signals are partly characterized by reward-related attributes such as valence and instrumentality (Sharot and Sunstein, 2020), while reward signals also contain informative attributes (e.g., winning $50 on a lottery allows the recipient to gain the reward amount but also information about the lottery itself; Wilson et al. 2014). Because of this “shared variance” it is not surprising that the neural substrates underlying reward value processing frequently overlap with those involved in optimizing information (Bromberg-Martin and Hikosaka, 2009; Kang et al., 2009; Blanchard et al., 2015; Bromberg-Martin and Hikosaka, 2011; Charpentier et al., 2018; Gruber et al., 2014; Jessup and O'Doherty, 2014; Kobayashi and Hsu, 2019; Smith et al., 2016; Tricomi and Fiez, 2012). This raises an interesting question as to whether information and reward are really two distinct signals. Simulations of our RL model which consists of independent value systems independently optimizing information and reward, however, suggest that fMRI analyses might reveal overlapping activity if the “shared variance” between reward and information is not taken into account.

This raises an interesting question as to whether information and reward are really two distinct signals. In other words, is information merely a kind of reward that is processed in the same fashion as more typical rewards, or is the calculation of information value independent of reward value computations? We believe that our methodology can answer this question as by applying orth function in SPM we can assign “*all shared variance to the un-orthogonalised term*” (from Reviewer 2’s comment1). Therefore, in GLM3 and 4 whatever it is left from the analysis is the un-shared variance between reward and information. This approach allows us to individuate specific regions in the brain which encode reward and information independently.

*7) Confounding of the information term.* We believe that our additional analyses (in particular point 4 and 5 above) address this issue.

Reviewer #1:The revised manuscript addresses some concerns raised by the reviews, by adding separate analyses that examine the effect of absolute versus relative reward and information gain, an analysis of parameter recovery and a formal comparison to a standard RL model. However, the revised manuscript still misses to address a number of critical issues:1. Novelty: The revised manuscript makes an attempt of distinguishing its claims from prior work. However, the reader may still be confused about how the authors situate their findings in contrast to prior work. The manuscript seeks to promote an "independent value system for information in human PFC", in contrast to single value system view that "vmPFC positively and dACC are negatively contributing to the net-value computation". The authors state that "[e]ven for studies in which dACC and vmPFC activity is dissociated activity in vmPFC is generally linked to reward value, while activity in dACC is often interpreted as indexing negative or non-rewarding attributes of a choice (including ambiguity, difficulty, negative reward value, cost and effort)". This statement requires further elaboration. Why can a non-rewarding attribute not be considered independent of reward, thus promoting a dual-systems view? That is, the reader might wonder: are the non-rewarding attributes of choice in the cited studies necessarily in opposition to reward, contributing to the same "net-value computation" and suggesting a symmetrical opposition? For some of these attributes, this is not necessarily the case (see Comment 2 for a specific example).

We agree with the reviewer that the previous frame of the manuscript may have failed to adequately highlight the novelty and importance of our findings. We have now extensively revised the frame of the manuscript to do so (see Introduction). In particular, recent advances in the neuroscience of information-seeking all seem to converge on a key principle: information is valuable, similar to primary or monetary rewards. Humans and other animals are oftentimes willing to incur a cost for obtaining information (Pierson and Goodman, 2014; Blanchard et al., 2015; Charpentier et al., 2018), even if the information provides no or few instrumental benefits. This is reflected in the brain where the opportunity to gain information relies on similar neural circuitry as the opportunity to gain rewards (Bromberg-Martin and Hikosaka, 2009; Kang et al., 2009; Blanchard et al., 2015; Bromberg-Martin and Hikosaka, 2011; Charpentier et al., 2018; Gruber, Gelman, and Ranganath, 2014; Jessup and O'Doherty, 2014; Kobayashi and Hsu, 2019; Smith, Rigney, and Delgado, 2016; Tricomi and Fiez, 2012).

However, reward and information often explain similar variance. Information signals are partly characterized by reward-related attributes such as valence and instrumentality (Sharot and Sunstein, 2020), while reward signals also contain informative attributes (e.g., winning $50 on a lottery allows the recipient to gain the reward amount but also provides information about the lottery itself; Wilson et al. 2014). Because of this “shared variance” it is not surprising that the neural substrates underlying reward value processing frequently overlap with those involved in optimizing information. This raises an interesting question as to whether information and reward are really two distinct signals. In other words, is information merely a kind of reward that is processed in the same fashion as more typical rewards, or is the calculation of information value independent of reward value computations? We believe this is an important and novel question that has yet to be answered.

We address this issue by using computational modeling, model-based fMRI analyses, and a novel experimental paradigm. Our results show that information is not processed in the same fashion as reward. Rather when we took into account the shared variance between reward and information, we observed a dedicated and independent value system for information in PFC. In particular, we showed that information value calculations occur within dACC, but not in vmPFC, while reward value computations occur within vmPFC, but not in dACC. Our control analysis additionally shows that dACC encodes both the non-instrumental and the instrumental value of information. However, the instrumental value of information also shows activity in reward regions in line with previous work on common neural code between reward and information (Kobayashi et al. 2019).

The fact that neural substrates underlying information processing do not always overlap with those involved in optimizing reward suggests that, in fact, information and reward are not the same signals. In other words, information is valuable on its own, independently of its rewarding attributes. This is the first study that shows that, alongside the often observed reliance of information on similar neural circuits as those of reward, information is also processed independently of it. Understanding how the brain implements information-seeking is still in its early stages, and the results of our study can definitely help to advance the field.

2. Mischaracterization of literature: The revised manuscript continues to mischaracterize the existing literature, suggesting that the studies of Shenhav et al. interpret ACC and vmPFC as participating in a single reward system-although the revision appears to partly avoid this characterization by dropping the Shenhav et al. (2018) reference. Regardless, there are several reasons for why one should not interpret these studies as advocating for and/or interpreting their findings as supportive of a single-value system:– As the authors of this manuscript state in their response, the Expected Value of Control (EVC; 2013) framework suggests that certain forms of conflict or uncertainty may be considered when computing EVC. However, the 2016 and 2018 studies do consider forms of conflict and uncertainty that are not explicitly taken into account when computing the expected value of a control signal. For instance, in the 2018 study, EVC is expressed a function of reward, not uncertainty or conflict (Equation 3):EV_t = [Pr(Mcorrect) * Reward(MCorrect)] + [Pr(Ccorrect) * Reward(CCorrect)]– Irrespective of the way in which the EVC is computed, EVC theory neither suggests that dACC activity is reflective of a single value, nor does it explicitly propose that dACC activity correlates with EVC. Shenhav et al. (2013, 2016) communicate this in their work.– It is worth noting that Shenhav et al. (2014, 2018b, 2019; see references below) make the case that dACC vs. vmPFC are engaged in different kinds of valuation (with dACC being more tied to uncertainty and vmPFC being more tied to reward). The authors may want to consider this work when situating their study in the existing literature (see Comment 1):Shenhav, A. and Buckner, R.L. (2014). Neural correlates of dueling affective reactions to win-win choices. Proceedings of the National Academy of Sciences 111(30): 10978-10983Shenhav, A., Dean Wolf, C.K., and Karmarkar, U.R. (2018). The evil of banality: When choosing between the mundane feels like choosing between the worst. Journal of Experimental Psychology: General 147(12): 1892-1904.Shenhav, A. and Karmarkar, U.R. (2019). Dissociable components of the reward circuit are involved in appraisal versus choice. Scientific Reports 9(1958): 1-12.– Finally, the statements made in Shenhav et al. do simply not express a single-value system view, as erroneously suggested by the authors of this manuscript. For convenience of the authors, reviewers and editors, I attached excerpts from the Discussion sections of the relevant 2016 and 2018 articles below.That is, the manuscript continues to mischaracterize these studies when stating that they interpret ACC and vmPFC as a single value reward system. I respectfully ask the authors to consider these points in their current and future work.

We have now re-written the introduction to avoid the perceived mischaracterization of the literature highlighted above by the reviewer (see Introduction). In particular, we do not refer to single or dual value system in the current version of the manuscript. Rather we frame our results as shared vs. independent neural networks involved in reward and information processing. We have better explained this point in our answer to comment 1.

3. Model fits: I commend the authors for including an analysis of parameter recoverability as well as a model comparison with a standard RL model. However, the model comparison should indicate which trials were included in the analysis (all first free choices, as used for parameter estimation?). As indicated in my previous review, the analysis should also report the outcome of this model comparison, separately for each participant (i.e. a table with BIC values for gkRL and standardRL, for each participant), so that the reader can evaluate whether gkRL is generally better (not just on average).

Both parameter recovery procedure and model comparison include first free choices only. The new text reads as follows: “As for the fitting procedure, the recovery procedure was run on first free choice trials.” “The fitting procedure was again run on first free choice trials.” Moreover, we have now a table (Supplementary File 2) with individual BIC values for both models.

4. Subjective information gain: I apologize if I missed this, but the authors' response doesn't seem to address the following issue raised earlier: The analysis refrains from taking into account the *subjective* information gain that is actually factored into the model's choice. As re-iterated by the editor, I was a bit surprised that the GLM implements – I_{t,j}(c) as regressor for information gain, instead of – omega * I_{t,j}(c). If omega is small, then the former regressor may not be a good proxy for information gain that is actually used to guide decision-making. This issue was implicitly raised in comment 8 of Reviewer 2, to which the authors replied that "all the neural activity relates to subjects' behavior". The claimed brain-behavior relationship would be more internally valid if the participant-specific parameter omega is considered in the analysis.

First, we note that subjective information in the gkrl model is calculated as: It, j(c)=(∑1tit,j(c))γ Here the parameter γ governs the shape of the information value function: γ values lower than 1 indicate a concave infoValue function, while values larger than 1 indicate a convex value function. Therefore, the information gain regressor (like the reward regressor) includes free parameters fitted to individual behavior. This is why our previous analyses already account for individual differences and can already link neural activity to behavior. We have added new text (pg. 19) to better explain the role of γ: “In other words, g governs the shape of the information value function: values lower than 1 indicate a concave value function, while values larger than 1 indicate a convex value function.”

Secondly, we acknowledge we did not explain well-enough the role of the omega parameter in the previous version of the manuscript. The parameter omega constitutes a ‘mixing’ parameter governing the relative importance of information value and reward value in generating behavior. The gkrl model could easily be rewritten with the omega modifying the reward term instead of the information term. Omega therefore is not part of the calculation of subjective information value itself, but only applies after subjective information value has been calculated to determine its influence on behavior when considered alongside reward values.

Nevertheless, we re-ran our analysis (GLM2) using the informationGain*omega regressor suggested by the reviewer. The inclusion of a constant multiplier on a regressor has no effect on the significance of that regressor’s β estimates. Significantly active voxels for our 1^st^ level analyses are identical whether the regressor is entered as only the InformationGain, or as InformationGain*Omega. However, because the regressor values were not standardized, the magnitude of the β estimates depended on omega, i.e., the β estimate of a voxel in the InformationGain*Omega analysis was simply the β estimate obtained in the InformationGain analysis multiplied by omega. Again, this has no effect on the significance of the 1^st^ level analysis since changing the scale of a regressor does not change its relationship with the dependent variable (e.g., age is still significantly related to the passage of time regardless whether time is measured in years, days, or hours). However, the scale of betas from the parameter omega affects significance at 2^nd^ level.

Indeed, 2^nd^ level analysis of our data using the InformationGain*omega regressor (again, values were not standardized) produced no significant activity in dACC. This is not necessarily surprising, since omega reflects the degree to which subjective information is used to inform behavior in conjunction with reward value, and not the calculation of subjective information itself. Another way of saying this is that omega only has a meaningful interpretation in the context of a comparison with reward. Yet another way of saying this is that omega indirectly reflects reward value, and since our manuscript is premised on dissociating reward from information value, we would not necessarily expect to observe reward-related activity in a region associated with information value.

To summarize:

1) The omega parameter in the gkrl model is not related to the calculation of subjective information value, but reflects the weighting of subjective information relative to reward value.

2) Regressing informationGain*omega against the BOLD signal does not influence significance at the 1^st^ level.

3) At the 2^nd^ level, no activity in ACC is observed, but we never hypothesized that ACC activity reflects the comparison of reward to information which is implicitly coded by the omega parameter.

In Author response image 1, we report 1^st^ level tmaps from Information Gain and Information Gain*omega from a randomly chosen subject. The image shows only one color as activity for Information Gain and Information Gain*w perfectly overlap at 1^st^ level.

Considering that we did not observe any activity related to Information Gain*w we did not investigate this any further. We have however added additional information in the Method section regarding the role of omega. The new text reads as follows: “The parameter omega constitutes a ‘mixing’ parameter governing the relative importance of information value and reward value in generating behavior.” “To note, the parameter omega is not added to the computation as it describes how much information is relevant respect to reward, rather the information value on itself.”

5. Consideration of Covariates: The authors state that "Standard Deviation and Chosen-Second were highly correlated with RelReward" and "[f]or this reason, these covariates were not considered any further". Does this mean that these covariates are confounded with relative reward, and that the authors cannot dissociate whether vmPFC encodes Standard Deviation (as the author's proxy for decision entropy) or Chosen-Second?

Firstly, we have updated the computation of decision entropy in the current version. Secondly, our results suggest that vmPFC encodes reward – whichever computation we choose – and among those computations relative reward value seems to better explain its activity. This was already suggested by Shenhav et al. 2014. Here, we are not making any claim on the specific reward computation encoded by vmPFC- which we believe is outside the scope of this manuscript. Since our task was designed to distinguish previously reported reward signals from information value, we did not feel that it was necessary to control for other definitions of reward that are *highly correlated* with our preferred definition: The standard deviation of reward values is greatest when there is a large difference between one option and the others, i.e., when the relative reward is high, and Chosen-Second – which is the expected reward value of the option minus the second best reward value – follows the same logic. Given the degree to which these three variables are correlated, it is unclear how our primary goal of distinguishing reward and information related signals would benefit from an experimental design that also dissociates these reward definitions.

Also, why aren't all covariates included when examining the effect of information gain (both relative and absolute) on both vmPFC and dACC activity, as well as the effect of expected reward? It would seem appropriate to include at least non-correlating covariates in all analyses to allow for a proper comparison across GLMs.

We have now collapsed some of the control analyses in the same glms and included alternative covariates suggested by the reviewer.

In particular, one concern raised in previous review rounds is that dACC may correlate with information value when controlling for relative reward, but it may not show activity when controlling for other reward computations. The reviewers asked us to investigate different definitions of reward, and helpfully suggested several, including average reward, minimum reward, maximum reward, second-best reward, and the relative value of the second option minus the average of the other two.

In previous version of this manuscript we showed that adding expected reward instead of relative reward did not change the results – activity in dACC associated with information value. Subsequently, the reviewers questioned why we did not include all definitions of reward in the same GLM.

Different ways of defining the same underlying concept (reward, in this case), however, may result in correlated values, potentially introducing problems in estimating a GLM (i.e., multicollinearity). In order to identify reward definitions that may be collinear, we calculated Variance Inflation Factors (Craney and Surles, 2002) for the suggested definitions of reward for each subject, and removed regressors with VIFs (averaged across subjects) above a threshold of 5 (interpreted as 80% of the variability in the regressor can be explained by the rest of the regressors).

We immediately experienced problems with this approach because, due to the high degree of collinearity, each definition of reward suggested by the reviewers (with the exception of the second-most-valuable reward) was almost perfectly predicted by the other definitions of reward, i.e., the VIF was infinite, or at least very, very large (keep in mind that a VIF of 5 is considered to be ‘high’). Rather than removing definitions at random, we introduced a small amount of normally-distributed random noise (mean = 0, σ = 0.01) to the regressor values and conducted our analysis again. Our intent in introducing noise was to reduce correlations amongst regressors sufficiently to make ordinal judgments about their VIFs.

Using this approach, we removed variables with the highest VIFS iteratively. In order, these were: the value of the chosen reward minus the second highest reward (VIF ≈ 2 X 10^5^, 99.9995% of variability explained by other regressors), and the maximum reward (VIF ≈ 320, 99.69% of variability explained by other regressors, after removing ChosenMinusSecond). After removing these two regressors, VIFs for the remaining regressors were all under 5 (average value VIF = 3.52, value of the second-best option VIF = 1.99, relative value VIF = 2.21, and minimum value VIF = 3.42).

We note that the AverageReward regressor had the highest VIF initially (VIF ≈ 3.6x 10^5^). However, we elected to retain this definition since behavior and brain activity have previously been linked to the overall level of reward across options (Cogliati Dezza et al., 2017, Kolling et al., 2012).

Our finding of very high VIFs for different definitions of reward suggests that, in our design, we are unable to dissociate reward definitions – the magnitude of the VIFs indicates that some of these regressors, being highly correlated, are essentially indistinguishable. It should be noted that differentiating amongst definitions of rewards was NOT a primary goal of this study. On a positive note, this analysis suggests that our conclusions regarding the dissociation of informationValue and rewardValue are not greatly impacted by the specific definition of reward (see new text below); effects associated with a regressor with an extremely high VIF (i.e., one that shares practically all its variance with one or more other regressors) would be captured by the remaining, collinear regressors following its removal.

The new text reads as follow:

*“*For GLM4, similar results as those reported above were observed when ExpReward was entered in GLM4 instead of RelReward (GLM4bis: t(19) = 4.4, p < 10-3) and when accounting for covariates for reward i.e., Average Value, the value of the best second option- mean (value chosen, value third option) and Min Value. The rationale behind choosing these reward covariates was as follows: we calculated Variance Inflation Factors 36 for the possible definitions of reward (suggested by reviewers) for each subject, and removed regressors with VIFs (averaged across subjects) above a threshold of 5 (interpreted as 80% of the variability in the regressor can be explained by the rest of the regressors). Due to high multicollinearity amongst the possible reward definitions, VIFs for each definition were indistinguishable (≈ infinity). Rather than removing definitions at random, we introduced a small amount of normally-distributed random noise (mean = 0, σ = 0.01) to the regressor values and conducted our analysis again. Our intent in introducing noise was to reduce correlations amongst regressors sufficiently to make ordinal judgments about their VIFs.

Using this approach, we removed variables with the highest VIFS iteratively. In order, these were: the value of the chosen reward minus the second highest reward (VIF ≈ 2 X 105, 99.9995% of variability explained by other regressors), and the maximum reward (VIF ≈ 320, 99.69% of variability explained by other regressors, after removing Chosen-Second). After removing these two regressors, VIFs for the remaining regressors were all under 5 (average value VIF = 3.52, value of the second-best option VIF = 1.99, relative value VIF = 2.21, and minimum value VIF = 3.42). We note that the AverageReward regressor had the highest VIF initially (VIF ≈ 3.6x 105). However, we elected to retain this definition since behavior and brain activity have previously been linked to the overall level of reward across options 13,32. Next, we ran a ROI analysis based on dACC and vmPFC coordinates observed in GLM1. Results showed significant activity in dACC which positively correlates with Information Gain (t(19) = 2.73, p = 0.013), while we found no correlated activity in vmPFC as was observed in GLM2 (t(19) = – 1.5, p = 0.1503). ”

Furthermore, it would seem reasonable to include RT as a covariate despite it not being correlated with information gain. First, information gain is not the only factor being considered. Second, a lack of statistical correlation with information gain doesn't imply that RT cannot explain additional variance when included in the GLM.

As suggested by the reviewer we have included reaction time in both GLM3 and 4 alongside additional covariates.

In particular, in GLM3 we have included reaction time and Relative Information (as it is possible that activity observed in vmPFC after accounting for information value may relate to the “relativeness” of the chosen option rather to its reward computation). The new section reads as follows. “For GLM3, similar results as those reported in the previous section were observed when accounting for covariates and an alternative definition of information. In particular, we entered the relative value of information (as it is possible that vmPFC computes the “relativeness” of the chosen options rather than its reward value) and first choice reaction time as covariates (GLM3bis). A ROI analysis based on dACC and vmPFC coordinates observed in GLM2 showed that activity in vmPFC remained positively correlated with RelReward (t(19) = 4.8, p < 0.001) after controlling for Information Gain. In contrast, no significant cluster was observed after removing variance associated with Information Gain (t(19) = -0.68, p = 0.505). ”

In GLM4 we have included reaction time alongside regressors accounting for choice difficulty and switch-stay behavior. The new text reads as follows “We then entered choice reaction time as additional regression in GLM4 alongside a proxy for across-option standard deviation (i.e., the standard deviation of expected reward values of the 3 options at time of the first free choice) as an additional proxy for choice difficulty. Moreover, in the same GLM we entered a switch-stay regressor as proxy for switching behavior (i.e., coded 0 if choices on the first free choice trial where the same as previous trial choices and 1 otherwise). We then used vmPFC and dACC ROIs from GLM1 to analyze the data. Results essentially replicate the above findings with significant activity in dACC which positively correlated with Information Gain (t(19) = 2.4, p = 0.027), while we found no significant activity in vmPFC (t(19) = – 0.757, p = 0.544).”

Excerpts of Discussion from Shenhav, A., Straccia, M. A., Botvinick, M. M., and Cohen, J. D. (2016). Dorsal anterior cingulate and ventromedial prefrontal cortex have inverse roles in both foraging and economic choice. Cognitive, Affective, and Behavioral Neuroscience, 16(6), 1127-1139:Excerpts of Discussion from Shenhav, A., Straccia, M. A., Musslick, S., Cohen, J. D., and Botvinick, M. M. (2018). Dissociable neural mechanisms track evidence accumulation for selection of attention versus action. Nature communications, 9:

We thank the reviewer to have added these excerpts to better strengthen her/his point. As reported in comment 1 and 2, extensive changes have been made to the manuscript which we believe should address those concerns. We thank the reviewer for the useful comments which have helped us to strength the manuscript.

Reviewer #2:The authors propose that there are "dedicated" and "independent" reward value and information value systems driving choice in the medial frontal cortex, respectively situated in vmPFC and dACC. This is a major claim that would be highly significant in understanding the neural basis of decision making, if true, and therefore commendable. In their revision, the authors clarified several important points. However, I am afraid that their response to my and the other reviewers' previous concerns has not convinced me that their data clearly support this claim. While some of my previous comments were addressed, others concerns were mostly unaddressed.The authors have designed a task that builds on a similar task from Wilson et al. (2014) that set out to experimentally dissociate reward difference and information difference between choice options. However, rather than use those terms for the fMRI analysis and/or focus separately on conditions in which these terms are experimentally dissociable (such as considering equal and unequal information trials separately), the way in which the relative reward and information terms are defined for the fMRI analysis introduces a significant confound between the relative reward and information gain terms (as the authors themselves point out in the manuscript and Figure 1). Apparently, this is done because the design does may not be sufficiently well powered to focus on each condition separately. The approach adopted to try to deal with this confound is instead to estimate separate GLMs (3 and 4) and orthogonalise one term with respect to the other in reverse order. This approach does not appear to "control for the confound", but instead assigns all shared variance to the un-orthogonalised term. That is, the technique does not affect the results of the orthogonalized regressor but instead assigns any shared variance to the unorthogonalised regressor (see e.g. Andrade et al., 1999). Thus, in their analysis of relative reward, the relative reward regressor is given explanatory power that might derive from information value, and in the analysis for information value, the reverse is true.

We would first like to address the reviewer’s characterization of our analysis approach. The reviewer states “*Thus, in their analysis of relative reward, the relative reward regressor is given explanatory power that might derive from information value, and in the analysis for information value, the reverse is true.”* This is incorrect; in our analysis of relative reward, explanatory power is assigned to all other regressors before it is assigned to relative reward (i.e., relative reward is orthogonalized w/r/t all other regressors). Similarly, in our information value analysis, explanatory power is assigned to other variables prior to information value. That is, any shared variance between our regressor of interest and other explanatory variables is explicitly NOT assigned to the regressor of interest. This approach does not lead to issues with interpreting the β parameter assigned to our regressor of interest (Mumford, Poline and Poldrack, 2015).

Notwithstanding the interesting results from their simulations, this approach makes the interpretation of the β coefficients and the claim of independence challenging, particularly if one of the scenarios included in the simulations (independent reward value and information systems, or single value system) does not accurately capture the true underlying processes in the brain. More generally, because the two terms are highly correlated, the model-based approach across all trials requires that their computational model can precisely identify each term of interest accurately and independently. This is the same problem noted in other previous studies of exploration/exploitation that experimentally dissociating relative reward and relative information was supposed to address (at least as I understood it).

While no one can guarantee that a model will ever perfectly capture the true mechanisms underlying behavior, we can compare how well models fit the data to determine which one provides a better (even if not perfect) explanation (Cogliati-Dezza et al., 2017). We can, furthermore, conduct repeated fits of the model to the data to determine how consistent/precise the parameter fits are. Generally speaking, if parameters in the model (such as reward and subjective information gain) are not independent, this will decrease the consistency of parameter estimates. Finally, we can produce simulated behavioral data from the model using known parameter values, conduct model fits to this synthetic data, and compare the recovered parameters to those used to generate the data in order to get an idea of how accurate the model is.

We address how we did all of this in previous versions of the manuscript in response to comments from other reviewers.

Relative Reward DefinitionAs I stated in my previous review, the precise definition of relative reward is important for determining what computation is driving the signals measured in vmPFC and dACC. Consideration of alternative terms was appropriately applied to analysis of vmPFC (following reviewer 1 and my suggestions), but not to analysis of dACC activity. Specifically, my suggestions to compare different relative value terms, specifically V2-E(V2,V3) and entropy(V1,V2,V3) (definitions that are based on dACC effects in past studies) for dACC activity were not performed (or reported). I suspect one of these would show an effect in dACC, but it's not possible to tell because they did not address the question or show the results. Importantly, as I pointed out previously, there is evidence in their Figure 4E for a marginal? negative effect of the relative reward term they did test in the dACC ROI. This suggests that getting the term's formulation correct could easily result in a significant negative effect. If there were an effect of either of these alternative terms in dACC, it would go against one of the central claims of the paper. In the authors' reply, rather than test these alternatives, they seemed to simply say they did not know what my comment had meant.

To answer the reviewer’s concerns, we first entered alternative reward terms in GLM4 to see whether activity in dACC associated with information still holds when accounting for additional reward computations. In the previous version of this manuscript we showed that adding expected reward instead of relative reward did not change the results – activity in dACC associated with information value. In the current version of the manuscript we have also added an additional GLM which includes additional reward regressors (see new text below).

Different ways of defining the same underlying concept (reward, in this case), however, may result in correlated values, potentially introducing problems in estimating a GLM (i.e., multicollinearity). In order to identify reward definitions that may be collinear, we calculated Variance Inflation Factors (Craney and Surles, 2002) for the suggested definitions of reward for each subject, and removed regressors with VIFs (averaged across subjects) above a threshold of 5 (interpreted as 80% of the variability in the regressor can be explained by the rest of the regressors).

We immediately experienced problems with this approach because, due to the high degree of collinearity, each definition of reward suggested by the reviewers (with the exception of the second-most-valuable reward) was almost perfectly predicted by the other definitions of reward, i.e., the VIF was infinite, or at least very, very large (keep in mind that a VIF of 5 is considered to be ‘high’). Rather than removing definitions at random, we introduced a small amount of normally-distributed random noise (mean = 0, σ = 0.01) to the regressor values and conducted our analysis again. Our intent in introducing noise was to reduce correlations amongst regressors sufficiently to make ordinal judgments about their VIFs.

Using this approach, we removed variables with the highest VIFS iteratively. In order, these were: the value of the chosen reward minus the second highest reward (VIF ≈ 2 X 10^5^, 99.9995% of variability explained by other regressors), and the maximum reward (VIF ≈ 320, 99.69% of variability explained by other regressors, after removing ChosenMinusSecond). After removing these two regressors, VIFs for the remaining regressors were all under 5 (average value VIF = 3.52, value of the second-best option VIF = 1.99, relative value VIF = 2.21, and minimum value VIF = 3.42).

We note that the AverageReward regressor had the highest VIF initially (VIF ≈ 3.6x 10^5^). However, we elected to retain this definition since behavior and brain activity have previously been linked to the overall level of reward across options (Cogliati Dezza et al., 2017, Kolling et al., 2012).

Our finding of very high VIFs for different definitions of reward suggests that, in our design, we are unable to dissociate reward definitions – the magnitude of the VIFs indicates that some of these regressors, being highly correlated, are essentially indistinguishable. It should be noted that differentiating amongst definitions of rewards was NOT a primary goal of this study. On a positive note, this analysis suggests that our conclusions regarding the dissociation of informationValue and rewardValue are not greatly impacted by the specific definition of reward; effects associated with a regressor with an extremely high VIF (i.e., one that shares practically all its variance with one or more other regressors) would be captured by the remaining, collinear regressors following its removal.

The new text reads as follow:

*“*For GLM4, similar results as those reported above were observed when ExpReward was entered in GLM4 instead of RelReward (GLM4bis: t(19) = 4.4, p < 10-3) and when accounting for covariates for reward i.e., Average Value, the value of the best second option- mean (value chosen, value third option) and Min Value. The rationale behind choosing these reward covariates was as follows: we calculated Variance Inflation Factors 36 for the possible definitions of reward (suggested by reviewers) for each subject, and removed regressors with VIFs (averaged across subjects) above a threshold of 5 (interpreted as 80% of the variability in the regressor can be explained by the rest of the regressors). Due to high multicollinearity amongst the possible reward definitions, VIFs for each definition were indistinguishable (≈ infinity). Rather than removing definitions at random, we introduced a small amount of normally-distributed random noise (mean = 0, σ = 0.01) to the regressor values and conducted our analysis again. Our intent in introducing noise was to reduce correlations amongst regressors sufficiently to make ordinal judgments about their VIFs.

Using this approach, we removed variables with the highest VIFS iteratively. In order, these were: the value of the chosen reward minus the second highest reward (VIF ≈ 2 X 105, 99.9995% of variability explained by other regressors), and the maximum reward (VIF ≈ 320, 99.69% of variability explained by other regressors, after removing Chosen-Second). After removing these two regressors, VIFs for the remaining regressors were all under 5 (average value VIF = 3.52, value of the second-best option VIF = 1.99, relative value VIF = 2.21, and minimum value VIF = 3.42). We note that the AverageReward regressor had the highest VIF initially (VIF ≈ 3.6x 105). However, we elected to retain this definition since behavior and brain activity have previously been linked to the overall level of reward across options 13,32. Next, we ran a ROI analysis based on dACC and vmPFC coordinates observed in GLM1. Results showed significant activity in dACC which positively correlates with Information Gain (t(19) = 2.73, p = 0.013), while we found no correlated activity in vmPFC as was observed in GLM2 (t(19) = – 1.5, p = 0.1503). ”

The second alternative hypothesis is that the dACC is signaling the evidence to change behavior (e.g. Hayden et al., Nature Neuroscience, 2011). That study shows that dACC neurons "encode a decision variable signaling the relative value of leaving a depleting resource for a new one". This is related to the claim already made by the authors; however, this decision variable incorporates not only the expected information gain from exploring but also the relative value of the currently sampled option to the value of the environment (in their study this term incorporates travel time). Based on this view, in the present study, such a signal could amount to the value of the alternative option(s) relative to the chosen option. One reasonable definition would simply be the best non-chosen option's value relative to the chosen option's value because this would capture the reward-related evidence for switching on the next trial. By definition, it is the case that the next best (non-chosen) option's value will be particularly high on trials where subjects explore. Since explore decisions were apparently the most frequent, and the information value is also high on these trials, it is unclear whether the observed signal in dACC reflects the information value alone or might also be related to the relative value of the next best (non-sampled) option.

Next, the second reviewer’s concern is that dACC may compute switching behavior (“change behavior” in the reviewer’s words) or choice difficulty rather than the information value as claimed in our previous versions of this manuscript. To account for this, we entered in GLM4 reaction time and across-option standard deviation (i.e., the standard deviation of expected reward values of the 3 options at time of the first free choice) as proxy for choice difficulty, and switch-stay regressor as proxy for switching behavior or “change behavior” (i.e., coded 0 if choices on the first free choice trial where the same as previous trial choice and 1 otherwise). Adding these additional regressors did not change the results for GLM4. The new text reads as follows “We then entered choice reaction time as additional regression in GLM4 alongside a proxy for across-option standard deviation (i.e., the standard deviation of expected reward values of the 3 options at time of the first free choice) as an additional proxy for choice difficulty. Moreover, in the same GLM we entered a switch-stay regressor as proxy for switching behavior (i.e., coded 0 if choices on the first free choice trial where the same as previous trial choices and 1 otherwise). We then used vmPFC and dACC ROIs from GLM1 to analyze the data. Results essentially replicate the above findings with significant activity in dACC which positively correlated with Information Gain (t(19) = 2.4, p = 0.027), while we found no significant activity in vmPFC (t(19) = – 0.757, p = 0.544).”

Information Value ConfoundsIn their task design the information term (I(c)) is directly proportional to the number of times an item is chosen during forced choice trials. Therefore, this term is highly correlated with the novelty/familiarity of that option. The authors clarified that they did not cue to subject about the horizon (as done in a previous task by Wilson et al. 2014). The authors state cueing the horizon did not matter in prior behavioral studies of their task (which is somewhat surprising if subjects are engaging in directed exploration), but in any case this means that the number of samples is the only variable determining the "information value". Had the horizon been cued, then one could at least test whether dACC activity increased with the planning horizon, since there would be an opportunity to benefit from exploration when this was greater than 1, and this manipulation would be useful because it would mean I(c) was not solely determined by the number of times an option had been chosen in forced choice trials and therefore highly correlated with familiarity/novelty of a deck.

The horizon condition from Wilson et al. 2014 explicitly relates to the instrumental value of information, i.e., the potential benefits of exploration for long-term reward maximization. We address instrumental information in response to Reviewer 3’s comments using a Bayesian model for the task, and find that dACC activity cannot only be explained as deriving from instrumental information gain vs non-instrumental information gain. In other words, this result suggests that dACC encodes both how much information will be useful for future choices as well as the ‘curiosity’ in receiving the novel information right away.

Importantly, the I(c) term also appears to be confounded by the choice of the subject. As I noted in my previous review comment, because the I(c) term is only modeled at the first free choice, its inverse (which is what is claimed to be reflected in dACC activity) will be highest whenever the subject explores. Therefore, the activity in dACC could instead be related to greater average activity on exploratory compared to exploitative choices, as has been suggested previously, without any encoding of "information value" per se.

Our understanding is that the reviewer’s use of the term “information value” relates solely to the instrumental utility of information, and the reviewer contrasts this with ‘exploratory’ choices. Our use of the term ‘information value’, however, relates specifically to the amount of information about an option that could be gained from selecting it (our calculation of InformationValue is defined in Methods around line 498). The distinction between a choice that is exploratory in nature and one that is directed toward maximizing information gain (regardless of the information’s instrumental utility) is not clear to us – certainly the reviewer agrees that selecting an option that does not provide any additional information could not be called ‘exploratory’ since the subject already knows everything about that option. Similarly, selecting an option about which the subject knows nothing (i.e., an option that provides the highest amount of new information, i.e., information value) seems to be a pretty good definition of exploration. Our I(c) term can be thought of as operationalizing the term ‘exploration’ as the subjective amount of information a subject stands to gain from choosing an option.

For this reason I suggested including the choice (exploratory vs exploitative or switching away from the most selected "default" option) as a covariate of no interest to account for this possibility. This comment was not directly addressed.

As discussed immediately above, the I(c) term is a quantification of ‘exploration’, and sidesteps issues surrounding labeling trials as either being purely exploratory or purely exploitative in nature (e.g., if all options have been sampled twice, but the subject picks the least-valuable option, is that exploratory? If so, is it exploratory in the same way that choosing a never seen option is? If the subject selects an option that has been observed most often in the forced choice stage, but the average value for that option is well under the long-run average for the entire experiment, is that choice truly exploitative? Or could choosing the never-seen option be both exploratory and exploitative?). The complexity of exploration as well as the impossibility to accurately categorize exploration vs. exploitation in certain scenarios has already been pointed out by others in the field (e.g., Mehlhorn et al. 2015, Gershman 2018).

Certainly, if we limited ourselves to considering only those trials in which it is unambiguous as to whether the subjects were exploring or exploiting via their choices, the explore/exploit covariate the reviewer requests would eliminate the effect of I(c) simply because I(c) and directed exploration describe the same thing (picking the option about which least is known). This is not because we have confounded exploration with information gain in our design, but because information gain is one way to define exploration.

Instead, the authors make a somewhat convoluted argument that switching away from a default in their task corresponds to not exploring, since explore choices were the more frequent free choices in their task (though this would not be the case for the current game, where the forced choices would likely determine the current "default" option).

We regret the reviewer was not convinced by our arguments regarding default behavior in this task. We acknowledge that there is some ambiguity on what constitutes a ‘default’ choice – at the level of a single game, the default could be considered to be the option that has been most frequently selected, even if this ‘choice’ was cued during the forced choice stage. Alternatively, the default behavior could be defined on the choices the subject makes freely without cueing. When given the opportunity, subjects in our experiment predominantly select the least-seen option at the first opportunity. There are thus two levels at which ‘default’ behavior can be defined. The reviewer suggests that it is based on the history of choices, forced or not, while we define it based on the subject’s own free choices, specifically those choices on the 1^st^ free choice trial.

Ultimately, the ‘true’ definition of default behavior doesn’t strike us as being the central issue here – the behavior of interest is whether subjects switch options (whether it be *away from* or *towards* a default option, depending on which definition of default behavior you prefer) on the first free-choice trial or stay with options they have selected previously. Analysis of ‘switch-stay’ behavior is commonly used to investigate the exploration-exploitation tradeoff in humans and animals (e.g., Chan, Asis, Sutton 2020; Domenech et al. 2020; Findling et al. 2019), and in our task, has the advantage of generalizing to trials in which there is no obvious default behavior (i.e., those trials in which each option has been selected twice during the forced-choice stage). We re-ran GLM 4 with a switch-stay regressor, and found activity comparable to the same analysis without the switch-stay regressor: significant activity in dACC which positively correlated with Information Gain (t(19) = 2.4, *p* = 0.027), while we found no significant activity in vmPFC (t(19) = – 0.757, *p* = 0.544).

If the dACC is in fact encoding the "information value" then this effect should remain despite inclusion of the explore/exploit choice, meaning it should be present on other trials when they do not choose to explore. This concern was not addressed, although it is unclear if their experimental design provides sufficient independent variance to test this on exploit trials.

Again, we note that there are issues with labeling trials as being exploratory or exploitative that we sought to avoid through model-based analyses. As we indicate above, when only considering trials in which behavior can be labelled unambiguously as exploratory or non-exploratory, information value entirely overlaps with those labels. However, the I(c) regressor encodes prospective information gain for *all* trials, including those for which all options have been observed an equal number of times, i.e., trials which cannot clearly be labelled as exploratory or non-exploratory. Since these trials cannot be clearly labelled one way or the other, they would have to be excluded from the analysis suggested by the reviewer as there is no principled way to categorize them.

However, it is exactly these trials that support our interpretation of dACC as signaling information value.

If these ambiguous trials were truly either explore/non-explore trials then, under the hypothesis that dACC activity simply indexes a binary exploration (high activity)/non-exploration (low activity) variable, regressing I(c) against dACC activity would result in a relatively poor fit. Approximately ½ of all trials in the experiment fall in this ambiguous category – assuming half of those are ‘true explore’ and the other half are ‘true non-explore’, *we would expect to see bimodal dACC activity* that would be poorly captured by the value of I(c) for those trials.

To test whether activity in dACC is consistent with an exploration signal, we created a glm with 128 regressors (one for each first free choice, i.e., a β series analysis). We then extracted the betas estimates for the dACC roi on the equal trials (i.e., those in which each option had been observed twice during the forced-choice phase, and thus could not be unambiguously categorized as exploratory/non-exploratory). We first visually inspected the distribution of the β estimates. In author response image 2 we plot for a subject (left) and for all subjects (right). As shown in both figures, the distribution did not look bimodal as the explore/exploit account would predict.

Next, we ran the Hartigan’s Dip Test to test whether the distribution is bimodal. Results (reported below) showed that the β estimates distribution for the dACC roi on equal trials is not bimodal for any subject. These findings suggest that dACC activity does not index binary activity (e.g., high for exploration and low for exploitation).

**Author response table 1. sa2table1:** 

Subject	Pvalues
Subject2	0.698
Subject3	0.926
Subject5	0.878
Subject6	0.9
Subject7	0.844
Subject8	0.988
Subject9	0.73
Subject10	0.866
Subject11	0.968
Subject12	0.38
Subject13	0.98
Subject14	0.476
Subject15	0.198
Subject16	0.958
Subject17	0.92
Subject18	0.34
Subject19	0.998
Subject20	0.922
Subject21	0.976
Subject22	0.666

As I pointed out in my original review, if the dACC shows no effect of relative value (or the alternative terms described above) as claimed, then there should be no effect of the relative value on the "equal information" trials. The authors do report no significant effects at a p<0.001 whole-brain threshold on these trials, which they acknowledge may be underpowered. It would be more convincing to show this analysis in the dACC ROI that is used for other analyses. A significant effect on these trials that experimentally dissociate relative reward from information would also run counter to one of the main claims in the paper.

We ran the analysis suggested by the reviewer in which dACC roi from GLM2 was added to investigate activity within that roi for the GLM with one parametric modulator (relative reward) on equal trials. As shown in Author response image 3, this analysis did not reveal any activity.

**Author response image 3. sa2fig3:** 

Lastly, we have run a univariate analysis comparing activity associated with “choosing the best rewarding option in previous trials history” and “not-choosing best rewarding options”. Which is another way to operationalize exploration-exploitation trade-off. We focus on trials in which reward is not confounded with information (i.e., equal information condition). We did not observe however any activity in dACC for the contrast equal exploration – equal exploitation (EqPlore-EqPloit; we have reported SPM output in Author response image 4 – panel on the left), while posterior cingulate cortex often associated with exploitation; (McCoy, Crowley, Haghighian, Dean, and Platt, 2003; Pearson, Heilbronner, Barack, Hayden, and Platt, 2011; Lebreton, Jorge, Michel, Thirion, and Pessiglione, 2009) was associated with equal exploitation – equal exploration (EqPloit-EqPlore; panel on the right). Activity on dACC was not observed even when using dACC ROI from GLM1.

**Author response image 4. sa2fig4:** 

**Author response image 5. sa2fig5:** 

For the above reasons, I do not believe my previous concerns were addressed and that the data support some of the central claims of the manuscript.References:Andrade A, Paradis AL, Rouquette S, Poline JB (1999) Ambiguous results in functional neuroimaging data analysis due to covariate correlation. Neuroimage 10:483-486.

We thank the reviewer for the useful comments which have helped us to strength the manuscript. We believe with our additional analyses we have addressed all reviewer’s concerns.

Reviewer #3:This is a greatly improved manuscript. The authors have done substantial work thoroughly revising their manuscript including carefully addressing almost all of my original comments. These have made the paper considerably more methodologically solid and clarified their presentation, but have even allowed them to uncover additional results that strengthen the paper, such as providing evidence that ACC activity related to information is especially related to the non-instrumental value of information.I do have a few questions and concerns for the authors to address regarding the methodological details of some of their newly added analyses (comments 1-4). I also have some suggestions to improve their presentation of the results to help readers better understand them (comments 5-6 and most minor comments).

We thank the reviewer for these positive comments. We believe that the additional changes have strengthened even more the manuscript and we thank the reviewer for the contribution.

1. The authors have included a new analysis that nicely addresses my comments about the need to distinguish between neural signals encoding Instrumental vs Non-Instrumental Information. I greatly appreciate them developing the quite painstaking and elaborate computations necessary to estimate the instrumental value of information in this task! Furthermore, this led them to report a very interesting result that supports their big picture message, that ACC activity is most related to their estimate of Information Gain from their gkRL model of behavior, while vmPFC is most related to their estimate of Instrumental Information. This is very striking, and if true, is an important contribution to our understanding of how neural systems value information.I have a few comments about this:- Overall, the whole approach is explained very clearly. It appears methodologically solid, strongly motivated, and well-founded in Bayesian decision theory.There is one missing step that needs to be explained, though: the final step for how they use the Bayes-optimal long term values for each option (which I will call "Bayes Instrumental Value") to compute the Instrumental Information which they used as a regressor. I don't quite understand how they did this. Reading the methods, it sounds almost as if they set Instrumental Information = Bayes Instrumental Value. If so, I don't think that would be correct, since then a large component of "Instrumental Information" would actually just be the ordinary expected gain of reward from choosing an option (e.g. based on prior beliefs about option reward distributions and on the mean experienced rewards from that option during the current game), not specifically the value of the information obtained from that option (i.e. the improvement in future rewards due to gaining knowledge to make better decisions on future trials). I worry that the authors might have done this, since their analysis suggests that Instrumental Information may be positively correlated with experienced rewards (e.g. it was correlated with similar activations in vmPFC as RelReward). It is also odd that "The percentage of trials in which most informative choices had positive Instrumental Information was ~ 22%" since surely the instrumental value of information in this task should almost always be at least somewhat positive and should only be zero if participants know they are at the end of the time horizon, which should never occur on first free choice trials.In general, a simple way to compute the instrumental value of information (VOI) in decision problems is to take the difference between the total reward value of choosing an option including the benefit of its information (i.e. the Bayes Instrumental Value that the authors have already computed) minus the reward value the option would have had if it provided the decision maker with the same reward but did not provide any information (which I the authors may be able to calculate by re-running their Bayesian procedure with a small modification, such as constraining the model to not update its within-game belief distributions based on the outcome of the first free choice trial, thus effectively ignoring the information from the first free-choice trial while still receiving its reward). Thus, we would have a simple decomposition of value like this:Bayes Instrumental Value = Reward Value Without Information + Instrumental Value of Information- Relatedly, I was surprised by their fit to behavior that predicts choices based on Instrumental Information and Information Gain. Given the strong effects of expected reward on choice (Figure 2D), I naively would have expected them to include an additional term for reward value, in order to decompose behavior into three parts, "Reward Value Without Information" representing simple reward seeking independent of information, "Instrumental Value of Information" representing the component of information seeking driven by the instrumental demands of the task, and "Information Gain" representing the variable the authors believe corresponds to the non-instrumental value of information (or total value of information) reflecting the subjective preferences of individuals. The way I suggested above to compute Instrumental Information would allow this kind of decomposition.- I suggest adding a caveat in the methods or discussion to mention that the Bayesian learner may not reflect each individual's subjective estimate of the instrumental value of information, since the task was designed to have some ambiguity allowing the participants to have different beliefs about the task, by not fully explaining the task's generative distribution.

As the reviewer suggested we computed the instrumental value of information as Bayesian Instrumental Value – Reward Value without information. The new text (in the methods) reads as follows: “The Bayes Instrumental Value corresponds to the overall expected reward value of choosing which includes both reward and information benefit. As the instrumental value of information is the difference between the overall expected reward value of choosing which includes both reward and information benefit (i.e., Bayes Instrumental Value) and the reward value obtained from an option without receiving information (i.e., Reward Value without information), the latter was also computed. To do so, the Bayesian procedure was implemented by constraining the model to not update its belief distribution based on the information provided on the first free choice trial. Next, expected instrumental value (Instrumental Information) for each option on the first free choice trial following the forced-choice trials specific to that game was computed as:

Instrumental Information c, j= Bayes\ Instrumental\ Valuec,j− Reward\ Value\ without\ informationc,j(7)

On additional note, as subjects was not aware of the reward distributions adopted in the task – therefore they might develop different beliefs- the above procedure may not reflect each individual’s subjective estimate, rather it reflects an objective estimate of the instrumental value of information.”

In the Results section, we have updated the results with the novel computation. The new text reads as follows: “We then entered Instrumental Information and Information Gain as parametric modulators into two independent GLMs. We investigated the effects of Information Gain after controlling for Instrumental Information (GLM5) and the effects of Instrumental Information after controlling for Information Gain (GLM6; Methods). Next, we run a ROI analysis based on dACC and vmPFC coordinates observed in GLM1 (as in both GLMs information signals were computed). Results showed that activity in dACC was positively correlated with Information Gain (after controlling for Instrumental Information; t(19) = 5.56, p < 10-4) and no significant vmPFC activity was found, t(19) = – 1.69, p = 0.108. On the contrary, in GLM6, both vmPFC (t(19)=4.83, p < 10-4) and dACC activity (t(19) = – 2.92, p = 0.009) were found. These results suggest that dACC encodes both information values, whit the non-instrumental value of information solely encoded in this region while the instrumental value of information was also encoded in reward regions.”

Also behavioral results look as previous analysis. The new text reads as follows: “We found a positive effect of Information Gain (β coefficient = 71.7 ± 16.76 (SE), z = 4.28, p = 10-5) and a negative effect of Instrumental Information (β coefficient = -2.3 ± 0.463 (SE), z = -4.98, p < 10-6) on most informative choices. The percentage of trials in which most informative choices had positive Instrumental Information was ~ 20% suggesting that in most of the trials informative options were not selected based on instrumental utility.” We do understand that it seems odd to the reviewer that the instrumental value of information can also be negative. However, this result is grounded in the current theories of information-seeking. In particular, from Sharot and Sunstein, 2020, NHB “What has often been overlooked, however, is that information can also have negative instrumental value. That is, knowledge can at times cause individuals to select actions that lead to worse outcomes, while deliberate ignorance can lead to better outcomes”. This is the case in our task. Indeed, the generative mean for each option was stable within a game but varied across games. The generative mean of the three decks had the same value in 50% of the games (*Equal Reward)* and different values (*Unequal Reward*) in the other 50% of the games. In the Unequal Reward condition, the generative means differed so that two options had the same *higher* reward values compared to the third one in 25% of the games (*High Reward*), and in 75% of the games, two options had the same *lower* reward values compared to the third one (*Low Reward*). This may lead to negative instrumental information in the unequal reward condition/unequal information condition when the most informative option has lower values compared to the other/s option/s. We acknowledge this in-depth explanation of the reward distribution was not included in the previous version of this manuscript, but only in previous publications of this task (Cogliati Dezza et al. 2017, 2019, 2021). We have now added this point in the method section. The new text reads as follows: “The generative mean for each option was stable within a game but varied across games where it had different values in the other 50% of the games. In these games, the generative means differed so that two options had the same higher reward values compared to the third one in 25% of the games, and in 75% of the games, two options had the same lower reward values compared to the third one.”

Lastly, we have added a sentence in the method section pointing to the objective computation (rather than subjective) of the instrumental value of information. The new text reads as follows: “On additional note, as subjects was not aware of the reward distributions adopted in the task – therefore they might develop different beliefs- the above procedure may not reflect each individual’s subjective estimate, rather it reflects an objective estimate of the instrumental value of information.”

2. The difference between the confounded analysis (Figure 3C) and non-confounded analysis (Figure 4E) is very striking and clear for vmPFC but is not quite as clear for ACC. It looks like the non-confounded analysis of ACC produces slightly less positive β for information and slightly less negative β for reward (from about -0.5 to about -0.35, though I cannot compare this data precisely because these plots do not have ticks on the y-axis). While is true the effect went from significant to non-significant, I am not sure how much this change supports the black and white interpretation in this paper that ACC BOLD signal reflects information completely independently of reward. It might support a less black and white (but still scientifically important and valuable) distinction that ACC primarily reflects information but may still weakly reflect reward. I understand it may not be possible to 'prove a negative' here, but it would be nice if the authors could show stronger evidence about this in ACC, e.g. by breaking down the 3-way ANOVA to ask whether removing the confound had significant effects in each area individually (vmPFC and ACC). Otherwise, it might be better to state the conclusions about ACC and information vs. reward in a less strictly dichotomous/independent way.

We have implemented two additional ANOVAs as the reviewer suggested and added to the main text. The new text reads follows: “Lastly, we check whether accounting for confounded reward and information signals had significant effects in both regions separately. To do so, we ran a 2-way ANOVA with Value Type (Information Gain, RelReward), Analysis type (confounded{GLM1and2}, no-confounded {GLM3 and 4}) for dACC ROI and a 2-way ANOVA with Value Type (Information Gain, RelReward), Analysis type (confounded{GLM1and2}, no-confounded {GLM3and4}) for vmPFC ROI. Results showed a significant 2-way interaction for the dACC ROI (F(1,19) = 25.7, p = 0.0001) as well as for the vmPFC ROI (F(1,19) = 13.7, p = 0.0015).”

3. The analysis of switching strategy is a valuable control, and I agree with the authors on their interesting point that in this task the default strategy would appear to be information seeking rather than exploitation. Can the authors report the results for ACC? The paper seems to describe this section as providing evidence about what ACC encodes, so I was expecting them to report no significant effect of Switch-Default in ACC, but I only see results reported for frontopolar cortex.

We have now added results from the dacc ROI analysis. The new text reads as follows: “We additionally analyzed the data using the dACC ROI from GLM1. Results showed no activity in the selected ROI (t(19) = 0.664, p = 0.514).”

4. The analysis of choice difficulty is also a very welcome control. My one suggestion for the authors to consider is also correlating chosen Information Gain with a direct index of choice difficulty according to the gkRL model of behavior that they fit to each participant (e.g. setting choice difficulty to be the difference between V of the best deck with V of the next-best deck). This would be more convincing to choice difficulty enthusiasts, because reaction times can be influenced by many factors in addition to choice difficulty, and previous papers arguing for dACC role in choice difficulty (e.g. Shenhav et al., Cogn Affect Behav Neurosci 2016) define choice difficulty in terms of model-derived value differences (or log odds of choosing an option). The gkRL model is put forward as the best model of behavior in this task in the paper, so it would presumably give a more direct estimate of choice difficulty than reaction time.

We thank the reviewer for this comment. We actually implemented a slightly different approach but still in line with what the reviewer suggested. We computed the SD across options. If SD is small, choosing among options is harder compared to greater values of SD across options. The new text reads as follows: “We then entered choice reaction time as additional regression in GLM4 alongside a proxy for across-option standard deviation (i.e., the standard deviation of expected reward values of the 3 options at time of the first free choice) as an additional proxy for choice difficulty. Moreover, in the same GLM we entered a switch-stay regressor as proxy for switching strategy (i.e., coded 0 if choices on the first free choice trial where the same as previous trial choices and 1 otherwise). We then used vmPFC and dACC ROIs from GLM1 to analyze the data. Results essentially replicate the above findings with significant activity in dACC which positively correlated with Information Gain (t(19) = 2.4, p = 0.027), while we found no significant activity in vmPFC (t(19) = – 0.757, p = 0.544).”

5. One of the main messages of the paper is that correlations between reward and information may cause ACC and vmPFC to appear to have opposing activity under univariate analysis, even if ACC and vmPFC actually have independent activity under multivariate analysis.On this point, the authors nicely addressed my first comment by greatly improving and clarifying Figure 1 showing their simulation results about this confound. The authors also addressed my third comment, which was also on this topic.However, part of the way they explain this in the text could do with some re-phrasing, since in its current form it could be confusing for readers. Specifically, they make some statements as if information and reward were uncorrelated ("Both Wilson et al. and our task orthogonalize reward and information by adding different task conditions." and "the use of the forced-choice task allows to orthogonalize available information and reward delivered to participants in the first free choice trial") but make other statements as if they were correlated ("However, in order to better estimate the neural activity over the overall performance we adopted a trial-by-trial model-based fMRI analyses. Since reward value and information value are correlated due to subjects' choices during the experiment, this introduces an information-reward confound in our analysis.").After puzzling over this and carefully re-reading the Wilson et al. paper I understand what they are saying. Here is what I think they want to say. On the first free-choice trial, if you consider all available options, an option's information and that option's reward are uncorrelated. However, if you only consider the chosen option (as most of their analyses do) that option's information and reward are negatively correlated.If this is the case, the authors need to clarify their explanations, especially specifying when "information" and "reward" refer to all options or only to the chosen option. For example, instead of "orthogonalize available information and reward delivered to participants in the first free choice trial" I think it would be more clear and accurate to say "orthogonalize experienced reward value and experienced information for all available options on the first free-choice trial".Relatedly, in my opinion this whole issue would be clearer if the authors took my earlier suggestion of simply showing a scatterplot of the two variables they claim are causing the problem due to their correlation (presumably "chosen RelReward" vs "chosen Information Gain"), for instance in Figure 1 or in a supplementary figure. This would be even more powerful if they put it next to a plot of the analogous variables from considering all options showing a lack of correlation (e.g. "all options RelReward" vs "all options Information Gain") thus demonstrating that the problematic correlation is induced by focusing the analysis on the chosen option. Then readers can clearly see what two variables they are talking about, how they are correlated, and why the correlation emerges. It seems critical to show this correlation prominently to the readers, since the authors are holding up this correlation, and the need to control for it, as a central message of the paper.

We thank the reviewer for this comment. In the current version we extensively revised this part and we believe this comment does not longer apply. Indeed, in the novel frame, we specify that reward and information often explain similar variance. Information signals are partly characterized by reward-related attributes such as valence and instrumentality (Sharot and Sunstein, 2020), while reward signals also contain informative attributes (e.g., winning $50 on a lottery allows the recipient to gain the reward amount but also information about the lottery itself; Wilson et al. 2014). Because of this “shared variance” it is not surprising that the neural substrates underlying reward value processing frequently overlap with those involved in optimizing information. This raises an interesting question as to whether information and reward are really two distinct signals. In other words, is information merely a kind of reward that is processed in the same fashion as more typical rewards, or is the calculation of information value independent of reward value computations? Simulations of our RL model which consists of independent value systems independently optimizing information and reward, however, suggest that fMRI analyses might reveal overlapping activity if the “shared variance” between reward and information is not taken into account. As we have decided that “shared variance” terminology is more appropriate than ‘correlation’ we do believe that the reviewer’s concern above does not longer apply to the current version of the manuscript.

Additionally, we would like to highlight that because of the above changes we have also modified figure 1 to only show activity of a dual-value system as the all point is to show that even if a system independently compute information and reward, if the shared variance is not taken into account, shared activity between reward and information is observed. In the new frame, we do not consider single vs. dual system rather whether a system computes values independently or not. However, we would like to thank the reviewer for her/his precious comments on this matter.

6. The paper has a very important discussion about how broadly applicable their approach to non-confounded analysis of decision variables is to the neuroscience literature. There is one part that seems especially important to me but may be hard for readers to understand from the current explanation:"Furthermore, we acknowledge that this confound may not explicitly emerge in every decision-making studies such as preference-based choice. However, the confound in those decision types is reflected in subjects' previous experiences (e.g., the expression of a preference for one type of food over another are consistent, and therefore the subject reliably selects that food type over others on a regular basis). The control of this confound is even more tricky as it is "baked in" by prior experiences rather than learned over the course of an experimental session. Additionally, our results suggest that other decision dimensions involved in most decision-making tasks (e.g., effort and motivation, cost, affective valence, or social interaction) may also be confounded in the same manner."Here is what I think the authors are saying: In their task, people made decisions based on both reward and other variables (i.e. information), which induced a correlation between the chosen option's reward value and its other variables. If the authors had analyzed neural activity related to the chosen option only in terms of reward value, they would have thought that ACC and vmPFC had negatively related activity to each other, and had opposite relationships to reward value. In effect, any brain area that purely encoded any one of the chosen option's variables (e.g. information) would appear to have activity somewhat related to all of the other decision-relevant variables. Thankfully, the authors were able to avoid this problem. However, crucially, they were only able to avoid this problem because their task was designed to manipulate the information variable. This let them model its effects on behavior, estimate the information gain on each trial, and do multivariate analysis to disentangle it from reward. Therefore, the same type of confound could be lurking in other experiments that analyze neural activity related to chosen options. This potential confound may be especially hazardous in tasks that are complex or that attempt to be ecologically-valid, since they may have less knowledge or control of the precise variables that participants are using to make their decisions, and hence less ability to uncover those hidden decision-relevant variables and disentangle them from reward value.If this is what they are saying, then I agree that this is a very important point. I suggest giving a more in-depth explanation to make this clear. Also, I think it would be good to state that this confound only applies to analyses of options that have correlated decision-relevant variables (e.g. due to the task design, or due to the analysis including options based on the participant's choice, such as analysis of the chosen option or the unchosen option).

We thank the reviewer for this comment. We agree that section was hard to understand. In the current version, we reframe the manuscript and decided to remove that section.

[Editors’ note: what follows is the authors’ response to the second round of review.]

The revised manuscript has been greatly improved and after some discussion the reviewers agreed that the concerns had been addressed in a responsive revision. However, two points arose in the revision that, after consultation, the reviewers still felt should be addressed, at the very least with clarification and clear discussion of their strengths and weaknesses in the text. These were:a) Reviewers were confused about why instrumental information was negative on the majority of trials, given that a Bayesian optimal learner was used. The concern is elaborated by a reviewer, as follows:"I am still puzzled why they say the most informative deck only has positive Instrumental Information on 20% of trials and often has negative Instrumental Information.The rebuttal is completely correct to point out that instrumental information can be negative. However, this happens when the decision maker is suboptimal at using information so it performs better without some information (e.g. the paper they cite uses examples like "not knowing whether a client is guilty could improve a solicitor's performance"). In this paper the formula is supposed to be using a Bayes optimal learner, that knows the structure of their task in more detail than the actual participants. A Bayes optimal learner should make optimal use of information, and should never be hurt by information, so instrumental information should always be positive or at least non-negative, right?The rebuttal says that instrumental information can be negative in a specific condition in this task with unequal reward, unequal information, and the most informative option has lower reward value than the other options. However, I do not see why this should be the case. Providing information should never hurt a Bayesian with an accurate model of the task.

The reviewers are correct that our Bayes optimal learner should not be ‘hurt’ by information. We have revisited our simulations and determined that (1) instances of negative instrumental information appear to be artefacts of our discretization; (2) this however does not affect our behavioral and neural analysis as we observed that instrumental values obtained with different discretization approaches are highly correlated (above 0.9) and (3) our claim regarding the proportion of trials in which subjects select a choice with negative information was mistakenly based on calculations using standardized instrumental information scores.

(1) As we mentioned previously, in order that the forward tree search finish in a reasonable time frame (read: before the heat death of the universe) we found it necessary to sparsely model the number of points received. Although subjects could receive points from 0 to 100 at 1pt intervals, in our Bayesian implementation the number of points was rounded to the nearest ‘5’ in a range from 5pts to 95pts. The probability distribution over rewards learned was therefore only coarsely approximated using 10 bins in our simulations.

As it happens, an unforeseen byproduct of this coarse representation is that under some circumstances, instrumental utility can take a negative value. The reviewers are correct that the most informative option has a positive instrumental information (and this is reflected in the estimates of the Bayesian learner). However, we note that instrumental information is estimated for each of the three options – the instrumental utility for less informative options frequently takes negative values.

To demonstrate the effect of sampling in our Bayesian learner, we conducted two additional simulations of the forward tree search for a single subject, one in which rewards were rounded to the nearest ‘0’ or ‘5’, and another in which rewards were rounded to the nearest ‘2.5’ – essentially the space between 0pts and 100pts was divided into 20 and 40 bins, respectively. In Author response image 6, we provide a graph demonstrating that, with increased resolution, the number of instances in which a negative instrumental information value is observed approaches zero.

**Author response image 6. sa2fig6:** 

Additionally, we show how the density of sampling influences posterior probability distributions over reward (integrated over σ). In particular, the sparse representation used for reward tends to produce skewed distributions. Author response image 7 shows an example of a distribution skewed to the right (dark green), suggesting that the mean of that distribution is artificially high. Selecting that option – and updating the probability distribution as a result – could result in elimination of the skew (or skew in the opposite direction) – the mean of the distribution thus changes both due to the reward received, but also due to artificial changes in skewness.

**Author response image 7. sa2fig7:** 

In order to calculate Instrumental Utility, in line with suggestions from the reviewers in previous rounds, we compare q-values for a forward tree search under normal Bayesian updating rules with q-values obtained from a forward tree search in which no updating occurs on the 1^st^ choice. As described above, because the non-updated probability distribution can sometimes be skewed positively, and updating can eliminate or reverse the skew, this can result in net negative instrumental utility (i.e., the non-updated distribution retains an artificial positive skew relative to the updated distribution). Symmetrically, we also observe a slight over-representation of positive instrumental utility for the 10 bins case (relative to 20 and 40 bins). This results from distribution estimates that have an artificial negative skew. 2) Although our sampling resolution influences how often a negative value is seen, this has little effect on the analyses we conducted in which we regressed instrumental information against brain activity. Instrumental Info values for the 20 and 40 bin simulations correlated at 0.936 (p<.0.001) and 0.948 (p<.0.001) respectively with the Instrumental Info values obtained from the 10 bin simulations.

We have updated our text around line 639 to acknowledge this artefact: “Although the coarse discretization resulted in somewhat less precise estimates of the distribution over μ and σ, this had minimal effect on our calculation of instrumental information (see below): for a single subject, instrumental utility over all trials when values for μ were discretized into 10 bins correlated at 0.936 (p < 0.001) and 0.948 (p < 0.001) when μ was discretized into 20 and 40 bins.”

Also, the specific condition should only be a fraction of trials, it is surprising if something occurring on those trials could account for the fact that 80% of the trials, they report have zero or negative instrumental information. Finally, it is strange that instrumental information would be negative on those trials because they are ones where one deck has not been sampled and has low expected reward, so it seems to me that there is high instrumental value in learning from information that it has a low value. This information is needed for the participant to learn that the deck has a low value and should be avoided going forward, otherwise if the participant did not learn anything from this information, they would be likely to choose it on their second choice. All of this makes me suspect a bug in the code."

3) The reviewers are correct to be skeptical, and we thank them for bringing the error to our attention. As is clear from our figure above showing the instrumental information of all trials for a single subject, options with negative instrumental information (due to the discretization issue described above) are the minority, an observation at odds with our claim that the most informative choices only had positive instrumental information on 20% of all trials.

Upon re-examining our analyses, we have determined that we based this claim on standardized instrumental information scores – clearly incorrect. Reentering the non-standardized instrumental information scores in our analysis shows that all the informative choices were associated with positive instrumental utility, as previously predicted by the reviewer. We have modified the text at pg. 13 to account for this miscalculation. We regret the error.

This issue is important to resolve. You might just consider working through the computation in a simple example the rebuttal letter.

We believe our responses above identify why our implementation of the Bayesian learner recovers negative information values. We respectfully submit that, given the recursive nature of the forward tree search, it is not possible to give a concise ‘manual’ example of the computations. Consider: for our limited implementation in which there are 10 bins representing rewards the model could receive for its choice, each ‘ply’ of the search involves choosing amongst 3 decks, receiving 1 of 10 rewards as a result, i.e., there are 30 possible states reachable from the 1^st^ free choice trial. Considering 2 choices into the future, there are 30^2 = 900 possible states the model could reach. Each of these states entails updating the posterior distributions for mean reward and variance; even a simple 2-step forward search would require pages of manual computations.

In lieu of that, we provide visualizations in Author response image 8 showing the evolution of the joint probability distribution over reward and variance both between and within games (top and bottom, respectively). The joint distribution develops according to the equations given in the methods.

**Author response image 8. sa2fig8:** 

Note that the visualizations for the local distribution (bottom row) are taken after the model has experienced 28 games total (i.e., the initial joint distribution is not the same as the global prior after game 4 despite the arrow’s implication …). Also note that the peak of the prior distribution after 28 games (bottom row, left panel) is located at approximately 40-50 points, with a σ of around 7.5. This matches the global parameters of the experiment described in the methods (“On each trial, the payoff was generated from a Gaussian distribution with a generative mean between 10 and 70 points and standard deviation of 8 points.”). Finally, because the learner’s estimate of reward variance (σ) is relatively low, its estimate of the reward for a specific deck is fairly precise even after a single sample.Given the above, we hope the reviewers are convinced that the Bayesian Learner was implemented correctly and that the high amount of negative IU was due to entering standardized IU scores into the analysis – that we have now corrected.

b) The omega*I versus I distinction was still confusing to reviewers. This concern is detailed by the reviewer, as follows:"my concern stems from part 3 of the rebuttal, the incorporation of the omega parameter in the information term. The authors report that if they follow our advice to replace -I with -I*omega then their main result in ACC is not significant anymore (at the second level of analysis used to test if activity across participants is significantly affected by information value). Roughly, as I interpret it, -I controls the functional form of how information value depends on samples, and omega controls its magnitude, so both are needed to model information value and its effect on choice. And my analysis suggests that unfortunately -I is not a reliable predictor across individuals of either -I*omega or the effect of information on choice. […] if -I*omega is a more accurate way to quantify information value, does this mean that their current results using -I are wrong or fail to support their major claims?"After discussion, the reviewers were satisfied that I can be a reasonable regressor but felt it important to clarify in the paper that this regressor reflects the functional form of how information value depends on the number of samples, but not the absolute magnitude of info value (e.g. its monetary worth) or its effect on the probability of choosing the most informative option. Further, it would be helpful to justify why it would be reasonable to expect ACC BOLD signals to be related to this type of relative/scaled value of info rather than its absolute monetary value. So, the distinction between these regressors should be acknowledged, along with the null result for the omega*I regressor.

We believe there is substantial ambiguity regarding what information value is, and even in the literature we believe this is not very standardized. The reviewers are correct that -I reflects the functional form of information value, while omega controls the contribution to behavior of information relative to reward value. As we note in the manuscript, we identify Information Value with epistemic value – how much does additional sampling of a deck improve the estimate of its underlying reward independent of the instrumental value of that reward.

The absolute magnitude of information value would combine both its monetary worth (instrumental utility; IU) and its non-monetary worth (non-instrumental info) (see Sharot and Sunstein 2020). We agree that we did not measure the absolute value in such way, as our goal was to dissociate epistemic information value from monetary or instrumental information value. We have clarified this in the revised manuscript “Information Gain was computed as − It, j(c). The negative value It, j(c) relates to the information to be gained about each deck by participants. In particular, − It, j(c)represents the functional form on how information value depends on the number of samples, rather than its absolute term.”

Furthermore, averaging informative choices across unequal trials and correlating these values with averaged information value (-I) result in high correlations (r=0.6, p=0.003). This suggests that -I drives informative choices. Additionally, logistic regression of the number of samples against behavior choice (essentially information +I in which the γ parameter is set to 1) show that information already acquired about an option significantly and negatively contributes to choices, i.e., subjects choose options that they have sampled less frequently. Therefore, we are not sure how the reviewers could claim the following “… -I is not a reliable predictor across individuals of either -I*omega or the effect of information on choice”.

Further, it would be helpful to justify why it would be reasonable to expect ACC BOLD signals to be related to this type of relative/scaled value of info rather than its absolute monetary value.

We respectfully submit that our simulations of a model with independent information value and reward value systems (epistemic and instrumental/reward values, respectively; figure 1 of the manuscript) provide this justification. As we argue in the introduction, independent info and reward systems can present as overlapping signals if shared variance is not taken into account, and we cite a wealth of literature showing this kind of functional overlap in vmPFC and dACC. Our analysis shows that both -I and IU (monetary value) shows activity in ACC, therefore ACC appears to be involved in processing both instrumental information and non-instrumental information signals, and that non-instrumental information signals are dissociable from instrumental information signals (Figure 5). Whether the combination of these signals into a unique value happens in ACC is something we are currently looking into using a different task, as the current task does not allow to investigate this point further.

So, the distinction between these regressors should be acknowledged, along with the null result for the omega*I regressor.

We have added the omega*I regressor results in the Supplementary File 11

c) Beyond these two points, the reviewers had suggestions for some additional clarifications.- If the reviewers' understanding is correct, that the current orthogonalization procedure controls for shared variance is really only applicable when compared to GLMs that only model reward or information, or model them in separate GLMs, rather than the many that model them in the same GLM. This could be clarified in the text.

We have added the following text around lines 703: “Under ideal circumstances, results of analyzing the final parametric modulator in a sequence (orthogonalized with respect to all others) should be highly similar to analyses in which no serial orthogonalization is performed (Mumford et al., 2015). Practically, however, we have observed differences in the strength of our results with and without serial orthogonalization. While GLMs that include both information and reward regressors may be able to dissociate information and reward signals without the analysis stream we describe here, we elected to adopt serial orthogonalization in order to ensure variance that could be attributed to the final parameter was instead allocated elsewhere.”

- For the sake of completeness and transparency, it would be helpful to include some of the findings of control analyses, including (1) whole-brain maps for some of the control GLMs tested, even if they go in the supplement or online; and (2) for ROI analyses of vmPFC and dACC, the effect sizes, t-statistics, and p-values of the other independent variables included in the respective GLMs. As examples, it would be informative to present the -SD, switch – stay, and explore – exploit effects in dACC and vmPFC (pg 12) and the negative effect of GLM 8 (choice probability). Currently only the relative reward and information gain terms from these GLMs are reported for these ROI analyses. Finally, I recommend reporting the analysis of relative reward on equal information trials in the dACC ROI.

1) We have added the whole-brain maps in the Supplementary Files 6-10.

2) No activity was observed in the negative contrast of GLM8. New text: “The negative contrast revealed no activity at uncorr p < 0.001.” Likewise, no activity was observed in the negative contrast for GLM4diff. New text: “No activity in dACC (t(19) = 0.7498, p = 0.4625) and vmPFC ROIs (t(19) = -0.4423, p = 0.6633 was observed in the negative contrast.”

3) Requested analysis has bene added at pg. 9. New text: “We also focus the analysis on trials in which subjects had equal information about the options (equal information conditions) and we observed no activity in dACC ROI associated to RelReward (t(19) = -0.0297, p = 0.9766).”

- It should be clarified that information gain is confounded with decisions to explore in this task design and the accompanying analysis. As acknowledged in the authors' reply: "Certainly if we limited ourselves to considering only those trials in which it is unambiguous as to whether the subjects were exploring or exploiting via their choices, the explore/exploit covariate the reviewer requests would eliminate the effect of I(c) simply because I(c) and directed exploration describe the same thing (picking the option about which least is known)." As the reviewer understood it at least, based on the title and abstract, the paper claims there are dedicated and independent information and reward signals in PFC. Other existing theories in the literature have argued that the role of the dACC is to decide to explore or to switch between exploratory and exploitative bouts (e.g. Karlson et al., Science, 2012). While these are clearly related, this is not synonymous with an "information system" that encodes the expected information gain of any decision at hand, or perhaps even when no decision is made at all. Other studies have shown it is possible to successfully dissociate related relative uncertainty terms from the decision to explore (e.g. Badre et al., Neuron, 2011; Trudel et al., Nature Human Behavior, 2021). Interestingly this latter study showed that the dACC (and vmPFC) signal related to relative uncertainty does in fact depend on whether subjects are in an exploratory or exploitative modes, defined largely based on the planning horizon. Thus, in the view of reviewers, these are not the same thing, and it should be acknowledged that this task design cannot in fact dissociate these ideas.

We have added this clarification in the current version of the manuscript. The new text read as follows: “We would like to acknowledge, however, that while dACC activity associated to Information Gain in our task is not affected by proxies of exploratory decisions (e.g., switch-stay analysis and Default vs. NoDefualt analsysis), our task cannot dissociate decisions to explore to gain information (i.e., directed exploration) and Information Gain. This is because Information Gain and directed exploration in our task describe the same thing – picking the option about which least is known.”

- It should be clarified whether the switch regressor was coded with respect to the previous free choice or the previous forced choice?

We have clarified this in the revised version. The new text reads as follows: “[..] we entered a switch-stay regressor as proxy for switching behavior (i.e., coded 0 if choices on the first free choice trial were the same as previous forced trial choices and 1 otherwise).”

- The authors make good points about their conclusions in paragraphs 2 and 3 of their discussion. I would suggest some rephrasing about the ways they refer to the previous literature, though, as the current phrasing (perhaps unintentionally) makes it sound like they are opposing certain previous work, when their results actually seem quite compatible with it. For example, they cite previous work suggesting that information and reward have similar neural circuitry, then say that this "apparent" overlap in neural substrates in only observed when a confounded analysis is used that does not account for their shared variance. This sounds like they are implying that the previous findings of overlap were simply due to confounds. This doesn't make sense to me. First, the reason reward and information are intermixed in this paper, and need to be deconfounded by an analysis approach, is due to the particular experimental design where people can choose based on reward, information, or a combination of the two, leading to correlations between these variables. This is an important point since this is a common task design relevant in real life. However, most of the studies they cite do not have designs that would fall prey to this (e.g. Kang et al. and Gruber et al. where no rewards were provided from the task used for fMRI, Iigaya et al. where choices only affected information and not reward, etc.). Second, the current paper does report overlap between information and reward as well (in the striatum), and several of the papers they cite as reporting "apparent" overlap also reported it in nearby regions of striatum, or in midbrain regions or neural populations which provide input to striatum and are thought to be a major contributor to its BOLD signals in tasks like this (e.g. Kobayashi and Hsu, Charpentier et al., Bromberg-Martin and Hikosaka, Tricomi and Fiez, etc.).So rather than this paper's findings suggesting that aspect of previous work was confounded, instead this paper replicates and supports that previous work. I would rephrase these paragraphs to make it clear that only certain experimental tasks need to be corrected for shared variance (and that it is in the context of such tasks where their finding of ACC vs vmPFC differences is interesting), and that their results are in agreement with some areas having overlap.

We have modified the text which reads as follows “Here, we show that the overlapping activity in PFC elicited between these two adaptive signals by our task design is only observed if their shared variance is not taken into account.” And additionally, “Our results further suggest that these independent value systems interact in the striatum, consistent with its hypothesized role in representing expected policies ^64^ and information-related cues ^41 24 21^”

- line 464 to the end of the paragraph, these statements seem a little off so I would suggest changing or removing them. I am not sure how this work speaks to multidimensional versus "pure" value encoding in striatum, at least in the way indicated by the cited studies. Those studies argued that single neurons encode multiple types of value-like variables or both value and non-value variables, while this study can't distinguish with its current approach (e.g. do some striatal cells signal Information Gain while others signal Relative Reward, or are those mixed; something like an fMRI-adaptation design could give evidence on this but that is not the goal of the current study). Also, one of the cited works recorded dopamine neurons not striatal neurons, though I suppose they may influence striatal BOLD as I mentioned above. Finally, this study and the cited study do not speak to the question of whether the basal ganglia use a softmax function since both of them simply built a softmax into their analysis without attempting to test softmax against competing functions (which is fine on its own, since identifying the function was not the goal of those studies, so they should be interpreted as such).

We have removed the sentence in the revised version of the manuscript.